# Identification of Causal Relationships in Linear Cyclic Models with Latent Variables

## Abstract

Causal discovery aims to model the intricate mechanisms underlying complex systems. Numerous methods have been proposed to identify causal relationships from observational data, but they often assume that the causal model is acyclic and all variables are observed. Those methods risk yielding misleading or spurious causal relationships when confronted with the challenges posed by cycles and latent variables. To address these challenges, we propose a novel method that leverages higher-order cumulants to recover the causal structure among observed variables, even in the presence of cycles and latent variables. Specifically, we construct two cumulant matrices that incorporate various (joint) cumulants of the observed variables. By utilizing these matrices, we provide identifiability theories that determine the existence of cycles and latent variables based on the rank differences of the constructed cumulant matrix, and determine the causal relationship between two observed variables. This innovative method provides a robust framework for accurate causal discovery in complex systems with inherent cyclic and latent structures. Experimental results in simulated and real-world data demonstrate the effectiveness of our proposed method.

## 1 Introduction

Causal discovery is fundamental for modeling the data generation process in complex systems such as biological systems (Benito et al., 2007), and economic processes (Haavelmo, 1943). In real-world scenarios, the causal relationships among observed variables may contain feedback loops (Mason, 2007), and some variables cannot be observed or collected (Chen et al., 2024), which introduces challenges for causal discovery (Spirtes et al., 2000). Most exciting methods (Spirtes et al., 2000; Shimizu et al., 2006), which lead to fundamental limitations when dealing with systems that have feedback mechanisms, as they cannot accurately capture the dynamic equilibrium and cyclic interactions within the system. Additionally, unobserved confounding variables can introduce spurious correlations, further complicating the identification of causal relationships. Thus, identifying the causal relationship in the presence of cycles and latent confounders is very important.

Some methods have relaxed the assumptions of acyclic and causal sufficient (i.e., without latent variables), allowing for the presence of cycles or hidden variables. Although many studies have investigated cyclic causal discovery (Spirtes, 1994; Hyttinen et al., 2012; Mooij & Heskes, 2013; Dai et al., 2024), some limitations remain. The CCD approach (Lacerda et al., 2008b) utilizes Independent Component Analysis (ICA) (Hyvärinen & Oja, 2000) technique to obtain the mixing matrix. But its permuted output matrix becomes non-unique when acyclic assumptions are violated, producing multiple candidate solutions. Additionally, CCD incurs prohibitively high computational costs. To reduce the costs in CCD, CCI (Strobl, 2019) uses conditional independence tests for inferring the cyclic causal model. But it remains that some causal edges cannot be determined, because different causal graphs may imply the same conditional independence condition. Building on Linear Non-Gaussian Acyclic Model (LiNGAM) (Shimizu et al., 2006), several studies (Hoyer et al., 2008; Hyvärinen & Smith, 2013; Tashiro et al., 2014; Wang & Drton, 2020; Chen et al., 2021) can be used to identify causal structures even with latent confounders. And recent cumulant-based approaches (Cai et al., 2023; Chen et al., 2024) establish formal identifiability conditions for edge orientation with known confounders in acyclic cases. However, these methods are unable to handle scenarios where there exist cycles.

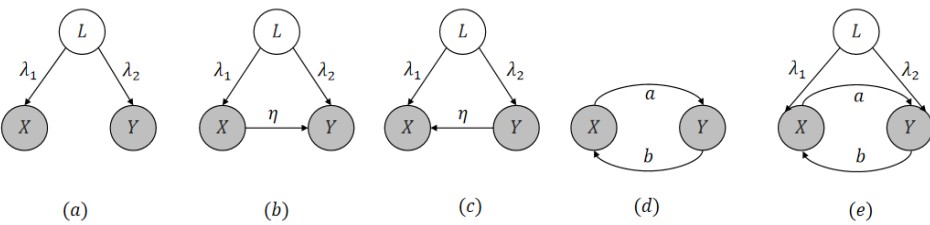

Figure 1: Five causal graphs, where the white node represents the latent variable and the gray node represents the observed variable.

Notably, most existing approaches focus on either latent confounding or cyclic relationships separately, requiring prior knowledge of whether the underlying causal structure contains latent confounders or feedback loops. Although some methods can simultaneously address the issues of cycles and latent variables, they are constraint-based and have limitations in identifying certain causal edges. This limitation motivates us to develop a framework that leverages higher-order cumulants to simultaneously identify direct causal relationships, feedback loops, and latent confounding from observational data, based on the linear non-Gaussian causal model. That is, our goal is to propose a unified approach to simultaneously identify the five types of causal graphs shown in Figure 1.

The key insight stems from the distinct higher-order cumulant patterns generated by different causal structures. Specifically, the statistical dependencies between joint higher-order cumulants directly indicate the existence of causal edges. Crucially, the marginal cumulants of cause and effect variables exhibit different noise characteristics compared to their joint cumulants, which provides intrinsic evidence for causal direction asymmetry. Together, these cumulants provide a powerful tool for identifying latent confounders - their joint behavior changes systematically when unobserved variables affect the measured relationship. Inspired by these, we define two types of cumulant matrices, denoted as $J^{(k_1, k_2)}$ and $M^{(k_1, k_2)}$, where $k_1$ and $k_2$ are two parameters affecting the order of used (joint) cumulants. Then, leveraging the different ranks of the constructed matrices, we propose some identifiablity theories for determining the causal structure between two observed variables, as well as determining whether there exist cycles or latent confounders, or both. Finally, we develop a unified method for identifying the causal relationship between two observed variables in the linear cyclic models with latent variables. It is noted that our approach offers three important advantages: 1) it works with standard observational data, which require neither interventions nor temporal information; 2) it handles all major causal scenarios, direct causation, feedback loops, and confounding, within a unified framework; 3) the method achieves superior accuracy while maintaining computational efficiency. Our experimental results demonstrate the effectiveness and correctness of our proposed method in identifying causal relationships under cyclic and latent confounder conditions, while maintaining robustness across different kinds of noise distributions.

## 2 PRELIMINARY

### 2.1 LINEAR NON-GAUSSIAN MODEL WITH LATENT VARIABLES

The Linear Non-Gaussian Model with Latent Variables strictly adheres to the theoretical framework (Lacerda et al., 2008a) while generalizing both acyclic and cyclic causal structures with latent confounders. Unlike traditional acyclic formulations, this extended framework maintains theoretical consistency when handling cyclic dependencies through carefully constrained feedback mechanisms. The data generation process is formalized as:

$$\mathbf{V} = \mathbf{B}\mathbf{V} + \mathbf{\Lambda}\mathbf{L} + \mathbf{E}. \tag{1}$$

where $\mathbf{V}$ represents the vector of observed variables, $\mathbf{L}$ denotes the vector of latent variables, $\mathbf{B}$ is the causal strength matrix, $\mathbf{\Lambda}$ is the coefficient matrix from $\mathbf{L}$ to $\mathbf{V}$, $\mathbf{E}$ is the vector of independent non-Gaussian noises. There is a key difference between the acyclic model and the cyclic model. For the acyclic model, $\mathbf{B}$ can be permuted to be strictly lower triangular, but this is not necessary for the cyclic model.

The above model can be transformed into the Overcomplete Independent Analysis (OICA) Model (Lewicki & Sejnowski, 2000), where the observed variables are expressed as linear combinations of independent non-Gaussian sources, giving the standard OICA representation $\mathbf{V} = \mathbf{AS}$, where $\mathbf{A}$ denotes the mixing matrix and $\mathbf{S}$ contains all independent non-Gaussian components, including both the latent confounders and noise terms.

For model (1) with cycle, the spectral radius $\rho(B)$ of the causal strength matrix $\mathbf{B}$ must be strictly less than 1. This stability condition ensures two critical requirements. First, the product of coefficients along any feedback loop must remain below unity to prevent system divergence. Second, the matrix $\mathbf{I} - \mathbf{B}$ needs to be invertible to guarantee the existence of a unique equilibrium solution.

For instance, for the data generated according to the causal graph shown in Figure 1(e), $X$ and $Y$ can be formalized as:

$$\begin{bmatrix} X \\ Y \end{bmatrix} = \begin{bmatrix} 0 & b \\ a & 0 \end{bmatrix} \begin{bmatrix} X \\ Y \end{bmatrix} + \begin{bmatrix} \lambda_1 \\ \lambda_2 \end{bmatrix} L + \begin{bmatrix} E_X \\ E_Y \end{bmatrix}$$

$$= \begin{bmatrix} \alpha_1 & \beta_1 & \gamma_1 \\ \alpha_2 & \beta_2 & \gamma_2 \end{bmatrix} \begin{bmatrix} L \\ E_X \\ E_Y \end{bmatrix} \tag{2}$$

$$= \frac{1}{1-ab} \begin{bmatrix} \lambda_1 + b\lambda_2 & 1 & b \\ \lambda_2 + a\lambda_1 & a & 1 \end{bmatrix} \begin{bmatrix} L \\ E_X \\ E_Y \end{bmatrix},$$

where $\alpha_1$, $\alpha_2$, $\beta_1$, $\beta_2$, $\gamma_1$ and $\gamma_2$ are the mixing coefficients of noise to observed variables. $E_X$ and $E_Y$ are the noises of $X$ and $Y$, respectively. For the system in Figure 1(e), the spectral radius $\rho(B)$ of the feedback matrix $B = \begin{bmatrix} 0 & b \\ a & 0 \end{bmatrix}$, whose spectral radius $\rho(B) = \sqrt{|ab|}$ is governed by the product of cyclic coefficients $ab$ along $X \to Y \to X$. The stability requirement $\rho(B) = \sqrt{|ab|} < 1$ is equivalent to $|ab| < 1$, ensuring both the existence of the resolvent $(I - B)^{-1} = (1-ab)^{-1} \begin{bmatrix} 1 & b \\ a & 1 \end{bmatrix}$ and the exponential decay of multi-cycle amplifications. This matrix inverse systematically scales all equilibrium relationships through the $(1-ab)^{-1}$ factor.

Specifically, when a self-loop exists, an observed variable $X_a$ is determined by:

$$X_a = b_{aa}X_a + \sum_{k \neq a, k=1}^{n} b_{ak}X_k + E_a, \tag{3}$$

where $n$ is the number of observed variables. When $b_{aa} \neq 1$, Eq. (3) can be rewritten as:

$$(1 - b_{aa})X_a = \sum_{k \neq a, k=1}^{n} b_{ak}X_k + E_a, \tag{4}$$

$$X_a = \frac{1}{(1 - b_{aa})} \sum_{k \neq a, k=1}^{n} b_{ak}X_k + E_a. \tag{5}$$

## 2.2 CUMULANT

Cumulants provide a complete characterization of joint probability distributions through their ability to capture higher-order statistical dependencies. For a random vector $X$, its cumulants are formally defined as:

**Definition 1** (**Cumulants** Brillinger (2001)). *Let $X = (X_1, X_2, \cdots, X_n)$ be a random vector of length $n$. The $k$-th order cumulant tensor of $X$ is defined as a table $n \times \cdots \times n$ ($k$ times) table $C^{(k)}$, whose entry at position $(i_1, \cdots, i_k)$ is*

$$C_{i_1, -\cdots, i_k}^{(k)} = Cum(X_{i_1}, \cdots, X_{i_k}) = \sum_{(D_1, \cdots, D_h)} (-1)^h (h-1)! E\left[\prod_{j \in D_i} X_j\right] \cdots E\left[\prod_{j \in D_h} X_j\right], \tag{6}$$

*where the sum is taken over all partitions $(D_1, \cdots, D_h)$ of set $\{i_1, \cdots, i_k\}$.*

For convenience, we use $C_i(X)$ to denote marginal cumulant $Cum(\underbrace{X, \cdots, X}_{i \ times})$, and use $C_{i,j}(X,Y)$ to denote joint cumulant $Cum(\underbrace{X, \cdots, X}_{i \ times}, \underbrace{Y, \cdots, Y}_{j \ times})$. For example, $C_4(X)$ represents $Cum(X,X,X,X)$, $C_{3,1}(X,Y)$ represents $Cum(X,X,X,Y)$. If each variable $X_i$ has zero mean, then the sum of the partitions with size 1 is 0 and can be omitted. Note that the first-order cumulant of a variable is the mean of the variable, and the second-order cumulant of variables corresponds to their covariance. Without loss of generality, we assume that the variables have a mean of 0 and a variance of 1.

For two variables $X$ and $Y$ generated by the structural equations in Eq. (2), their third-order cumulants are:

$$Cum(X,X,X) = (1-ab)^{-3}[(\lambda_1 + b\lambda_2)^3 C_3(L) + C_3(E_X) + b^3 C_3(E_Y)],$$
$$Cum(X,X,Y) = (1-ab)^{-3}[(\lambda_1 + b\lambda_2)^2(\lambda_2 + a\lambda_1)C_3(L) + aC_3(E_X) + b^2 C_3(E_Y)],$$
$$Cum(X,Y,Y) = (1-ab)^{-3}[(\lambda_1 + b\lambda_2)(\lambda_2 + a\lambda_1)^2 C_3(L) + a^2 C_3(E_X) + bC_3(E_Y)].$$

## 3 INTUITION

For any two observed variables $X$ and $Y$ in Figure 1, our approach leverages a key observation: higher-order cumulants exhibit distinct patterns depending on the underlying causal structure. We first demonstrate that joint cumulants of the same order share identical higher-order noise terms while preserving the systematic relationships among their coefficients. As shown for all causal structures in Figure 1, the $k$ order joint cumulants can be expressed as :

$$\begin{bmatrix} C_{k-1,1}(X,Y) \\ C_{k-2,2}(X,Y) \\ \vdots \\ C_{1,k-1}(X,Y) \end{bmatrix} = \begin{bmatrix} \alpha_1^{(k-1)}\alpha_2 & \beta_1^{(k-1)}\beta_2 & \gamma_1^{(k-1)}\gamma_2 \\ \alpha_1^{(k-2)}\alpha_2^2 & \beta_1^{(k-2)}\beta_2^2 & \gamma_1^{(k-2)}\gamma_2^2 \\ \vdots & \vdots & \vdots \\ \alpha_1\alpha_2^{(k-1)} & \beta_1\beta_2^{(k-1)} & \gamma_1\gamma_2^{(k-1)} \end{bmatrix} \begin{bmatrix} C_k(L) \\ C_k(E_X) \\ C_k(E_Y) \end{bmatrix}. \tag{7}$$

The coefficient matrix in Eq. (7) exhibits a systematic row-wise proportionality: adjacent rows of the coefficient matrix maintain fixed multiplicative ratios across all noise components. Formally, for any two consecutive rows $R_i$ and $R_{i+1}$, their elements satisfy:

$$R_{i+1} = R_i \begin{bmatrix} \alpha_1^{-1}\alpha_2 & 0 & 0 \\ 0 & \beta_1^{-1}\beta_2 & 0 \\ 0 & 0 & \gamma_1^{-1}\gamma_2 \end{bmatrix}. \tag{8}$$

When the causal edge between the observed variables is absent (*i.e.*, $\beta_1\beta_2 = \gamma_1\gamma_2 = 0$), the row transformation in Eq. (8) collapses to a scalar operation: $R_{i+1} = \alpha_1^{-1}\alpha_2 R_i$. This leads to the matrix $J^{(k_1,k_2)}$ having rank 1, with rank increasing to 2 when $X$ and $Y$ are directly connected. The explicit form of $J^{(k_1,k_2)}$ is given by:

$$J^{(k_1,k_2)} = \begin{pmatrix} C_{k_1-1,1}(X,Y) & C_{k_1-2,2}(X,Y) \\ \vdots & \vdots \\ C_{2,k_1-2}(X,Y) & C_{1,k_1-1}(X,Y) \\ \overline{C_{k_2-1,1}(X,Y)} & \overline{C_{k_2-2,2}(X,Y)} \\ \vdots & \vdots \\ C_{2,k_2-2}(X,Y) & C_{1,k_2-1}(X,Y) \end{pmatrix}, \tag{9}$$

where the parameters $k_1$ and $k_2$ satisfy $3 \leq k_1 < k_2$.

The above analysis demonstrates that the relationships between joint cumulants can be used to determine the existence of edges. By examining the relationships between marginal and joint cumulants, we can infer the causal direction between two observed variables. To formalize the relationship between marginal and joint cumulants for variables $X$ and $Y$ in Figure 1, we first derive their respective

marginal cumulant expansions:

$$
\begin{bmatrix} C_k(X) \\ C_k(Y) \\ C_{i,j}(X,Y) \end{bmatrix} = \begin{bmatrix} \alpha_1^{(k)} & \beta_1^{(k)} & \gamma_1^{(k)} \\ \alpha_2^{(k)} & \beta_2^{(k)} & \gamma_2^{(k)} \\ \alpha_1^{(i)}\alpha_2^{(j)} & \beta_1^{(i)}\beta_2^{(j)} & \gamma_1^{(i)}\gamma_2^{(j)} \end{bmatrix} \begin{bmatrix} C_k(L) \\ C_k(E_X) \\ C_k(E_Y) \end{bmatrix},
\tag{10}
$$

where the parameters $i$ and $j$ must satisfy $i + j = k, i, j > 0$. We find that in the presence of edges while maintaining acyclic structure, the joint cumulant $C_{(i,j)}(X,Y)$ contains identical higher-order noise terms to those appearing in the higher-order cumulants of causal variables. Under the causal direction $X \rightarrow Y$ with $\gamma_1 = 0$, as shown in Figure 1(b), $C_k(X)$ and $C_{i,j}(X,Y)$ share identical higher-order noise terms $C_k(L)$ and $C_k(E_X)$, while $C_k(Y)$ uniquely contains the additional noise term $C_k(E_Y)$ due to the absence of any backward edge from $Y$ to $X$. Similar conclusions on inferring the causal direction $Y \rightarrow X$, as shown in Figure 1(c).

In the presence of cycles, $E_X$ can propagate to $Y$ and $E_Y$ to $X$ through feedback paths, causing $C_k(X), C_k(Y)$, and $C_{i,j}(X,Y)$ to share identical higher-order noise terms. This bidirectional noise transmission eliminates the distinctive asymmetry that can be observed in acyclic causal structures.

Our analysis demonstrates how extending the matrix $J^{(k_1,k_2)}$ to include marginal higher-order cumulants of both $X$ and $Y$ creates a unified framework for causal discovery. This enhanced approach simultaneously identifies causal directions through asymmetric noise distributions and detects cyclic structures via symmetric noise sharing across cumulants. The extension preserves the original matrix structure while adding new capabilities to map cumulant patterns to graph properties. The expanded matrix $M_{v_1}^{(k_1,k_2)}$ is:

$$
M_{v_1}^{(k_1,k_2)} = \begin{pmatrix} C_{k_1}(v_1) & C_{k_1-1,1}(v_1,v_2) & C_{k_1-2,2}(v_1,v_2) \\ \vdots & \vdots & \vdots \\ C_{3,k_1-3}(v_1,v_2) & C_{2,k_1-2}(v_1,v_2) & C_{1,k_1-1}(v_1,v_2) \\ \hline C_{k_2}(v_1) & C_{k_2-1,1}(v_1,v_2) & C_{k_2-2,2}(v_1,v_2) \\ \vdots & \vdots & \vdots \\ C_{3,k_2-3}(v_1,v_2) & C_{2,k_2-2}(v_1,v_2) & C_{1,k_2-1}(v_1,v_2) \end{pmatrix}
\tag{11}
$$

where the parameters $k_1$ and $k_2$ must satisfy $3 \le k_1 < k_2$ and $v_1, v_2$ represent two distinct random variables. The matrices $M_X$ and $M_Y$ exhibit distinct algebraic characteristics that correspond to their structural roles in Figure 1. In Figure 1(b), this manifests as a rank disparity with $rank(M_X) = 2$ and $rank(M_Y) = 3$.

## 4 IDENTIFIABILITY AND METHOD

### 4.1 IDENTIFIABILITY

Inspired by the intuition, we extend the matrices $J^{(k_1,k_2)}$ and $M^{(k_1,k_2)}$ to the case with $l$ latent variables, and define the $J^{(k_1,k_2)}$ and $M^{(k_1,k_2)}$ as:

$$
J^{(k_1,k_2)} = \begin{pmatrix} C_{k_1-1,1} & C_{k_1-2,2} & \cdots & C_{k_1-l-1,l+1} \\ \vdots & \vdots & \ddots & \vdots \\ C_{l+1,k_1-l-1} & C_{l,k_1-l} & \cdots & C_{1,k_1-1} \\ \hline C_{k_2-1,1} & C_{k_2-2,2} & \cdots & C_{k_2-l-1,l+1} \\ \vdots & \vdots & \ddots & \vdots \\ C_{l+1,k_2-l-1} & C_{l,k_2-l} & \cdots & C_{1,k_2-1} \end{pmatrix},
\tag{12}
$$

$$
M^{(k_1,k_2)} = \begin{pmatrix} C_{k_1} & C_{k_1-1,1} & C_{k_1-2,2} & \cdots & C_{k_1-l-1,l+1} \\ \vdots & \vdots & \vdots & \ddots & \vdots \\ C_{l+2,k_1-l-2} & C_{l+1,k_1-l-1} & C_{l,k_1-l} & \cdots & C_{1,k_1-1} \\ \hline C_{k_2} & C_{k_2-1,1} & C_{k_2-2,2} & \cdots & C_{k_2-l-1,l+1} \\ \vdots & \vdots & \vdots & \ddots & \vdots \\ C_{l+2,k_2-l-2} & C_{l+1,k_2-l-1} & C_{l,k_2-l} & \cdots & C_{1,k_2-1} \end{pmatrix}.
\tag{13}
$$

Leveraging the rank of $J^{(k_1,k_2)}$ and $M^{(k_1,k_2)}$, the following theorems establish necessary and sufficient conditions for identifying causal directions between observed variables generated by Eq. (1).

**Theorem 1.** *Assume that two observed variables $X$ and $Y$ are generated by Eq.(1), and they share $l$ latent confounders. For any integers $k_1, k_2$ satisfying $l + 2 \leq k_1, 2l + 2 \leq k_2$, there is no directed edge between $X$ and $Y$ if and only if $rank(J^{(k_1,k_2)}) = l$.*

Theorem 1 establishes a fundamental criterion for detecting the existence of directed edges between observed variables, When we cannot identify that there is no directed edge between variables according to Theorem 1, it becomes necessary to further investigate both the direction of potential causal edges and the possible existence of cyclic structures in the relationship between these variables.

**Theorem 2.** *Assume that two observed variables $X$ and $Y$ are generated by Eq. (1), and they share $l$ latent confounders. For any integers $k_1, k_2$ satisfying $l + 2 \leq k_1, 2l + 2 \leq k_2$, $X$ is a cause of $Y$ if and only if $rank(M_X^{(k_1,k_2)}) = l + 1$ and $rank(M_Y^{(k_1,k_2)}) = l + 2$.*

Theorem 2 collectively establish a complete characterization for unidirectional causal relationships between $X$ and $Y$. Specifically, the rank conditions on $M_X$ and $M_Y$ not only determine the existence of causal edges but also precisely identify their directionality through the asymmetry in matrix ranks. This rank asymmetry pattern, characterized by the causal variable's matrix $M^{(k_1,k_2)}$ has rank $l + 1$ while the effect variable's matrix $M^{(k_1,k_2)}$ has rank $l + 2$, provides a mathematical signature of unidirectional causality.

**Theorem 3.** *Assume that two observed variables $X$ and $Y$ are generated by Eq. (1), and they share $l$ latent confounders. For any integers $k_1, k_2$ satisfying $l + 2 \leq k_1, 2l + 2 \leq k_2$, $X$ and $Y$ form a cycle if and only if $rank(J^{(k_1,k_2)}) = l + 1, rank(M_X^{(k_1,k_2)}) = rank(M_Y^{(k_1,k_2)}) = l + 2$.*

Theorem 3 establishes a criterion to identify the cyclic structure in the presence of latent variables. According to Theorem 1-3, we can identify the existence of a causal edge, the direction of the causal edge, and the existence of cyclic structure in the presence of latent variables. So the causal structure between observed variables can be identified. The identifiability is summarized in Theorem 4.

**Theorem 4.** *Assume that two observed variables $X$ and $Y$ are generated by Eq. (1), and they share $l$ latent confounders. Then the causal structure between observed variables can be identified.*

## 4.2 METHOD

Based on the above identifiability results, we now present a practical method for inferring causal structure from observed data with latent confounders. The proposed algorithm employs an iterative procedure to simultaneously determine the number of latent variables and the causal relationships between observed variables. For the case of one latent confounder, we employ third-order and fourth-order cumulants $J^{(3,4)}$ to identify the existence of edges between observed variables, while their extended moment conditions $M_X^{(3,4)}$ and $M_Y^{(3,4)}$ determine both edge directions and cyclic structures. The extension to the case with more than one latent confounder is achieved by expanding the $J^{(k_1,k_2)}$ matrix with additional cumulant terms as shown in Eq. (12), and augmenting it with Eq. (13) constraints to form $M^{(k_1,k_2)}$. Then firstly, we will choose $m = 1, k_1 = 3, k_2 = 4$ for the initial calculation of $J^{(k_1,k_2)}, M_X^{(k_1,k_2)}$ and $M_Y^{(k_1,k_2)}$. When $rank(J^{(k_1,k_2)}) = m$ and $rank(J^{(k_1+1,k_2+2)}) = m$ are satisfied, we consider $X$ and $Y$ to be independent. If $rank(M_X^{(k_1,k_2)}) = m + 1$ and $rank(M_Y^{(k_1,k_2)}) = m + 2$ are satisfied, then $X$ is a source of $Y$ and vice versa. When $rank(J^{(k_1,k_2)}) \neq m$ and $rank(M_X^{(k_1,k_2)}) = rank(M_Y^{(k_1,k_2)}) = M_X^{(k_1+1,k_2+2)} = m + 2$ are satisfied, we identify that $X$ and $Y$ form a cycle . If none of the above conditions are met, then let $m = m + 1, k_1 = m + 2, k_2 = 2m + 2$, and it calculate the new $J^{(k_1,k_2)}, M_X^{(k_1,k_2)}$ and $M_Y^{(k_1,k_2)}$ until one of the conditions is met. The pseudo-code of our proposed algorithm is summarized in Algorithm 1 in Appendix C.

# 5 EXPERIMENT

## 5.1 SIMULATION DATA

To evaluate the performance of our proposed method, we conducted experiments on synthetic data, considering the following three cases:

[Case 1]: A causal graph over two observed variables that are influenced by a single latent variable, where the observed variables have either no edge between them or only one directed edge, i.e., Figure 1(a).

[Case 2]: A causal graph over two observed variables sharing a single latent confounder, with only one directed edge between them, i.e., Figures 1(b) - 1(c).

[Case 3]: A causal graph over two observed variables without latent variables, where the observed variables must form a directed cycle, as depicted in the figure, i.e., Figure 1(d).

[Case 4]: A causal graph over two observed variables that are influenced by a single latent variable, where the observed variables must form a directed cycle, as depicted in the figure, i.e., Figure 1(e).

The causal coefficients were sampled uniformly from $[-1.5, -0.5] \cup [0.5, 1.5]$. For the noise terms, we employed three different probability distributions: Exponential Distribution, Gamma Distribution with shape parameter $k = \frac{3}{2}$, and Gumbel Distribution. For each model, the sample size N is varied among $\{5000, 10000, 50000, 100000\}$. For each setting, we generate 100 datasets.

To further validate the generalization capability of our method, we also constructed 100 hybrid scenarios by randomly combining the five causal structures in Figure 1, with coefficients and confounding strengths in $[-1.5, -0.5] \cup [0.5, 1.5]$, random edge directions, and sample sizes varying across $\{5000, 10000, 50000, 100000\}$. All experiments are conducted on a machine equipped with the Ubuntu 16.04.5 LTS operating system with an Intel Core i7-6700 CPU, 15GB of RAM.

## 5.2 BASELINE METHODS AND EVALUATION METRICS

In our experiments, we use DirectLiNGAM (DL) (Shimizu et al., 2011), cumulant-based method (Chen et al., 2024) and ReLiNGAM (Re) (Schkoda et al., 2024) and CCI (Strobl, 2019) that can handle cyclic structures, as the baseline methods. Among these methods, each represents a distinct approach to causal discovery with unique theoretical foundations and application scopes.

DirectLiNGAM implements a regression-based approach that sequentially identifies exogenous variables through residual independence tests, providing a deterministic solution for acyclic causal structures under the causal sufficiency assumption. Chen's method employs higher-order cumulants to identify causal directions in the presence of latent confounders, offering theoretical identifiability guarantees for acyclic systems with non-Gaussian noise distributions. CCI adopts a constraint-based framework combining conditional independence tests with rank-based criteria, specifically designed to handle both cyclic relationships and latent confounding effects.

We examine how accurately each method reproduces the true causal graph across varying sample sizes, counting only perfect matches where all edges and directions are correct. This binary accuracy measure is assessed at significance levels ranging from $\alpha = 0.2$ to $\alpha = 0.01$.

**Evaluation on acyclic causal structures.** The experimental results for acyclic structures with latent confounders are presented in Table 1 for Case 1 and Case 2. Across both scenarios, DirectLiNGAM shows limited effectiveness in Case 1 due to its fundamental difficulty in separating latent confounding effects from potential causal edges between observed variables, while demonstrating better performance in Case 2, where it achieves approximately 60% accuracy in identifying the correct directed edge. Chen's method exhibits gradual improvement with larger sample sizes in both cases, but its accuracy remains constrained by the precision of its parameter estimation, particularly for weaker edge strengths. ReLiNGAM achieves approximately 0.50 accuracy for no-edge detection and around 0.60 accuracy for unidirectional edges. This performance limitation stems from its two-stage estimation process: while cumulant-based methods can identify the causal ordering, they still require subsequent parameter estimation to determine both the existence and strength of edges between variables. The accuracy of edge detection is therefore constrained by the precision of these parameter estimates, particularly for weaker causal effects where estimation uncertainty is greater.

Table 1: Accuracy of the different methods varies across different distributions in Case 1 and 2.

| | | Case 1 | | | | | Case 2 | | | |
|---|---|---|---|---|---|---|---|---|---|---|
| | Size | DL | Chen | Re | CCI | Ours | DL | Chen | Re | CCI | Ours |
| Exp | 5000 | 0.00 | 0.28 | 0.44 | 0.00 | **0.72** | 0.60 | 0.46 | 0.59 | 0.00 | **0.77** |
| | 10000 | 0.00 | 0.36 | 0.49 | 0.00 | **0.75** | 0.62 | 0.54 | 0.60 | 0.00 | **0.86** |
| | 50000 | 0.00 | 0.57 | 0.45 | 0.00 | **0.92** | 0.61 | 0.55 | 0.57 | 0.00 | **0.91** |
| | 100000 | 0.00 | 0.67 | 0.49 | 0.00 | **0.95** | 0.62 | 0.55 | 0.58 | 0.00 | **0.95** |
| Gumble | 5000 | 0.00 | 0.24 | 0.30 | 0.00 | **0.70** | 0.62 | 0.37 | 0.67 | 0.00 | **0.70** |
| | 10000 | 0.00 | 0.29 | 0.32 | 0.00 | **0.79** | 0.59 | 0.38 | 0.63 | 0.00 | **0.78** |
| | 50000 | 0.00 | 0.52 | 0.39 | 0.00 | **0.87** | 0.59 | 0.41 | 0.65 | 0.00 | **0.92** |
| | 100000 | 0.00 | 0.65 | 0.42 | 0.00 | **0.95** | 0.59 | 0.40 | 0.68 | 0.00 | **0.94** |
| Gamma | 5000 | 0.00 | 0.26 | 0.38 | 0.65 | **0.66** | 0.59 | 0.25 | 0.58 | 0.00 | **0.66** |
| | 10000 | 0.00 | 0.22 | 0.43 | **0.82** | 0.77 | 0.60 | 0.32 | 0.56 | 0.00 | **0.79** |
| | 50000 | 0.00 | 0.45 | 0.36 | 0.85 | **0.91** | 0.60 | 0.32 | 0.61 | 0.00 | **0.92** |
| | 100000 | 0.00 | 0.57 | 0.33 | 0.86 | **0.93** | 0.62 | 0.33 | 0.59 | 0.00 | **0.95** |

Table 2: Accuracy of the different methods varies across different distributions in Case 3 and 4.

| | | Case 3 | | | | | Case 4 | | | |
|---|---|---|---|---|---|---|---|---|---|---|
| | Size | DL | Chen | Re | CCI | Ours | DL | Chen | Re | CCI | Ours |
| Exp | 5000 | 0.00 | 0.00 | 0.00 | 0.89 | **0.91** | 0.00 | 0.00 | 0.00 | 0.89 | **0.94** |
| | 10000 | 0.00 | 0.00 | 0.00 | 0.94 | **0.96** | 0.00 | 0.00 | 0.00 | 0.94 | **0.98** |
| | 50000 | 0.00 | 0.00 | 0.00 | 0.99 | **1.00** | 0.00 | 0.00 | 0.00 | **0.99** | 0.98 |
| | 100000 | 0.00 | 0.00 | 0.00 | 0.99 | **1.00** | 0.00 | 0.00 | 0.00 | 0.99 | **1.00** |
| Gumble | 5000 | 0.00 | 0.00 | 0.00 | **0.89** | 0.88 | 0.00 | 0.00 | 0.00 | 0.91 | **0.94** |
| | 10000 | 0.00 | 0.00 | 0.00 | 0.92 | **0.94** | 0.00 | 0.00 | 0.00 | 0.92 | **0.98** |
| | 50000 | 0.00 | 0.00 | 0.00 | **0.99** | 0.95 | 0.00 | 0.00 | 0.00 | 0.98 | **0.99** |
| | 100000 | 0.00 | 0.00 | 0.00 | 0.99 | **1.00** | 0.00 | 0.00 | 0.00 | 0.98 | **0.99** |
| Gamma | 5000 | 0.00 | 0.00 | 0.00 | 0.00 | **0.85** | 0.00 | 0.00 | 0.00 | 0.00 | **0.92** |
| | 10000 | 0.00 | 0.00 | 0.00 | 0.00 | **0.91** | 0.00 | 0.00 | 0.00 | 0.00 | **0.95** |
| | 50000 | 0.00 | 0.00 | 0.00 | 0.00 | **0.95** | 0.00 | 0.00 | 0.00 | 0.00 | **0.97** |
| | 100000 | 0.00 | 0.00 | 0.00 | 0.00 | **0.97** | 0.00 | 0.00 | 0.00 | 0.00 | **0.98** |

CCI displays inconsistent performance that varies substantially across noise distributions, working adequately only under Gamma noise conditions where its test assumptions happen to hold.

Our method delivers stable performance across all three noise types, achieving 0.7 accuracy at 5000 samples and 0.8 at 10000 samples. This is because higher-order cumulant estimation requires large sample sizes - insufficient data leads to significant computation errors. With adequate data, our method reaches 0.93-0.95 accuracy at 50000-100000 samples, showing reliable performance in handling latent confounding and correctly identifying edge relationships between observed variables.

**Evaluation on cyclic causal structures.** The results for cyclic structures appear in Table 2 for Case 3 and Case 4. Both DirectLiNGAM, ReLiNGAM and Chen's method show minimal effectiveness in these cyclic scenarios, with accuracy near zero, consistent with their fundamental acyclicity assumptions. CCI demonstrates effective performance with 0.85 to 0.99 accuracy in cyclic scenarios under Exponential and Gumbel noise distributions. However, in the tested Gamma noise conditions for both Case 3 and Case 4, the method shows substantially reduced performance, with accuracy measurements at or near zero in these particular configurations.

Our method maintains stable performance across all tested conditions. In pure cyclic scenarios without latent confounders, its accuracy achieves 0.85 to 0.97. For cyclic scenarios with latent confounders, it attains 0.94 to 0.99 accuracy, demonstrating consistent effectiveness. The method's accuracy reaches 0.99 at larger sample sizes regardless of noise type or the presence of latent con-

Table 3: Accuracy of the different methods varies across different distributions in hybrid structures.

| | Sample Size | DirectLiNGAM | Chen | ReLiNGAM | CCI | Ours |
|---|---|---|---|---|---|---|
| Exponential | 5000 | 0.10 | 0.10 | 0.21 | 0.52 | **0.84** |
| | 10000 | 0.12 | 0.11 | 0.21 | 0.59 | **0.88** |
| | 50000 | 0.08 | 0.13 | 0.23 | 0.59 | **0.90** |
| | 100000 | 0.10 | 0.16 | 0.22 | 0.60 | **0.88** |
| Gumbel | 5000 | 0.11 | 0.11 | 0.16 | 0.52 | **0.70** |
| | 10000 | 0.10 | 0.07 | 0.17 | 0.59 | **0.78** |
| | 50000 | 0.11 | 0.07 | 0.15 | 0.60 | **0.87** |
| | 100000 | 0.11 | 0.09 | 0.15 | 0.60 | **0.83** |
| Gamma | 5000 | 0.11 | 0.07 | 0.22 | 0.22 | **0.79** |
| | 10000 | 0.10 | 0.12 | 0.23 | 0.20 | **0.87** |
| | 50000 | 0.12 | 0.08 | 0.22 | 0.23 | **0.87** |
| | 100000 | 0.09 | 0.13 | 0.25 | 0.22 | **0.90** |

founders. Compared to these results, our approach demonstrates more uniform performance across the full range of tested conditions.

**Evaluation on hybrid causal structures.** As shown in Table 3 , the experimental results demonstrate notable performance variations across methods when evaluating mixed causal structures. Both DirectLiNGAM and Chen's method show constrained performance, with accuracy remaining around 0.10. The accuracy of ReLiNGAM ranges between 0.15 and 0.25, primarily due to their theoretical foundations not accounting for combined cyclic and confounded relationships. The CCI method achieves a competitive accuracy of 0.60 under exponential and Gumbel noise conditions, which can be attributed to its excellent performance in identifying cyclic structures. However, its performance becomes more variable in Gamma noise environments, where accuracy decreases to around 0.25. Our approach outperforms others. With 5000 samples, our method achieves 0.70 accuracy, and this performance improves to around 0.90 accuracy when sample sizes exceed 10000. The results show that our method maintains more consistent accuracy levels across different noise distributions with 0.90 accuracy, while the performance of CCI drops to around 0.25.

### 5.3 REAL PSYCHOLOGICAL DATA

In this section, we apply our approach to real psychological data (McNally et al., 2017) to assess our practical applicability. The dataset is used for examining the causal relationships between symptoms of obsessive-compulsive disorder (OCD) and depression, which consists of 408 observations and 26 variables, comprising 16 depression symptoms and 10 OCD symptoms, all with no missing values. Based on the findings, we specifically investigate three pairwise relationships. Our method revealed that SAD $\rightarrow$ SCD and ANH $\rightarrow$ SCD, which aligns with the seminal finding (Beck et al., 1985; Gillissie et al., 2023). Furthermore, we identified that ANH and SAD form a cycle, corroborating longitudinal findings (Lo et al., 2025; McNally et al., 2017) on the reciprocal relationship between anhedonia and sadness. These demonstrate the validity and effectiveness of our approach.

## 6 CONCLUSION

This paper introduces a novel framework for causal discovery in complex scenarios involving cycles and latent confounders, based on cumulant matrix rank analysis. By systematically investigating the mathematical properties of marginal and joint cumulants between cause and effect variables, we establish a groundbreaking theoretical foundation that reveals a deterministic correspondence between cumulant matrix rank characteristics and underlying causal structures. Under non-Gaussian assumptions, the method identifies distinct rank patterns for different causal scenarios, providing a powerful tool for disentangling causal relationships. Experimental results demonstrate the method's exceptional performance in challenging conditions, including scenarios with latent confounders and cyclic dependencies. Future research will focus on extending this innovative approach to more general and complex causal discovery settings.

## 7 ETHICS STATEMENT

This work adheres to the ICLR Code of Ethics. The study utilizes only publicly available, well-established benchmark datasets. Their use is in full compliance with the respective licenses and terms of use, and they contain no personally identifiable information. The research involved no human subjects or animal experimentation, and we have taken measures to address potential biases in our analysis.

## 8 RCREPRODUCIBILITY STATEMENT

To facilitate the reproducibility of our results, we commit to releasing the full source code and the datasets used in this study upon the acceptance of this paper. Our implementation details and parameter settings are thoroughly described in Section 5 to provide sufficient clarity for replication.

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

## A PROOFS

### A.1 PROOF OF THEOREM 1

**Theorem 1** Assume that two observed variables $X$ and $Y$ are generated by Eq.(1), and they share $l$ latent confounders. For any integers $k_1, k_2$ satisfying $l + 2 \leq k_1, 2l + 2 \leq k_2$, there is no directed edge between $X$ and $Y$ if and only if $rank(J^{(k_1,k_2)}) = l$.

*Proof.* Suppose $X$ and $Y$ are generated by Eq. (1), and they share $l$ latent confounders. Then the generating process of $X$ and $Y$ can be written as:

$$X = \lambda_{11}L_1 + \lambda_{12}L_2 + \cdots + \lambda_{1l}L_l + E_X + \eta_2 E_Y = \sum_{k=1}^{l} \lambda_{1k}L_k + \eta_{12}Y + E_x,$$

$$Y = \lambda_{21}L_1 + \lambda_{22}L_2 + \cdots + \lambda_{2l}L_l + \eta_1 E_X + E_Y = \sum_{k=1}^{l} \lambda_{2k}L_k + \eta_{21}X + E_Y, \tag{14}$$

where $\lambda_{1k}$ and $\lambda_{2k}$ represent the direct causal strengths from the latent variable $L_k$ to observed variables $X$ and $Y$, while $\eta_{21}$ is the direct causal strength from $X$ to $Y$, $\eta_{12}$ is the direct causal strength from $Y$ to $X$. Then the Eq. (14) can be written as Overcomplete Independence Component Analysis (OICA) Model:

$$\begin{bmatrix} X \\ Y \end{bmatrix} = \begin{bmatrix} \alpha_{11} & \alpha_{12} & \cdots & \alpha_{1k} & \beta_1 & \gamma_1 \\ \alpha_{21} & \alpha_{22} & \cdots & \alpha_{2k} & \beta_2 & \gamma_2 \end{bmatrix} \begin{bmatrix} L_1 \\ L_2 \\ \vdots \\ L_l \\ E_X \\ E_Y \end{bmatrix}, \tag{15}$$

where $\alpha_{1k}$ and $\alpha_{2k}$ are the mixing coefficients of $L_k$ on $X$ and $Y$, respectively. $\beta_1$ and $\beta_2$ are the mixing coefficients of $E_X$ on $X$ and $Y$ respectively and $\gamma_1$ and $\gamma_2$ is the mixing coefficients of $E_Y$ on $X$ and $Y$ respectively. $E_Y$ and $E_Y$ are two independent noise terms and follow non-Gaussian distributions.

1) The "if part": Follow the Definition 1, the $k_1$-th order and $k_2$-th order cumulant of $X$ and $Y$ is expressed as:

$$\begin{aligned} C_{k_1}(X) =& \alpha_{11}^{(k_1)}C_{k_1}(L_1) + \alpha_{12}^{(k_1)}C_{k_1}(L_2) + \cdots + \alpha_{1l}^{(k_1)}C_{k_1}(L_l) \\ & + \beta_1^{(k_1)}C_{k_1}(E_X) + \gamma_1^{(k_1)}C_{k_1}(E_Y), \\ C_{k_1}(Y) =& \alpha_{21}^{(k_1)}C_{k_1}(L_1) + \alpha_{22}^{(k_1)}C_{k_1}(L_2) + \cdots + \alpha_{2l}^{(k_1)}C_{k_1}(L_l) \\ & + \beta_2^{(k_1)}C_{k_1}(E_X) + \gamma_2^{(k_1)}C_{k_1}(E_Y), \\ C_{k_2}(X) =& \alpha_{11}^{(k_2)}C_{k_2}(L_1) + \alpha_{12}^{(k_2)}C_{k_2}(L_2) + \cdots + \alpha_{1l}^{(k_2)}C_{k_2}(L_l) \\ & + \beta_1^{(k_1)}C_{k_2}(E_X) + \gamma_1^{(k_1)}C_{k_2}(E_Y), \\ C_{k_2}(Y) =& \alpha_{21}^{(k_2)}C_{k_2}(L_1) + \alpha_{22}^{(k_2)}C_{k_2}(L_2) + \cdots + \alpha_{2l}^{(k_2)}C_{k_2}(L_l) \\ & + \beta_2^{(k_1)}C_{k_2}(E_X) + \gamma_2^{(k_1)}C_{k_2}(E_Y), \\ C_{k_1-i,i}(X,Y) =& \alpha_{11}^{(k_1-i)}\alpha_{21}^{(i)}C_{k_1}(L_1) + \alpha_{12}^{(k_1-i)}\alpha_{22}^{(i)}C_{k_1}(L_2) + \cdots + \alpha_{1l}^{(k_1-i)}\alpha_{2l}^{(i)}C_{k_1}(L_l) \\ & + \beta_1^{(k_1-i)}\beta_2^{(i)}C_{k_1}(E_X) + \gamma_1^{(k_1-i)}\gamma_2^{(i)}C_{k_1}(E_Y), \\ C_{k_2-j,j}(X,Y) =& \alpha_{11}^{(k_2-j)}\alpha_{21}^{(j)}C_{k_2}(L_1) + \alpha_{12}^{(k_2-j)}\alpha_{22}^{(j)}C_{k_2}(L_2) + \cdots + \alpha_{1l}^{(k_2-j)}\alpha_{2l}^{(j)}C_{k_2}(L_l) \\ & + \beta_1^{(k_2-j)}\beta_2^{(j)}C_{k_2}(E_X) + \gamma_1^{(k_2-j)}\gamma_2^{(j)}C_{k_2}(E_Y). \end{aligned} \tag{16}$$

For convenience, we let $a_{1k} = \alpha_{1k}^{(k_1)}C_{k_1}(L_k), b_1 = \beta_1^{(k_1)}C_{k_1}(E_X), c_1 = \gamma_1^{(k_1)}C_{k_1}(E_Y)$ and $a_{2k} = \alpha_{1k}^{(k_2)}C_{k_1}(L_k), b_2 = \beta_2^{(k_2)}C_{k_2}(E_X), c_2 = \gamma_2^{(k_2)}C_{k_2}(E_Y)$.

When proving the "if part", we instead prove its contrapositive statement—If there is not no directed edge between $X$ and $Y$, then $rank(J^{(k_1,k_2)}) \neq l$.

1.1) consider the direct edge, assume $X \to Y$ ($\gamma_1\gamma_2 = 0$), For any integers $k_1, k_2$ satisfying $l+2 \leq k_1, 2l+2 \leq k_2$, we have $J^{(k_1,k_2)}$ as follows :

$$
J^{(k_1,k_2)} = \left(\begin{array}{cccc}
C_{k_1-1,1} & C_{k_1-2,2} & \cdots & C_{k_1-l-1,l+1} \\
\vdots & \vdots & & \vdots \\
C_{l+1,k_1-l-1} & C_{l,k_1-l} & \cdots & C_{1,k_1-1} \\
\hline
C_{k_2-1,1} & C_{k_2-2,2} & \cdots & C_{k_2-l-2,l+2} \\
\vdots & \vdots & & \vdots \\
C_{l+1,k_2-l-1} & C_{l,k_2-l} & \cdots & C_{1,k_2-1}
\end{array}\right)
$$

$$
= \left(\begin{array}{ccc}
\sum_{i=1}^{l} \frac{\alpha_{2i}}{\alpha_{1i}}{}^{(1)}a_{1i} + \left(\frac{\beta_2}{\beta_1}\right)^1 b_1 & \cdots & \sum_{i=1}^{l}\left(\frac{\alpha_{2i}}{\alpha_{1i}}\right)^{(1+1)}a_{1i} + \left(\frac{\beta_2}{\beta_1}\right)^{l+1} b_1 \\
\vdots & \cdots & \vdots \\
\sum_{i=1}^{l} \frac{\alpha_{2i}}{\alpha_{1i}}{}^{(k_1-l-1)}a_{1i} + \left(\frac{\beta_2}{\beta_1}\right)^{(l+1)} b_1 & \cdots & \sum_{i=1}^{l}\left(\frac{\alpha_{21}}{\alpha_{11}}\right)^{(k_1-1)}a_{1i} + \left(\frac{\beta_2}{\beta_1}\right)^{k_1-1} b_1 \\
\hline
\sum_{i=1}^{l} \frac{\alpha_{2i}}{\alpha_{1i}}{}^{(1)}a_{2i} + \left(\frac{\beta_2}{\beta_1}\right)^1 b_2 & \cdots & \sum_{i=1}^{l}\left(\frac{\alpha_{21}}{\alpha_{11}}\right)^{(1+1)}a_{2i} + \left(\frac{\beta_2}{\beta_1}\right)^{l+1} b_2 \\
\vdots & \cdots & \vdots \\
\sum_{i=1}^{l} \frac{\alpha_{2i}}{\alpha_{1i}}{}^{(k_2-l-1)}a_{2i} + \left(\frac{\beta_2}{\beta_1}\right)^{(l+1)} b_2 & \cdots & \sum_{i=1}^{l}\left(\frac{\alpha_{2i}}{\alpha_{1i}}\right)^{(k_2-1)}a_{2i} + \left(\frac{\beta_2}{\beta_1}\right)^{k_2-1} b_2
\end{array}\right)
$$

$$
= C^{(k_1,k_2)} F_l, \tag{17}
$$

where

$$
C^{(k_1,k_2)} = \left(\begin{array}{ccccc}
\left(\frac{\alpha_{21}}{\alpha_{11}}\right)^1 a_{11} & \left(\frac{\alpha_{22}}{\alpha_{12}}\right)^1 a_{12} & \cdots & \left(\frac{\alpha_{2l}}{\alpha_{1l}}\right)^l a_{1l} & \left(\frac{\beta_2}{\beta_1}\right)^1 b_1 \\
\left(\frac{\alpha_{21}}{\alpha_{11}}\right)^2 a_{11} & \left(\frac{\alpha_{22}}{\alpha_{12}}\right)^2 a_{12} & \cdots & \left(\frac{\alpha_{2l}}{\alpha_{1l}}\right)^2 a_{1l} & \left(\frac{\beta_2}{\beta_1}\right)^2 b_1 \\
\vdots & \vdots & \ddots & \vdots & \vdots \\
\left(\frac{\alpha_{21}}{\alpha_{11}}\right)^{k_1-l-1} a_{11} & \left(\frac{\alpha_{22}}{\alpha_{12}}\right)^{k_1-l-1} a_{12} & \cdots & \left(\frac{\alpha_{2l}}{\alpha_{1l}}\right)^v a_{1l} & \left(\frac{\beta_2}{\beta_1}\right)^{k_1-l-1} b_1 \\
\hline
\left(\frac{\alpha_{21}}{\alpha_{11}}\right)^1 a_{21} & \left(\frac{\alpha_{22}}{\alpha_{12}}\right)^1 a_{22} & \cdots & \left(\frac{\alpha_{2l}}{\alpha_{1l}}\right)^1 a_{2l} & \left(\frac{\beta_2}{\beta_1}\right)^1 b_2 \\
\left(\frac{\alpha_{21}}{\alpha_{11}}\right)^2 a_{21} & \left(\frac{\alpha_{22}}{\alpha_{12}}\right)^2 a_{22} & \cdots & \left(\frac{\alpha_{2l}}{\alpha_{1l}}\right)^2 a_{2l} & \left(\frac{\beta_2}{\beta_1}\right)^2 b_2 \\
\vdots & \vdots & \ddots & \vdots & \vdots \\
\left(\frac{\alpha_{21}}{\alpha_{11}}\right)^{k_2-l-1} a_{21} & \left(\frac{\alpha_{22}}{\alpha_{12}}\right)^{k_2-l-1} a_{22} & \cdots & \left(\frac{\alpha_{2l}}{\alpha_{1l}}\right)^{k_2-l-1} a_{2l} & \left(\frac{\beta_2}{\beta_1}\right)^{k_2-l-1} b_2,
\end{array}\right),
$$

$$
F_l = \left(\begin{array}{ccccc}
1 & \frac{\alpha_{21}}{\alpha_{11}} & \left(\frac{\alpha_{21}}{\alpha_{11}}\right)^2 & \cdots & \left(\frac{\alpha_{21}}{\alpha_{11}}\right)^l \\
1 & \frac{\alpha_{22}}{\alpha_{12}} & \left(\frac{\alpha_{22}}{\alpha_{12}}\right)^2 & \cdots & \left(\frac{\alpha_{22}}{\alpha_{12}}\right)^l \\
\vdots & \vdots & \vdots & \cdots & \vdots \\
1 & \frac{\alpha_{2l}}{\alpha_{1l}} & \left(\frac{\alpha_{2l}}{\alpha_{1l}}\right)^2 & \cdots & \left(\frac{\alpha_{2l}}{\alpha_{1l}}\right)^l \\
1 & \frac{\beta_2}{\beta_1} & \left(\frac{\beta_2}{\beta_1}\right)^2 & \cdots & \left(\frac{\beta_2}{\beta_1}\right)^l
\end{array}\right). \tag{18}
$$

$C^{(k_1,k_2)}$ also can be decomposed into the product of $R$ and $\begin{pmatrix} F_{k_1}^T N_1 \\ F_{k_1}^T N_2 \end{pmatrix}$, i.e, $C^{(k_1,k_2)} = \begin{pmatrix} F_{k_1}^T N_1 \\ F_{k_2}^T N_2 \end{pmatrix} R$,

where

$$N_1 = \begin{pmatrix} a_{11} & 0 & \cdots & 0 & 0 \\ 0 & a_{12} & \cdots & 0 & 0 \\ \vdots & \vdots & \ddots & \vdots & \vdots \\ 0 & 0 & \cdots & a_{1l} & 0 \\ 0 & 0 & \cdots & 0 & b_1 \end{pmatrix},$$

$$N_2 = \begin{pmatrix} a_{21} & 0 & \cdots & 0 & 0 \\ 0 & a_{22} & \cdots & 0 & 0 \\ \vdots & \vdots & \ddots & \vdots & \vdots \\ 0 & 0 & \cdots & a_{2l} & 0 \\ 0 & 0 & \cdots & 0 & b_2 \end{pmatrix},$$

$$F_{k_1} = \begin{pmatrix} 1 & \frac{\alpha_{21}}{\alpha_{11}} & \left(\frac{\alpha_{21}}{\alpha_{11}}\right)^2 & \cdots & \left(\frac{\alpha_{21}}{\alpha_{11}}\right)^{k_1-l-2} \\ 1 & \frac{\alpha_{22}}{\alpha_{12}} & \left(\frac{\alpha_{22}}{\alpha_{12}}\right)^2 & \cdots & \left(\frac{\alpha_{22}}{\alpha_{12}}\right)^{k_1-l-2} \\ \vdots & \vdots & \vdots & \cdots & \vdots \\ 1 & \frac{\alpha_{2l}}{\alpha_{1l}} & \left(\frac{\alpha_{2l}}{\alpha_{1l}}\right)^2 & \cdots & \left(\frac{\alpha_{2l}}{\alpha_{1l}}\right)^{k_1-l-2} \\ 1 & \frac{\beta_2}{\beta_1} & \left(\frac{\beta_2}{\beta_1}\right)^2 & \cdots & \left(\frac{\beta_2}{\beta_1}\right)^{k_1-l-2)} \end{pmatrix}, \tag{19}$$

$$F_{k_2} = \begin{pmatrix} 1 & \frac{\alpha_{21}}{\alpha_{11}} & \left(\frac{\alpha_{21}}{\alpha_{11}}\right)^2 & \cdots & \left(\frac{\alpha_{21}}{\alpha_{11}}\right)^{k_2-l-2} \\ 1 & \frac{\alpha_{22}}{\alpha_{12}} & \left(\frac{\alpha_{22}}{\alpha_{12}}\right)^2 & \cdots & \left(\frac{\alpha_{22}}{\alpha_{12}}\right)^{k_2-l-2} \\ \vdots & \vdots & \vdots & \cdots & \vdots \\ 1 & \frac{\alpha_{2l}}{\alpha_{1l}} & \left(\frac{\alpha_{2l}}{\alpha_{1l}}\right)^2 & \cdots & \left(\frac{\alpha_{2l}}{\alpha_{1l}}\right)^{k_2-1-2} \\ 1 & \frac{\beta_2}{\beta_1} & \left(\frac{\beta_2}{\beta_1}\right)^2 & \cdots & \left(\frac{\beta_2}{\beta_1}\right)^{k_2-l-2)} \end{pmatrix},$$

$$R = \begin{pmatrix} \frac{\alpha_{21}}{\alpha_{11}} & 0 & \cdots & 0 & 0 \\ 0 & \frac{\alpha_{22}}{\alpha_{12}} & \cdots & 0 & 0 \\ \vdots & \vdots & \ddots & \vdots & \vdots \\ 0 & 0 & \cdots & \frac{\alpha_{2l}}{\alpha_{1l}} & 0 \\ 0 & 0 & \cdots & 0 & \frac{\beta_2}{\beta_1} \end{pmatrix}.$$

Since $F_l$ is a vandermonde matrix with bases $\{\frac{\alpha_{21}}{\alpha_{11}}, \frac{\alpha_{22}}{\alpha_{12}}, \cdots, \frac{\alpha_{2l}}{\alpha_{1l}}, \frac{\beta_2}{\beta_1}\}$, $F_l$ is invertible when the bases are not equal. Therefore, $rank(J^{(k_1,k_2)}) = rank(C^{(k_1,k_2)})$. Obviously, $R$ is invertible. So the rank of $C^{(k_1,k_2)}$ depends on $\begin{pmatrix} F_{k_1}^T N_1 \\ F_{k_2}^T N_2 \end{pmatrix}$. $\begin{pmatrix} F_{k_1}^T N_1 \\ F_{k_2}^T N_2 \end{pmatrix}$ is a $(k_1 + k_2 - 2l - 2) \times (l+1)$ matrix.

Thus, for any integers $k_1, k_2$ satisfying $l + 2 \le k_1, 2l + 2 \le k_2$, $rank\begin{pmatrix} F_{k_1}^T N_1 \\ F_{k_2}^T N_2 \end{pmatrix} \le l + 1$ and $rank(F_{k_2}^T) = l + 1$. Obviously, $N_2$ is invertible. $rank(F_{k_2}^T) = rank(F_{k_2}^T N_2) = l + 1$, because $rank(F_{k_2}^T N_2) = l + 1 \le rank\begin{pmatrix} F_{k_1}^T \\ F_{k_2}^T \end{pmatrix} \le l + 1$. Thus, $rank\begin{pmatrix} F_{k_1}^T \\ F_{k_2}^T \end{pmatrix} = l + 1 = rank(J^{(k_1,k_2)})$.

1.2) Considering $X \leftrightarrow Y$. For any integers $k_1, k_2$ satisfying $l + 2 \le k_1, 2l + 2 \le k_2$, we have $J^{(k_1,k_2)}$ as

$$J^{(k_1,k_2)} = C^{(k_1,k_2)} F_l, \tag{20}$$

where

$$
C^{(k_1,k_2)} = \left(\begin{array}{cccccc}
(\frac{\alpha_{21}}{\alpha_{11}})^1 a_{11} & (\frac{\alpha_{22}}{\alpha_{12}})^1 a_{12} & \cdots & (\frac{\alpha_{2l}}{\alpha_{1l}})^1 a_{1l} & (\frac{\beta_2}{\beta_1})^1 b_1 & (\frac{\gamma_2}{\gamma_1})^1 c_1 \\
(\frac{\alpha_{21}}{\alpha_{11}})^2 a_{11} & (\frac{\alpha_{22}}{\alpha_{12}})^2 a_{12} & \cdots & (\frac{\alpha_{2l}}{\alpha_{1l}})^2 a_{11} & (\frac{\beta_2}{\beta_1})^2 b_1 & (\frac{\gamma_2}{\gamma_1})^2 c_1 \\
\vdots & \vdots & \ddots & \vdots & \vdots & \vdots \\
(\frac{\alpha_{21}}{\alpha_{11}})^{k_1-l-1} a_{11} & (\frac{\alpha_{22}}{\alpha_{12}})^{k_1-l-1} a_{12} & \cdots & (\frac{\alpha_{2l}}{\alpha_{1l}})^{k_1-l-1} a_{1l} & (\frac{\beta_2}{\beta_1})^{k_1-l-1} b_1 & (\frac{\gamma_2}{\gamma_1})^{(k_1-l-1)} c_1 \\
\hline
(\frac{\alpha_{21}}{\alpha_{11}})^1 a_{21} & (\frac{\alpha_{22}}{\alpha_{12}})^1 a_{22} & \cdots & (\frac{\alpha_{2l}}{\alpha_{1l}})^1 a_{2l} & (\frac{\beta_2}{\beta_1})^1 b_2 & (\frac{\gamma_2}{\gamma_1})^1 c_2 \\
(\frac{\alpha_{21}}{\alpha_{11}})^2 a_{21} & (\frac{\alpha_{22}}{\alpha_{12}})^2 a_{22} & \cdots & (\frac{\alpha_{2l}}{\alpha_{1l}})^2 a_{2l} & (\frac{\beta_2}{\beta_1})^2 b_2 & (\frac{\gamma_2}{\gamma_1})^2 c_2 \\
\vdots & \vdots & \ddots & \vdots & \vdots & \vdots \\
(\frac{\alpha_{21}}{\alpha_{11}})^{k_2-l-1} a_{21} & (\frac{\alpha_{22}}{\alpha_{12}})^{k_2-l-1} a_{22} & \cdots & (\frac{\alpha_{2l}}{\alpha_{1l}})^{k_2-l-1} a_{2l} & (\frac{\beta_2}{\beta_1})^{l+1} b_2 & (\frac{\gamma_2}{\gamma_1})^{k_2-l-1} c_2
\end{array}\right),
$$

$$
F_l = \left(\begin{array}{ccccc}
1 & \frac{\alpha_{21}}{\alpha_{11}} & (\frac{\alpha_{21}}{\alpha_{11}})^2 & \cdots & (\frac{\alpha_{21}}{\alpha_{11}})^l \\
1 & \frac{\alpha_{22}}{\alpha_{12}} & (\frac{\alpha_{22}}{\alpha_{12}})^2 & \cdots & (\frac{\alpha_{22}}{\alpha_{12}})^l \\
\vdots & \vdots & \vdots & \ddots & \vdots \\
1 & \frac{\alpha_{2l}}{\alpha_{1l}} & (\frac{\alpha_{2l}}{\alpha_{1l}})^2 & \cdots & (\frac{\alpha_{2l}}{\alpha_{1l}})^l \\
1 & \frac{\beta_2}{\beta_1} & (\frac{\beta_2}{\beta_1})^2 & \cdots & (\frac{\beta_2}{\beta_1})^l \\
1 & \frac{\gamma_2}{\gamma_1} & (\frac{\gamma_2}{\gamma_1})^2 & \cdots & (\frac{\gamma_2}{\gamma_1})^l
\end{array}\right).
$$

$$(21)$$

$C^{(k_1,k_2)}$ also can be decomposed into the product of $R$ and $\begin{pmatrix} F_{k_1}^T N_1 \\ F_{k_1}^T N_2 \end{pmatrix}$, i.e, $C^{(k_1,k_2)} = \begin{pmatrix} F_{k_1}^T N_1 \\ F_{k_2}^T N_2 \end{pmatrix} R$,

where

$$
N_1 = \left(\begin{array}{cccccc}
a_{11} & 0 & \cdots & 0 & 0 & 0 \\
0 & a_{12} & \cdots & 0 & 0 & 0 \\
\vdots & \vdots & \ddots & \vdots & \vdots & \vdots \\
0 & 0 & \cdots & a_{1l} & 0 & 0 \\
0 & 0 & \cdots & 0 & b_1 & 0 \\
0 & 0 & \cdots & 0 & 0 & c_1
\end{array}\right),
$$

$$
N_2 = \left(\begin{array}{cccccc}
a_{21} & 0 & \cdots & 0 & 0 & 0 \\
0 & a_{22} & \cdots & 0 & 0 & 0 \\
\vdots & \vdots & \ddots & \vdots & \vdots & \vdots \\
0 & 0 & \cdots & a_{2l} & 0 & 0 \\
0 & 0 & \cdots & 0 & b_2 & 0 \\
0 & 0 & \cdots & 0 & 0 & c_2
\end{array}\right),
$$

$$
F_{k_1} = \left(\begin{array}{ccccc}
1 & \frac{\alpha_{21}}{\alpha_{11}} & (\frac{\alpha_{21}}{\alpha_{11}})^2 & \cdots & (\frac{\alpha_{21}}{\alpha_{11}})^{k_1-l-2} \\
1 & \frac{\alpha_{22}}{\alpha_{12}} & (\frac{\alpha_{22}}{\alpha_{12}})^2 & \cdots & (\frac{\alpha_{22}}{\alpha_{12}})^{k_1-l-2} \\
\vdots & \vdots & \vdots & \cdots & \vdots \\
1 & \frac{\alpha_{2l}}{\alpha_{1l}} & (\frac{\alpha_{2l}}{\alpha_{1l}})^2 & \cdots & (\frac{\alpha_{2l}}{\alpha_{1l}})^{k_1-l-2} \\
1 & \frac{\beta_2}{\beta_1} & (\frac{\beta_2}{\beta_1})^2 & \cdots & (\frac{\beta_2}{\beta_1})^{k_1-l-2)} \\
1 & \frac{\gamma_2}{\gamma_1} & (\frac{\gamma_2}{\gamma_1})^2 & \cdots & (\frac{\gamma_2}{\gamma_1})^{k_1-l-2)}
\end{array}\right),
$$

$$
F_{k_2} = \begin{pmatrix}
1 & \frac{\alpha_{21}}{\alpha_{11}} & \left(\frac{\alpha_{21}}{\alpha_{11}}\right)^2 & \cdots & \left(\frac{\alpha_{21}}{\alpha_{11}}\right)^{k_2-l-2} \\
1 & \frac{\alpha_{22}}{\alpha_{12}} & \left(\frac{\alpha_{22}}{\alpha_{12}}\right)^2 & \cdots & \left(\frac{\alpha_{22}}{\alpha_{12}}\right)^{k_2-l-2} \\
\vdots & \vdots & \vdots & \cdots & \vdots \\
1 & \frac{\alpha_{2l}}{\alpha_{1l}} & \left(\frac{\alpha_{2l}}{\alpha_{1l}}\right)^2 & \cdots & \left(\frac{\alpha_{2l}}{\alpha_{1l}}\right)^{k_2-1-2} \\
1 & \frac{\beta_2}{\beta_1} & \left(\frac{\beta_2}{\beta_1}\right)^2 & \cdots & \left(\frac{\beta_2}{\beta_1}\right)^{k_2-l-2)} \\
1 & \frac{\gamma_2}{\gamma_1} & \left(\frac{\gamma_2}{\gamma_1}\right)^2 & \cdots & \left(\frac{\gamma_2}{\gamma_1}\right)^{k_2-l-2)}
\end{pmatrix},
$$

$$
R = \begin{pmatrix}
\frac{\alpha_{21}}{\alpha_{11}} & 0 & \cdots & 0 & 0 & 0 \\
0 & \frac{\alpha_{22}}{\alpha_{12}} & \cdots & 0 & 0 & 0 \\
\vdots & \vdots & \ddots & \vdots & \vdots & \vdots \\
0 & 0 & \cdots & \frac{\alpha_{2l}}{\alpha_{1l}} & 0 & 0 \\
0 & 0 & \cdots & 0 & \frac{\beta_2}{\beta_1} & 0 \\
0 & 0 & \cdots & 0 & 0 & \frac{\gamma_2}{\gamma_1}
\end{pmatrix}.
$$

(22)

In such a case, $\begin{pmatrix} F_{k_1}^T N_1 \\ F_{k_2}^T N_2 \end{pmatrix}$ is a $(k_1 + k_2 - 2l - 2) \times (l+2)$ matrix. Thus, $rank\begin{pmatrix} F_{k_1}^T N_1 \\ F_{k_2}^T N_2 \end{pmatrix} \leq l+2$ and $rank(F_{k_2}^T) = min(k_2 - l - 1, l + 2)$.

Because $k_2 \geq 2l + 2$, there exist two possible cases: $k_2 - l - 1 \geq l + 2$ and $k_2 - l - 1 = l + 1$.

- If $k_2 - l - 1 \geq l + 2$, $F_{k_2}$ is a Vandermonde matrix, and its rank is $l + 2$. $R$ is invertible, so $rank(F_{k_2}^T) = rank\begin{pmatrix} F_{k_1}^T N_1 \\ F_{k_2}^T N_2 \end{pmatrix} = l + 2$. Since $rank(F_{k_2}^T) = l + 2 \leq rank(C^{(k_1,k_2)}) \leq l + 2$, $rank(C^{(k_1,k_2)}) = l + 2$, it implies that $C^{(k_1,k_2)}$ is of full column rank. Thus, $rank(J^{(k_1,k_2)}) = rank(F_l) = l + 1 > l$.

- If $k_2 - l - 1 = l + 1$, we consider a submatrix $U$ of $C^{(k_1,k_2)}$ as follows:

$$
U = \begin{pmatrix}
\left(\frac{\alpha_{21}}{\alpha_{11}}\right)^1 a_{11} & \left(\frac{\alpha_{22}}{\alpha_{12}}\right)^1 a_{12} & \cdots & \left(\frac{\alpha_{2l}}{\alpha_{1l}}\right)^1 a_{1l} & \left(\frac{\beta_2}{\beta_1}\right)^1 b_1 & \left(\frac{\gamma_2}{\gamma_1}\right)^1 c_1 \\
\left(\frac{\alpha_{21}}{\alpha_{11}}\right)^1 a_{21} & \left(\frac{\alpha_{22}}{\alpha_{12}}\right)^1 a_{22} & \cdots & \left(\frac{\alpha_{2l}}{\alpha_{1l}}\right)^1 a_{2l} & \left(\frac{\beta_2}{\beta_1}\right)^1 b_2 & \left(\frac{\gamma_2}{\gamma_1}\right)^1 c_2 \\
\left(\frac{\alpha_{21}}{\alpha_{11}}\right)^2 a_{21} & \left(\frac{\alpha_{22}}{\alpha_{12}}\right)^2 a_{22} & \cdots & \left(\frac{\alpha_{2l}}{\alpha_{1l}}\right)^2 a_{2l} & \left(\frac{\beta_2}{\beta_1}\right)^2 b_2 & \left(\frac{\gamma_2}{\gamma_1}\right)^2 c_2 \\
\vdots & \vdots & \ddots & \vdots & \vdots & \vdots \\
\left(\frac{\alpha_{21}}{\alpha_{11}}\right)^{l+1} a_{21} & \left(\frac{\alpha_{22}}{\alpha_{12}}\right)^{l+1} a_{22} & \cdots & \left(\frac{\alpha_{2l}}{\alpha_{1l}}\right)^{l+1} a_{2l} & \left(\frac{\beta_2}{\beta_1}\right)^{l+1} b_2 & \left(\frac{\gamma_2}{\gamma_1}\right)^{l+1} c_2
\end{pmatrix}. \quad (23)
$$

That is, $U$ is composed of the first row of $F_{k_1}^T$ and the matrix $F_{k_2}^T$. If there exists a set of coefficients $\{c_1, c_2, \cdots, c_{l+2}\}$ that satisfy $c_1 u_1 + c_2 u_2 + \cdots + c_{l+2} u_{l+2} = 0$, then we have

$$
\begin{cases}
w_1 u_{1,1} + w_2 u_{2,1} + \cdots + w_{l+2} u_{1,l+2} = w_1 u_{1,1} + \left(\sum_{j=0}^{l}(\frac{\alpha_{21}}{\alpha_{11}})^j w_{j+2,1}\right) u_{2,1} = 0, \\
w_1 u_{1,2} + w_2 u_{2,2} + \cdots + w_{l+2} u_{2,2} = w_1 u_{1,2} + \left(\sum_{j=0}^{l}(\frac{\alpha_{22}}{\alpha_{12}})^j w_{j+2,2}\right) u_{2,2} = 0, \\
\vdots \\
w_1 u_{l+2,1} + w_2 u_{2,l+2} + \cdots + w_{l+2} u_{l+2,l+2} = w_1 u_{1,l} + \left(\sum_{j=0}^{l}(\frac{\alpha_{2l}}{\alpha_{1l}})^j w_{j+2,l+2}\right) u_{2,l+2} = 0.
\end{cases} \quad (24)
$$

Obviously, the first column $u_1$ and the second column $u_2$ of $U$ are linearly independent. Thus, $w_1 = 0$, then Eq. (24) degenerates into

$$
\begin{cases}
w_2 u_{2,1} + \cdots + w_{l+2} u_{1,l+2} = 0, \\
w_2 u_{2,2} + \cdots + w_{l+2} u_{2,l+2} = 0, \\
\vdots \\
w_2 u_{2,l+2} + \cdots + w_{l+2} u_{l+2,l+2} = 0.
\end{cases} \quad (25)
$$

Since $rank(F_{k_2}^T N_2) = l + 1$, $u_2, u_3, \cdots, u_{l+2}$ are linearly independent. Therefore, $w_2 = w_3 = \cdots = w_{l+2} = w_1 = 0$, which means $rank(U) = l + 2$. Because $rank(U) = l + 2 \leq C^{(k_1,k_2)} \leq l + 2$, the $C^{(k_1,k_2)}$ has full column rank. Thus, $rank(J^{(k_1,k_2)}) = rank(F_l) = l + 1$.

In conclusion, $rank(J^{(k_1,k_2)}) = rank(F_l) = l + 1$ when $X$ and $Y$ form a cycle.

2) The "only if" part: if there is no directed edge between $X$ and $Y$, then $\beta_2 = \gamma_1 = 0$. The matrix $J^{(k_1,k_2)}$ can be express as:

$$J^{(k_1,k_2)} = C^{(k_1,k_2)}F$$

$$= \begin{pmatrix} (\frac{\alpha_{21}}{\alpha_{11}})^1 a_{11} & (\frac{\alpha_{22}}{\alpha_{12}})^1 a_{12} & \cdots & (\frac{\alpha_{2l}}{\alpha_{1l}})^1 a_{1l} \\ (\frac{\alpha_{21}}{\alpha_{11}})^2 a_{11} & (\frac{\alpha_{22}}{\alpha_{12}})^2 a_{12} & \cdots & (\frac{\alpha_{2l}}{\alpha_{1l}})^2 a_{1l} \\ \vdots & \vdots & \ddots & \vdots \\ (\frac{\alpha_{21}}{\alpha_{11}})^{k_1-l-1} a_{11} & (\frac{\alpha_{22}}{\alpha_{12}})^{k_1-l-1} a_{12} & \cdots & (\frac{\alpha_{2l}}{\alpha_{1l}})^{k_1-l-1} a_{1l} \\ \hline (\frac{\alpha_{21}}{\alpha_{11}})^1 a_{21} & (\frac{\alpha_{22}}{\alpha_{12}})^1 a_{22} & \cdots & (\frac{\alpha_{2l}}{\alpha_{1l}})^1 a_{2l} \\ (\frac{\alpha_{21}}{\alpha_{11}})^1 a_{22} & (\frac{\alpha_{22}}{\alpha_{12}})^2 a_{22} & \cdots & (\frac{\alpha_{2l}}{\alpha_{1l}})^2 a_{2l} \\ \vdots & \vdots & \ddots & \vdots \\ (\frac{\alpha_{21}}{\alpha_{11}})^{k_2-l-1} a_{21} & (\frac{\alpha_{22}}{\alpha_{12}})^{k_2-l-1} a_{22} & \cdots & (\frac{\alpha_{2l}}{\alpha_{1l}})^{k_2-l-1} a_{2l} \end{pmatrix} \begin{pmatrix} 1 & \frac{\alpha_{21}}{\alpha_{11}} & (\frac{\alpha_{21}}{\alpha_{11}})^2 & \cdots & (\frac{\alpha_{21}}{\alpha_{11}})^l \\ 1 & \frac{\alpha_{22}}{\alpha_{12}} & (\frac{\alpha_{22}}{\alpha_{12}})^2 & \cdots & (\frac{\alpha_{22}}{\alpha_{12}})^l \\ \vdots & \vdots & \vdots & \cdots & \vdots \\ 1 & \frac{\alpha_{2l}}{\alpha_{1l}} & (\frac{\alpha_{2l}}{\alpha_{1l}})^2 & \cdots & (\frac{\alpha_{2l}}{\alpha_{1l}})^l \end{pmatrix}.$$

$$(26)$$

$C^{(k_1,k_2)}$ also can be decomposed into the product of $\begin{pmatrix} F_1^T N_1 \\ F_2^T N_2 \end{pmatrix}$ and $R$, as follows:

$$N_1 = \begin{pmatrix} a_{11} & 0 & \cdots & 0 \\ 0 & a_{12} & \cdots & 0 \\ \vdots & \vdots & \ddots & \vdots \\ 0 & 0 & \cdots & a_{1l} \end{pmatrix},$$

$$N_2 = \begin{pmatrix} a_{21} & 0 & \cdots & 0 \\ 0 & a_{22} & \cdots & 0 \\ \vdots & \vdots & \ddots & \vdots \\ 0 & 0 & \cdots & a_{2l} \end{pmatrix},$$

$$F_{k_1} = \begin{pmatrix} 1 & \frac{\alpha_{21}}{\alpha_{11}} & (\frac{\alpha_{21}}{\alpha_{11}})^2 & \cdots & (\frac{\alpha_{21}}{\alpha_{11}})^{k_1-l-2} \\ 1 & \frac{\alpha_{22}}{\alpha_{12}} & (\frac{\alpha_{22}}{\alpha_{12}})^2 & \cdots & (\frac{\alpha_{22}}{\alpha_{12}})^{k_1-l-2} \\ \vdots & \vdots & \vdots & \cdots & \vdots \\ 1 & \frac{\alpha_{2l}}{\alpha_{1l}} & (\frac{\alpha_{2l}}{\alpha_{1l}})^2 & \cdots & (\frac{\alpha_{2l}}{\alpha_{1l}})^{k_1-l-2} \end{pmatrix},$$

$$F_{k_2} = \begin{pmatrix} 1 & \frac{\alpha_{21}}{\alpha_{11}} & (\frac{\alpha_{21}}{\alpha_{11}})^2 & \cdots & (\frac{\alpha_{21}}{\alpha_{11}})^{k_2-l-2} \\ 1 & \frac{\alpha_{22}}{\alpha_{12}} & (\frac{\alpha_{22}}{\alpha_{12}})^2 & \cdots & (\frac{\alpha_{22}}{\alpha_{12}})^{k_2-l-2} \\ \vdots & \vdots & \vdots & \cdots & \vdots \\ 1 & \frac{\alpha_{2l}}{\alpha_{1l}} & (\frac{\alpha_{2l}}{\alpha_{1l}})^2 & \cdots & (\frac{\alpha_{2l}}{\alpha_{1l}})^{k_2-1-2} \end{pmatrix},$$

$$R = \begin{pmatrix} \frac{\alpha_{21}}{\alpha_{11}} & 0 & \cdots & 0 \\ 0 & \frac{\alpha_{22}}{\alpha_{12}} & \cdots & 0 \\ \vdots & \vdots & \ddots & \vdots \\ 0 & 0 & \cdots & \frac{\alpha_{2l}}{\alpha_{1l}} \end{pmatrix}. \tag{27}$$

In such a case, $\begin{pmatrix} F_{k_1}^T N_1 \\ F_{k_2}^T N_2 \end{pmatrix}$ is a $(k_1 + k_2 - 2l - 2) \times l$ matrix. Thus, $rank \begin{pmatrix} F_{k_1}^T N_1 \\ F_{k_2}^T N_2 \end{pmatrix} \leq l$ and $rank(F_{k_2}^T) = l$. Obviously, $N_2$ is invertible. Therefore, $rank(F_{k_2}^T) = rank(F_{k_2}^T N_2) = l$. Since $F_{k_2}^T N_2 = l$ has a full column rank, $rank(J^{k_1,K_2}) = rank(F_l) = l$.

Therefore, the theorem is proven.

$\square$

### A.2 PROOF OF THEOREM 2

**Theorem 2.** *Assume that two observed variables $X$ and $Y$ are generated by Eq. (1), and they share $l$ latent confounders. For any integers $k_1, k_2$ satisfying $l + 2 \leq k_1$, $2l + 2 \leq k_2$, $X$ is a cause of $Y$ if and only if $rank(M_X^{(k_1,k_2)}) = l + 1$ and $rank(M_Y^{(k_1,k_2)}) = l + 2$.*

*Proof.* Assume that two observed variables $X$ and $Y$ are generated by Eq. (1), and they share $l$ latent confounders. 1) the "if" part: we prove its contrapositive statement–if $X$ is not a cause of $Y$ or the causal relationship between $X$ and $Y$ forms a cycle, then $rank(M_X^{(k_1,k_2)}) \neq l + 1$ or $rank(M_Y^{(k_1,k_2)}) \neq l + 2$.

1.1) If $X$ is not a cause of $Y$, we first consider $Y \to X$. Then, the matrix $M_X^{(k_1,k_2)}$ can be expressed as $M_X^{(k_1,k_2)} = C^{(k_1,k_2)} F_l$, where

$$
C^{(k_1,k_2)} = \left(
\begin{array}{cccccc}
(\frac{\alpha_{21}}{\alpha_{11}})^1 a_{11} & (\frac{\alpha_{22}}{\alpha_{12}})^1 a_{12} & \cdots & (\frac{\alpha_{2l}}{\alpha_{1l}})^l a_{1l} & (\frac{\beta_2}{\beta_1})^1 b_1 & (\frac{\gamma_2}{\gamma_1})^1 c_1 \\
(\frac{\alpha_{21}}{\alpha_{11}})^2 a_{11} & (\frac{\alpha_{22}}{\alpha_{12}})^2 a_{12} & \cdots & (\frac{\alpha_{2l}}{\alpha_{1l}})^2 a_{1l} & (\frac{\beta_2}{\beta_1})^2 b_1 & (\frac{\gamma_2}{\gamma_1})^2 c_1 \\
\vdots & \vdots & \ddots & \vdots & \vdots & \\
(\frac{\alpha_{21}}{\alpha_{11}})^{k_1-l-1} a_{11} & (\frac{\alpha_{22}}{\alpha_{12}})^{k_1-l-1} a_{12} & \cdots & (\frac{\alpha_{2l}}{\alpha_{1l}})^{k_1-l-1} a_{1l} & (\frac{\beta_2}{\beta_1})^{k_1-l-1} b_1 & (\frac{\gamma_2}{\gamma_1})^{k_1-l-1} c_1 \\
(\frac{\alpha_{21}}{\alpha_{11}})^1 a_{21} & (\frac{\alpha_{22}}{\alpha_{12}})^1 a_{22} & \cdots & (\frac{\alpha_{2l}}{\alpha_{1l}})^1 a_{2l} & (\frac{\beta_2}{\beta_1})^1 b_2 & (\frac{\gamma_2}{\gamma_1})^1 c_2 \\
(\frac{\alpha_{21}}{\alpha_{11}})^2 a_{21} & (\frac{\alpha_{22}}{\alpha_{12}})^2 a_{22} & \cdots & (\frac{\alpha_{2l}}{\alpha_{1l}})^2 a_{2l} & (\frac{\beta_2}{\beta_1})^2 b_2 & (\frac{\gamma_2}{\gamma_1})^2 c_2 \\
\vdots & \vdots & \ddots & \vdots & \vdots & \\
(\frac{\alpha_{21}}{\alpha_{11}})^{k_2-l-1} a_{21} & (\frac{\alpha_{22}}{\alpha_{12}})^{k_2-l-1} a_{22} & \cdots & (\frac{\alpha_{2l}}{\alpha_{1l}})^{k_2-l-1} a_{2l} & (\frac{\beta_2}{\beta_1})^{k_2-l-1} b_2 & (\frac{\gamma_2}{\gamma_1})^{k_2-l-1} c_2
\end{array}
\right),
$$

$$
F_l = \left(
\begin{array}{cccccc}
(\frac{\alpha_{21}}{\alpha_{11}})^{-1} & 1 & \frac{\alpha_{21}}{\alpha_{11}} & (\frac{\alpha_{21}}{\alpha_{11}})^2 & \cdots & (\frac{\alpha_{21}}{\alpha_{11}})^l \\
(\frac{\alpha_{22}}{\alpha_{12}})^{-1} & 1 & \frac{\alpha_{22}}{\alpha_{12}} & (\frac{\alpha_{22}}{\alpha_{12}})^2 & \cdots & (\frac{\alpha_{22}}{\alpha_{12}})^l \\
\vdots & \vdots & \vdots & & \cdots & \vdots \\
(\frac{\alpha_{2l}}{\alpha_{1l}})^{-1} & 1 & \frac{\alpha_{2l}}{\alpha_{1l}} & (\frac{\alpha_{2l}}{\alpha_{1l}})^2 & \cdots & (\frac{\alpha_{2l}}{\alpha_{1l}})^l \\
(\frac{\beta_2}{\beta_1})^{-1} & 1 & \frac{\beta_2}{\beta_1} & (\frac{\beta_2}{\beta_1})^2 & \cdots & (\frac{\beta_2}{\beta_1})^l \\
(\frac{\gamma_2}{\gamma_1})^{-1} & 1 & \frac{\gamma_2}{\gamma_1} & (\frac{\gamma_2}{\gamma_1})^2 & \cdots & (\frac{\gamma_2}{\gamma_1})^l
\end{array}
\right).
$$

$$(28)$$

$C^{(k_1,k_2)}$ also can be decomposed into the product of $\begin{pmatrix} F_{k_1}^T N_1 \\ F_{k_2}^T N_2 \end{pmatrix}$ and $R$ as follows:

$$
N_1 = \left(
\begin{array}{cccccc}
a_{11} & 0 & \cdots & 0 & 0 & 0 \\
0 & a_{12} & \cdots & 0 & 0 & 0 \\
\vdots & \vdots & \ddots & \vdots & \vdots & \vdots \\
0 & 0 & \cdots & a_{1l} & 0 & 0 \\
0 & 0 & \cdots & 0 & b_1 & 0 \\
0 & 0 & \cdots & 0 & 0 & c_1
\end{array}
\right),
$$

$$
N_2 = \left(
\begin{array}{cccccc}
a_{21} & 0 & \cdots & 0 & 0 & 0 \\
0 & a_{22} & \cdots & 0 & 0 & 0 \\
\vdots & \vdots & \ddots & \vdots & \vdots & \vdots \\
0 & 0 & \cdots & a_{2l} & 0 & 0 \\
0 & 0 & \cdots & 0 & b_2 & 0 \\
0 & 0 & \cdots & 0 & 0 & c_2
\end{array}
\right),
$$

$$
F_{k_1} = \begin{pmatrix}
1 & \frac{\alpha_{21}}{\alpha_{11}} & (\frac{\alpha_{21}}{\alpha_{11}})^2 & \cdots & (\frac{\alpha_{21}}{\alpha_{11}})^{k_1-l-2} \\
1 & \frac{\alpha_{22}}{\alpha_{12}} & (\frac{\alpha_{22}}{\alpha_{12}})^2 & \cdots & (\frac{\alpha_{22}}{\alpha_{12}})^{k_1-l-2} \\
\vdots & \vdots & \vdots & \cdots & \vdots \\
1 & \frac{\alpha_{2l}}{\alpha_{1l}} & (\frac{\alpha_{2l}}{\alpha_{1l}})^2 & \cdots & (\frac{\alpha_{2l}}{\alpha_{1l}})^{k_1-l-2} \\
1 & \frac{\beta_2}{\beta_1} & (\frac{\beta_2}{\beta_1})^2 & \cdots & (\frac{\beta_2}{\beta_1})^{k_1-l-2} \\
1 & \frac{\gamma_2}{\gamma_1} & (\frac{\gamma_2}{\gamma_1})^2 & \cdots & (\frac{\gamma_2}{\gamma_1})^{k_1-l-2}
\end{pmatrix},
$$

$$
F_{k_2} = \begin{pmatrix}
1 & \frac{\alpha_{21}}{\alpha_{11}} & (\frac{\alpha_{21}}{\alpha_{11}})^2 & \cdots & (\frac{\alpha_{21}}{\alpha_{11}})^{k_2-l-2} \\
1 & \frac{\alpha_{22}}{\alpha_{12}} & (\frac{\alpha_{22}}{\alpha_{12}})^2 & \cdots & (\frac{\alpha_{22}}{\alpha_{12}})^{k_2-l-2} \\
\vdots & \vdots & \vdots & \cdots & \vdots \\
1 & \frac{\alpha_{2l}}{\alpha_{1l}} & (\frac{\alpha_{2l}}{\alpha_{1l}})^2 & \cdots & (\frac{\alpha_{2l}}{\alpha_{1l}})^{k_2-1-2} \\
1 & \frac{\beta_2}{\beta_1} & (\frac{\beta_2}{\beta_1})^2 & \cdots & (\frac{\beta_2}{\beta_1})^{k_2-l-2} \\
1 & \frac{\gamma_2}{\gamma_1} & (\frac{\gamma_2}{\gamma_1})^2 & \cdots & (\frac{\gamma_2}{\gamma_1})^{k_2-l-2}
\end{pmatrix}, \tag{29}
$$

$$
R = \begin{pmatrix}
\frac{\alpha_{21}}{\alpha_{11}} & 0 & \cdots & 0 & 0 & 0 \\
0 & \frac{\alpha_{22}}{\alpha_{12}} & \cdots & 0 & 0 & 0 \\
\vdots & \vdots & \ddots & \vdots & \vdots & \vdots \\
0 & 0 & \cdots & \frac{\alpha_{2l}}{\alpha_{1l}} & 0 & 0 \\
0 & 0 & \cdots & 0 & \frac{\beta_2}{\beta_1} & 0 \\
0 & 0 & \cdots & 0 & 0 & \frac{\gamma_2}{\gamma_1}
\end{pmatrix}.
$$

Since $F_l$ is a vandermonde matrix with bases $\{\frac{\alpha_{21}}{\alpha_{11}}, \cdots, \frac{\alpha_{2l}}{\alpha_{1l}}, \frac{\beta_2}{\beta_1}, \frac{\gamma_2}{\gamma_1}\}$, $F$ is invertible when the bases are not equal. Therefore, $rank(J^{(k_1,k_2)}) = rank(C^{(k_1,k_2)})$. Obviously $R$ is invertible. So the rank of $C^{(k_1,k_2)}$ depends on $\begin{pmatrix} F_{k_1}^T N_1 \\ F_{k_2}^T N_2 \end{pmatrix}$. According to the proof of Theorem 1, we can obtain $rank\begin{pmatrix} F_{k_1}^T \\ F_{k_2}^T \end{pmatrix} = l + 2$. Thus, $rank(M^{(k_1,k_2)}) = l + 2$.

Similarly, if $X \to Y$, $rank(M^{(k_1,k_2)}) = l + 2$ also holds.

1.2) Considering $Y \leftrightarrow X$, $M_X^{(k_1,k_2)}$ is same as Eq. (28), thus, $rank(M_X^{k_1,k_2}) = l + 2 > l + 1$.

2) the "only if" part: if $X$ is a cause of $Y$, according to the previous formalization, we have $M_X^{(k_1,k_2)} = C^{(k_1,k_2)} F_l$, where

$$
C^{(k_1,k_2)} = \left(
\begin{array}{cccc|c}
(\frac{\alpha_{21}}{\alpha_{11}})^1 a_{11} & (\frac{\alpha_{22}}{\alpha_{12}})^1 a_{12} & \cdots & (\frac{\alpha_{2l}}{\alpha_{1l}})^1 a_{1l} & (\frac{\beta_2}{\beta_1})^1 b_1 \\
(\frac{\alpha_{21}}{\alpha_{11}})^2 a_{11} & (\frac{\alpha_{22}}{\alpha_{12}})^2 a_{12} & \cdots & (\frac{\alpha_{2l}}{\alpha_{1l}})^2 a_{1l} & (\frac{\beta_2}{\beta_1})^2 b_1 \\
\vdots & \vdots & \ddots & \vdots & \vdots \\
(\frac{\alpha_{21}}{\alpha_{11}})^{k_1-l-1} a_{11} & (\frac{\alpha_{22}}{\alpha_{12}})^{k_1-l-1} a_{12} & \cdots & (\frac{\alpha_{2l}}{\alpha_{1l}})^{k_1-l-1} a_{1l} & (\frac{\beta_2}{\beta_1})^{k_1-l-1} b_1 \\
(\frac{\alpha_{21}}{\alpha_{11}})^1 a_{21} & (\frac{\alpha_{22}}{\alpha_{12}})^1 a_{22} & \cdots & (\frac{\alpha_{2l}}{\alpha_{1l}})^1 a_{2l} & (\frac{\beta_2}{\beta_1})^1 b_2 \\
(\frac{\alpha_{21}}{\alpha_{11}})^2 a_{21} & (\frac{\alpha_{22}}{\alpha_{12}})^2 a_{22} & \cdots & (\frac{\alpha_{2l}}{\alpha_{1l}})^2 a_{2l} & (\frac{\beta_2}{\beta_1})^2 b_2 \\
\vdots & \vdots & \ddots & \vdots & \vdots \\
(\frac{\alpha_{21}}{\alpha_{11}})^{k_2-l-1} a_{21} & (\frac{\alpha_{22}}{\alpha_{12}})^{k_2-l-1} a_{22} & \cdots & (\frac{\alpha_{2l}}{\alpha_{1l}})^{k_2-l-1} a_{2l} & (\frac{\beta_2}{\beta_1})^{k_2-l-1} b_2
\end{array}
\right), \tag{30}
$$

$$
F_l = \begin{pmatrix}
(\frac{\alpha_{21}}{\alpha_{11}})^{-1} & 1 & \frac{\alpha_{21}}{\alpha_{11}} & (\frac{\alpha_{21}}{\alpha_{11}})^2 & \cdots & (\frac{\alpha_{21}}{\alpha_{11}})^l \\
(\frac{\alpha_{22}}{\alpha_{12}})^{-1} & 1 & \frac{\alpha_{22}}{\alpha_{12}} & (\frac{\alpha_{22}}{\alpha_{12}})^2 & \cdots & (\frac{\alpha_{22}}{\alpha_{12}})^l \\
\vdots & \vdots & \vdots & \vdots & \cdots & \vdots \\
(\frac{\alpha_{2l}}{\alpha_{1l}})^{-1} & 1 & \frac{\alpha_{2l}}{\alpha_{1l}} & (\frac{\alpha_{2l}}{\alpha_{1l}})^2 & \cdots & (\frac{\alpha_{2l}}{\alpha_{1l}})^l \\
(\frac{\beta_2}{\beta_1})^{-1} & 1 & \frac{\beta_2}{\beta_1} & (\frac{\beta_2}{\beta_1})^2 & \cdots & (\frac{\beta_2}{\beta_1})^l
\end{pmatrix}.
$$

$C^{(k_1,k_2)}$ also can be decomposed into the product of $R$ and $\begin{pmatrix} F_{k_1}^T N_1 \\ F_{k_2}^T N_2 \end{pmatrix}$ as follows:

$$
N_1 = \begin{pmatrix}
a_{11} & 0 & \cdots & 0 & 0 \\
0 & a_{12} & \cdots & 0 & 0 \\
\vdots & \vdots & \ddots & \vdots & \vdots \\
0 & 0 & \cdots & a_{1l} & 0 \\
0 & 0 & \cdots & 0 & b_1
\end{pmatrix},
$$

$$
N_2 = \begin{pmatrix}
a_{21} & 0 & \cdots & 0 & 0 \\
0 & a_{22} & \cdots & 0 & 0 \\
\vdots & \vdots & \ddots & \vdots & \vdots \\
0 & 0 & \cdots & a_{2l} & 0 \\
0 & 0 & \cdots & 0 & b_2
\end{pmatrix},
$$

$$
F_{k_1} = \begin{pmatrix}
1 & \frac{\alpha_{21}}{\alpha_{11}} & (\frac{\alpha_{21}}{\alpha_{11}})^2 & \cdots & (\frac{\alpha_{21}}{\alpha_{11}})^{k_1-l-2} \\
1 & \frac{\alpha_{22}}{\alpha_{12}} & (\frac{\alpha_{22}}{\alpha_{12}})^2 & \cdots & (\frac{\alpha_{22}}{\alpha_{12}})^{k_1-l-2} \\
\vdots & \vdots & \vdots & \cdots & \vdots \\
1 & \frac{\alpha_{2l}}{\alpha_{1l}} & (\frac{\alpha_{2l}}{\alpha_{1l}})^2 & \cdots & (\frac{\alpha_{2l}}{\alpha_{1l}})^{k_1-l-2} \\
1 & \frac{\beta_2}{\beta_1} & (\frac{\beta_2}{\beta_1})^2 & \cdots & (\frac{\beta_2}{\beta_1})^{k_1-l-2}
\end{pmatrix},
$$

(31)

$$
F_{k_2} = \begin{pmatrix}
1 & \frac{\alpha_{21}}{\alpha_{11}} & (\frac{\alpha_{21}}{\alpha_{11}})^2 & \cdots & (\frac{\alpha_{21}}{\alpha_{11}})^{k_2-l-2} \\
1 & \frac{\alpha_{22}}{\alpha_{12}} & (\frac{\alpha_{22}}{\alpha_{12}})^2 & \cdots & (\frac{\alpha_{22}}{\alpha_{12}})^{k_2-l-2} \\
\vdots & \vdots & \vdots & \cdots & \vdots \\
1 & \frac{\alpha_{2l}}{\alpha_{1l}} & (\frac{\alpha_{2l}}{\alpha_{1l}})^2 & \cdots & (\frac{\alpha_{2l}}{\alpha_{1l}})^{k_2-1-2} \\
1 & \frac{\beta_2}{\beta_1} & (\frac{\beta_2}{\beta_1})^2 & \cdots & (\frac{\beta_2}{\beta_1})^{k_2-l-2}
\end{pmatrix},
$$

$$
R = \begin{pmatrix}
\frac{\alpha_{21}}{\alpha_{11}} & 0 & \cdots & 0 & 0 \\
0 & \frac{\alpha_{22}}{\alpha_{12}} & \cdots & 0 & 0 \\
\vdots & \vdots & \ddots & \vdots & \vdots \\
0 & 0 & \cdots & \frac{\alpha_{2l}}{\alpha_{1l}} & 0 \\
0 & 0 & \cdots & 0 & \frac{\beta_2}{\beta_1}
\end{pmatrix}.
$$

$\begin{pmatrix} F_{k_1}^T N_1 \\ F_{k_2}^T N_2 \end{pmatrix}$ is a $(k_1 + k_2 - 2l - 2) \times (l+1)$ matrix. Thus $rank\begin{pmatrix} F_{k_1}^T N_1 \\ F_{k_2}^T N_2 \end{pmatrix} \le l+1$ and $rank(F_{k_2}^T) = l+1$. Obviously, $N_2$ is invertible. Therefore $rank(F_{k_2}^T) = rank(F_{k_2}^T N_2) = l+1$. Since $F_{k_2}^T N_2 = l$ has a full column rank, $rank(M_X^{k_1,K_2}) = rank(F_l) = l+1$.

For the matrix $M_Y^{(k_1,k_2)} =$, we have $M_Y^{(k_1,k_2)} = C^{(k_1,k_2)} F_l$, where

$$
C^{(k_1,k_2)} = \left(
\begin{array}{ccccccc}
(\frac{\alpha_{11}}{\alpha_{21}})^1 a_{11} & (\frac{\alpha_{12}}{\alpha_{22}})^1 a_{12} & \cdots & (\frac{\alpha_{1l}}{\alpha_{2l}})^l a_{1l} & (\frac{\beta_1}{\beta_2})^1 b_1 & (\frac{\gamma_1}{\gamma_2})^1 c_1 \\
(\frac{\alpha_{11}}{\alpha_{21}})^2 a_{11} & (\frac{\alpha_{12}}{\alpha_{22}})^2 a_{12} & \cdots & (\frac{\alpha_{1l}}{\alpha_{2l}})^2 a_{1l} & (\frac{\beta_1}{\beta_2})^2 b_1 & (\frac{\gamma_1}{\gamma_2})^2 c_1 \\
\vdots & \vdots & \ddots & \vdots & \vdots \\
(\frac{\alpha_{11}}{\alpha_{21}})^{k_1-l-1} a_{11} & (\frac{\alpha_{12}}{\alpha_{22}})^{k_1-l-1} a_{12} & \cdots & (\frac{\alpha_{1l}}{\alpha_{2l}})^{k_1-l-1} a_{1l} & (\frac{\beta_1}{\beta_2})^{k_1-l-1} b_1 & (\frac{\gamma_1}{\gamma_2})^{k_1-l-1} c_1 \\
\hline
(\frac{\alpha_{11}}{\alpha_{21}})^1 a_{21} & (\frac{\alpha_{12}}{\alpha_{22}})^1 a_{22} & \cdots & (\frac{\alpha_{1l}}{\alpha_{2l}})^1 a_{2l} & (\frac{\beta_1}{\beta_2})^1 b_2 & (\frac{\gamma_1}{\gamma_2})^1 c_2 \\
(\frac{\alpha_{11}}{\alpha_{21}})^2 a_{21} & (\frac{\alpha_{12}}{\alpha_{22}})^2 a_{22} & \cdots & (\frac{\alpha_{1l}}{\alpha_{2l}})^2 a_{2l} & (\frac{\beta_1}{\beta_2})^2 b_2 & (\frac{\gamma_1}{\gamma_2})^2 c_2 \\
\vdots & \vdots & \ddots & \vdots & \vdots \\
(\frac{\alpha_{11}}{\alpha_{21}})^{k_2-l-1} a_{21} & (\frac{\alpha_{12}}{\alpha_{22}})^{k_2-l-1} a_{22} & \cdots & (\frac{\alpha_{1l}}{\alpha_{2l}})^{k_2-l-1} a_{2l} & (\frac{\beta_1}{\beta_2})^{k_2-l-1} b_2 & (\frac{\gamma_1}{\gamma_2})^{k_2-l-1} c_2
\end{array}
\right),
$$

(32)

$$F_l = \begin{pmatrix} \left(\frac{\alpha_{11}}{\alpha_{21}}\right)^{-1} & 1 & \frac{\alpha_{11}}{\alpha_{21}} & \left(\frac{\alpha_{11}}{\alpha_{21}}\right)^2 & \cdots & \left(\frac{\alpha_{11}}{\alpha_{21}}\right)^l \\ \left(\frac{\alpha_{12}}{\alpha_{22}}\right)^{-1} & 1 & \frac{\alpha_{12}}{\alpha_{22}} & \left(\frac{\alpha_{12}}{\alpha_{22}}\right)^2 & \cdots & \left(\frac{\alpha_{12}}{\alpha_{22}}\right)^l \\ \vdots & \vdots & \vdots & & \cdots & \vdots \\ \left(\frac{\alpha_{1l}}{\alpha_{2l}}\right)^{-1} & 1 & \frac{\alpha_{1l}}{\alpha_{2l}} & \left(\frac{\alpha_{1l}}{\alpha_{2l}}\right)^2 & \cdots & \left(\frac{\alpha_{1l}}{\alpha_{2l}}\right)^l \\ \left(\frac{\beta_1}{\beta_2}\right)^{-1} & 1 & \frac{\beta_1}{\beta_2} & \left(\frac{\beta_1}{\beta_2}\right)^2 & \cdots & \left(\frac{\beta_1}{\beta_2}\right)^l \\ \left(\frac{\gamma_1}{\gamma_2}\right)^{-1} & 1 & \frac{\gamma_1}{\gamma_2} & \left(\frac{\gamma_1}{\gamma_2}\right)^2 & \cdots & \left(\frac{\gamma_1}{\gamma_2}\right)^l \end{pmatrix}. \tag{33}$$

$C^{(k_1,k_2)}$ also can be decomposed into the product of $R$ and $\begin{pmatrix} F_{k_1}^T N_1 \\ F_{k_2}^T N_2 \end{pmatrix}$ as follows:

$$N_1 = \begin{pmatrix} a_{11} & 0 & \cdots & 0 & 0 & 0 \\ 0 & a_{12} & \cdots & 0 & 0 & 0 \\ \vdots & \vdots & \ddots & \vdots & \vdots & \vdots \\ 0 & 0 & \cdots & a_{1l} & 0 & 0 \\ 0 & 0 & \cdots & a_{1l} & b_1 & 0 \\ 0 & 0 & \cdots & 0 & 0 & c_1 \end{pmatrix},$$

$$N_2 = \begin{pmatrix} a_{21} & 0 & \cdots & 0 & 0 & 0 \\ 0 & a_{22} & \cdots & 0 & 0 & 0 \\ \vdots & \vdots & \ddots & \vdots & \vdots & \vdots \\ 0 & 0 & \cdots & a_{2l} & 0 & 0 \\ 0 & 0 & \cdots & 0 & b_2 & 0 \\ 0 & 0 & \cdots & 0 & 0 & c_2 \end{pmatrix},$$

$$F_{k_1} = \begin{pmatrix} 1 & \frac{\alpha_{21}}{\alpha_{11}} & \left(\frac{\alpha_{21}}{\alpha_{11}}\right)^2 & \cdots & \left(\frac{\alpha_{21}}{\alpha_{11}}\right)^{k_1-l-2} \\ 1 & \frac{\alpha_{22}}{\alpha_{12}} & \left(\frac{\alpha_{22}}{\alpha_{12}}\right)^2 & \cdots & \left(\frac{\alpha_{22}}{\alpha_{12}}\right)^{k_1-l-2} \\ \vdots & \vdots & \vdots & \cdots & \vdots \\ 1 & \frac{\alpha_{2l}}{\alpha_{1l}} & \left(\frac{\alpha_{2l}}{\alpha_{1l}}\right)^2 & \cdots & \left(\frac{\alpha_{2l}}{\alpha_{1l}}\right)^{k_1-l-2} \\ 1 & \frac{\beta_2}{\beta_1} & \left(\frac{\beta_2}{\beta_1}\right)^2 & \cdots & \left(\frac{\beta_2}{\beta_1}\right)^{k_1-l-2)} \\ 1 & \frac{\gamma_2}{\gamma_1} & \left(\frac{\gamma_2}{\gamma_1}\right)^2 & \cdots & \left(\frac{\gamma_2}{\gamma_1}\right)^{k_1-l-2} \end{pmatrix},$$

$$F_{k_2} = \begin{pmatrix} 1 & \frac{\alpha_{21}}{\alpha_{11}} & \left(\frac{\alpha_{21}}{\alpha_{11}}\right)^2 & \cdots & \left(\frac{\alpha_{21}}{\alpha_{11}}\right)^{k_2-l-2} \\ 1 & \frac{\alpha_{22}}{\alpha_{12}} & \left(\frac{\alpha_{22}}{\alpha_{12}}\right)^2 & \cdots & \left(\frac{\alpha_{22}}{\alpha_{12}}\right)^{k_2-l-2} \\ \vdots & \vdots & \vdots & \cdots & \vdots \\ 1 & \frac{\alpha_{2l}}{\alpha_{1l}} & \left(\frac{\alpha_{2l}}{\alpha_{1l}}\right)^2 & \cdots & \left(\frac{\alpha_{2l}}{\alpha_{1l}}\right)^{k_2-1-2} \\ 1 & \frac{\beta_2}{\beta_1} & \left(\frac{\beta_2}{\beta_1}\right)^2 & \cdots & \left(\frac{\beta_2}{\beta_1}\right)^{k_2-l-2)} \\ 1 & \frac{\gamma_2}{\gamma_1} & \left(\frac{\gamma_2}{\gamma_1}\right)^2 & \cdots & \left(\frac{\gamma_2}{\gamma_1}\right)^{k_2-l-2} \end{pmatrix},$$

$$R = \begin{pmatrix} \frac{\alpha_{11}}{\alpha_{21}} & 0 & \cdots & 0 & 0 & 0 \\ 0 & \frac{\alpha_{12}}{\alpha_{22}} & \cdots & 0 & 0 & 0 \\ \vdots & \vdots & \ddots & \vdots & \vdots & \vdots \\ 0 & 0 & \cdots & \frac{\alpha_{1L}}{\alpha_{2l}} & 0 & 0 \\ 0 & 0 & \cdots & 0 & \frac{\beta_1}{\beta_2} & 0 \\ 0 & 0 & \cdots & 0 & 0 & \frac{\gamma_1}{\gamma_2} \end{pmatrix}. \tag{34}$$

Since $F_l$ is a vandermonde matrix with bias $\frac{\alpha_{21}}{\alpha_{11}}, \cdots, \frac{\alpha_{2l}}{\alpha_{1l}}, \frac{\beta_2}{\beta_1}, \frac{\gamma_2}{\gamma_1}$, $F$ is invertible when the bases are not equal. Therefore, $rank(J^{(k_1,k_2)}) = rank(C^{(k_1,k_2)})$. Obviously $R$ is invertible. So the rank of $C^{(k_1,k_2)}$ depends on $\begin{pmatrix} F_{k_1}^T N_1 \\ F_{k_2}^T N_2 \end{pmatrix}$. According to the proof of Theorem 1, $rank\begin{pmatrix} F_{k_1}^T \\ F_{k_2}^T \end{pmatrix} = l + 2$ thus $rank(M_Y^{(k_1,k_2)}) = l + 2$.

Therefore, the theorem is proven. $\qquad\square$

### A.3 Proof of Theorem 3

**Theorem 3.** *Assume that two observed variables $X$ and $Y$ are generated by Eq. (1), and they share $l$ latent confounders. For any integers $k_1, k_2$ satisfying $l + 2 \le k_1, 2l + 2 \le k_2$, $X$ and $Y$ form a cycle if and only if $rank(J^{(k_1,k_2)}) = l + 1, rank(M_X^{(k_1,k_2)}) = rank(M_Y^{(k_1,k_2)}) = l + 2$.*

*Proof.* 1) The "if" part: we rove its contrapositive statement—If $X$ and $Y$ don't form a cycle, then $rank(M_X^{(k_1,k_2)}) \ne l + 2$ or $rank(M_Y^{(k_1,k_2)}) \ne l + 2$.

1.1) Considering $X \to Y$, according to the proof of Theorem 2, $M_X^{(k_1,k_2)} = l + 1 \ne l + 2$.

1.2) Considering $Y \to X$, according to the proof of Theorem 2, $M_Y^{(k_1,k_2)} = l + 1 \ne l + 2$.

1.3) Considering $X$ and $Y$ are independent, according to the proof of Theorem 1, $rank(J^{(k_1,k_2)}) = l$. Since $J_X^{(k_1,k_2)})$ has only one more column than $M_X^{(k_1,k_2)})$, thus $rank(M_X^{(k_1,k_2)}) \le rank(J_X^{(k_1,k_2)}) + 1 < l + 2$.

2) The "only if" part: according the proof of Theorem 2, $M_X^{k_1,k_2} = M_Y^{k_1,k_2} = l + 2$.

Therefore, the theorem is proven. $\square$

### A.4 Proof of Theorem 4

**Theorem 4.** *Assume that two observed variables $X$ and $Y$ are generated by Eq. (1), and they share $l$ latent confounders. Then the causal structure between observed variables can be identified.*

*Proof.* Assume that two observed variables $X$ and $Y$ are generated by Eq. (1), and the number of latent confounders that $X$ and $Y$ share is $l$. Then the causal structure between $X$ and $Y$ is one of the following four kinds:

1. there is no directed edge between $X$ and $Y$;

2. $X$ is a cause of $Y$;

3. $X$ is a effect of $Y$;

4. there is a cycle between $X$ and $Y$.

The difference between the first case and the last three cases is the existence of the directed edge between $X$ and $Y$. This difference can be identified by using Theorem 1, which leverages the rank of higher-order cumulant matrices $J^{(k_1,k_2)}$. To identify the second and third cases, we can utilize Theorem 2 to achieve this. The fourth kind can be identified by Theorem 4. All of this kinds can be identified by using the rank of $J^{(k_1,k_2)}$, $M_X^{(k_1,k_2)}$, and $M_Y^{(k_1,k_2)}$, where the integers $k_1$ and $k_2$ must satisfy $k_1 \ge l + 2$ and $k_2 \ge 2l + 2$, respectively.

Therefore, the Theorem holds. $\square$

## B Experimental Results on Psychological Data

We apply our approach and baseline methods to psychological dataset (McNally et al., 2017) to assess the practical applicability of our method. The dataset consists of 408 observations and 26 variables, comprising 16 depression symptoms and 10 OCD symptoms, all with no missing values. The severity of each symptom was measured using a four-point Likert scale, with 0 indicating no symptoms and 3 indicating extreme symptoms. In this experiment, we focused on determining the causal relationships between symptoms of obsessive-compulsive disorder (OCD) and depression. Based on the findings of McNally, anhedonia (ANH), sadness (SAD), and suicide (SCD) constitute core "gateway symptoms" of depression. As essential diagnostic criteria in DSM-5, these symptoms significantly influence secondary symptoms such as fatigue, attention deficits. Therefore, we specifically investigate these three pairwise relationships.

From the results, we observed that both DirectLiNGAM (DL) (Shimizu et al., 2011) and ReLiNGAM (Re) (Schkoda et al., 2024) identified the causal relationships SAD → SCD and ANH → SCD, which aligns with the seminal finding (Beck et al., 1985; Gillissie et al., 2023). Both DL and Re identify that SAD → AND without a cycle between those two, since they assume the causal graph is acyclic. The results of CCI shows that SAD ↔ SCD, SCD↔ ANH, ANH ↔ SAD. The method proposed by Chen et al. (2024) identified that SAD, SCD and ANH are independent of each other. This might be due to the small sample size, which leads to large parameter errors in the estimation process. Our method revealed that SAD → SCD and ANH → SCD, which aligns with the seminal finding (Beck et al., 1985; Gillissie et al., 2023). Furthermore, we identified that ANH and SAD form a cycle, corroborating longitudinal findings (Lo et al., 2025; McNally et al., 2017) on the reciprocal relationship between anhedonia and sadness. These results demonstrate the validity and effectiveness of our approach.

## C  ALGORITHM 1:IDENTIFICATION OF CAUSAL RELATIONSHIP BETWEEN OBSERVED VARIABLES

---

**Algorithm 1** Identification of causal relationship between observed variables

---

1: **Input:** Data of observed variables $X, Y$.
2: **Output:** Causal relationships between $X$ and $Y$.
3: Initialize $m := 1, k_1 := 3, k_2 := 4, flag := 0$
4: Calculate $J^{(k_1,k_2)}, M_X^{(k_1,k_2)}, M_Y^{(k_1,k_2)}$
5: **while** $flag = 0$ **do**
6:    **if** $rank(J^{(k_1,k_2)}) = m$ **then**
7:      **if** $rank(J^{(k_1+1,k_2+2)}) = m$ **then**
8:        Infer $X \perp\!\!\!\perp Y \mid (L_1, \ldots, L_m)$
9:        $flag := 1$
10:      **end if**
11:    **else if** $rank(M_X^{(k_1,k_2)}) = m + 1$ and $rank(M_Y^{(k_1,k_2)}) = m + 2$ **then**
12:      Infer $X \to Y \mid (L_1, \ldots, L_m)$
13:      $flag := 1$
14:    **else if** $rank(M_X^{(k_1,k_2)}) = m + 2$ and $rank(M_Y^{(k_1,k_2)}) = m + 1$ **then**
15:      Infer $Y \to X \mid (L_1, \ldots, L_m)$
16:      $flag := 1$
17:    **else if** $rank(M_X^{(k_1,k_2)}) = rank(M_Y^{(k_1,k_2)}) = m + 2$ **then**
18:      **if** $rank(M_X^{(k_1+1,k_2+2)}) = m + 2$ **then**
19:        Infer $X \leftrightarrow Y \mid (L_1, \ldots, L_m)$
20:        $flag := 1$
21:      **end if**
22:    **end if**
23:    $m := m + 1$
24:    $k_1 := m + 2$
25:    $k_2 = 2m + 2$
26:    Update $J^{(k_1,k_2)}, M_X^{(k_1,k_2)}, M_Y^{(k_1,k_2)}$
27: **end while**

---

## D  STRATEGY FROM LOCAL TO FULL GRAPH

Our approach builds upon the fundamental property that Theorems 1–3, though originally established for bivariate cases, maintain their validity when applied to any variable pair within a global causal graph. This enables reliable determination of causal precedence between any two variables in the system. Through systematic pairwise analysis across all variables, we can not only reconstruct the complete causal order in acyclic structures, but also identify cycles directly at the pairwise level. Specifically, when two variables are found to form a cyclic component, our method yields a distinct

output indicating their cyclic relationship. This inherent capability to detect cyclic structures during pairwise analysis represents a key advantage of our framework.

We first demonstrate that Theorems 1-3 indeed hold in the global context. We subsequently validate this theoretical framework through experimental evaluation on a causal structure comprising five observed variables and one latent confounder, as illustrated in Eq. (35).

$$
\begin{bmatrix} X_1 \\ X_2 \\ X_3 \\ X_4 \\ X_5 \end{bmatrix} = \begin{bmatrix} 0 & b_{12} & 0 & 0 & 0 \\ b_{21} & 0 & 0 & 0 & 0 \\ 0 & b_{32} & 0 & 0 & 0 \\ 0 & b_{42} & 0 & 0 & 0 \\ 0 & 0 & b_{53} & b_{45} & 0 \end{bmatrix} \begin{bmatrix} X_1 \\ X_2 \\ X_3 \\ X_4 \\ X_5 \end{bmatrix} + \begin{bmatrix} 0 \\ 0 \\ 1 \\ 0 \\ 1 \end{bmatrix} L + \begin{bmatrix} E_{X_1} \\ E_{X_2} \\ E_{X_3} \\ E_{X_4} \\ E_{X_5} \end{bmatrix} \tag{35}
$$

## D.1 THEORY GUARANTEE

First, within the framework of the OICA model where $V = AS$, we clearly classify the noise that affects any two observed variables $X_i$ and $X_j$. As shown below, all independent sources $S$ are divided into the following five mutually exclusive components:

(1) Shared sources $G_{ij}$: $G_{ij} = \{g_k | g_k \in \mathbf{S} \setminus \{e_i, e_j\}, A_{i,g_k} \neq 0, A_{j,g_k} \neq 0\}$,comprising all sources other than $e_i$ and $e_j$ that exert influence on both $X_i$ and $X_j$.

(2) Unique sources of $X_i$: $U_i = \{r_k | r_k \in \mathbf{S} \setminus \{e_i, e_j\}, A_{i,r_k} \neq 0, A_{j,r_k} = 0\}$.

(3) Unique sources of $X_j$: $U_j = \{n_k | n_k \in \mathbf{S} \setminus \{e_i, e_j\}, A_{i,n_k} = 0, A_{j,n_k} \neq 0\}$.

(4) Intrinsic noise of $X_i$: $e_i$

(5) Intrinsic noise of $X_j$: $e_j$

For the purpose of this proof, we consider the causal direction $X_i \to X_j$. Then, $X_i$ and $X_j$ are expressed as:

$$
X_i = \sum_{k=1}^{n=|G_{ij}|} A_{i,g_k} g_k + \sum_{k=1}^{|U_i|} A_{i,r_k} r_k + e_i \tag{36}
$$

$$
X_j = \sum_{k=1}^{|G_{ij}|} A_{j,g_k} g_k + \sum_{k=1}^{|U_j|} A_{j,r_k} r_k + A_{ji} e_i + e_j \tag{37}
$$

Under this formulation, the joint-cumulant is employed to capture the cumulative effect of the shared noise. Therefore, the $k$ order cross-cumulant between $X_i$ and $X_j$ is given as follows

$$
\begin{bmatrix} C_{k-1,1}(X_i, X_j) \\ C_{k-2,2}(X_i, X_j) \\ \vdots \\ C_{1,k-1}(X_i, X_j) \end{bmatrix} = \begin{bmatrix} A_{i,g_1}^{(k-1)} A_{j,g_1} & A_{i,g_2}^{(k-1)} A_{j,g_2} & \cdots & A_{i,g_n}^{(k-1)} A_{j,g_n} & A_{ji} \\ A_{i,g_1}^{(k-2)} A_{j,g_1}^2 & A_{i,g_2}^{(k-2)} A_{j,g_2}^2 & \cdots & A_{i,g_n}^{(k-2)} A_{j,g_n}^2 & A_{ji}^2 \\ \vdots & \vdots & \ddots & \vdots & \vdots \\ A_{i,g_1} A_{j,g_1}^{(k-1)} & A_{i,g_2} A_{j,g_2}^{(k-1)} & \cdots & A_{i,g_n} A_{j,g_n}^{(k-1)} & A_{ji}^{(k-1)} \end{bmatrix} \begin{bmatrix} C_k(g_1) \\ C_k(g_2) \\ \vdots \\ C_k(g_n) \\ C_k(e_i) \end{bmatrix}. \tag{38}
$$

It can be observed from Eq. 38 that the mathematical structure of the derived cross-cumulants $\{C_{k-1,1}(X_i, X_j), C_{k-2,2}(X_i, X_j), \ldots, C_{1,k-1}(X_i, X_j)\}$ between $X_i$ and $X_j$ is identical to that obtained under the classical causal model where two observed variables share $n$ latent confounders with a direct causal relationship $X_i \to X_j$.

This equivalence in the cross-cumulant patterns arises because both models account for sources of shared randomness in a similar manner. In the classical latent variable causal model, the shared stochasticity originates from both the shared latent confounders and the propagated noise from the cause variable to the effect variable, while in our OICA framework, the shared randomness is precisely categorized into the shared sources $G_{ij}$ and the propagated intrinsic noise $e_i$ from $X_i$ to $X_j$.

Although the two models differ in their interpretation and classification of noise sources, they generate mathematically equivalent cross-cumulant structures. Due to this equivalence in the cross-cumulant patterns, the core argument in Theorem 2 for identifying the causal direction remains valid. The detailed proof, which primarily relies on analyzing the linear dependencies among cross-cumulants of different orders, is provided in Appendix A.2. The same line of reasoning applies to scenarios involving independent and cyclic causal relations.

Table 4: Precision, recall and f1-scores of our Method and baseline methods.

|          | DirectLiNGAM | ReLiNGAM | CCI  | Ours     |
|----------|--------------|----------|------|----------|
| precision| 0.68         | 0.31     | 0.30 | **0.84** |
| recall   | **0.86**     | 0.39     | 0.38 | 0.76     |
| F1       | 0.76         | 0.35     | 0.34 | **0.80** |

Table 5: Accuracy of our and baseline methods in Cases 1–4 and hybrid structures.

| Case             | Sample Size | DirectLiNGAM | Chen | ReLiNGAM | CCI      | Ours     |
|------------------|-------------|--------------|------|----------|----------|----------|
|                  | 200         | 0.04         | 0.36 | 0.14     | 0.01     | **0.44** |
| Case 1           | 500         | 0.01         | 0.37 | 0.34     | 0.00     | **0.52** |
|                  | 1000        | 0.00         | 0.53 | 0.36     | 0.00     | **0.64** |
|                  | 200         | 0.45         | 0.10 | **0.56** | 0.00     | 0.49     |
| Case 2           | 500         | 0.57         | 0.09 | **0.68** | 0.00     | 0.54     |
|                  | 1000        | 0.53         | 0.16 | **0.64** | 0.00     | 0.59     |
|                  | 200         | 0.00         | 0.00 | 0.00     | **0.75** | 0.34     |
| Case 3           | 500         | 0.00         | 0.00 | 0.00     | **0.82** | 0.57     |
|                  | 1000        | 0.00         | 0.00 | 0.00     | **0.77** | 0.72     |
|                  | 200         | 0.00         | 0.00 | 0.00     | **0.72** | 0.58     |
| Case 4           | 500         | 0.00         | 0.00 | 0.00     | **0.79** | 0.72     |
|                  | 1000        | 0.00         | 0.00 | 0.00     | **0.90** | 0.84     |
|                  | 200         | 0.10         | 0.10 | 0.14     | 0.21     | **0.37** |
| hybrid structures| 5000        | 0.11         | 0.07 | 0.18     | 0.36     | **0.45** |
|                  | 1000        | 0.14         | 0.08 | 0.17     | 0.33     | **0.50** |

## D.2 EXPERIMENTAL

Second, the proposed method was evaluated under the following experimental set up: a system of five observed variables with one latent confounder, as illustrated in Eq. (35), using exponentially distributed noise. Each dataset contained 10,000 samples, and causal order determination employed a rank-test threshold of 0.01. The experiment was repeated over 20 independently generated datasets, with the average Recall scores across all trials reported in Table 4. The experimental results demonstrated that our method achieved a precision of 0.76 and a recall of 0.84. Furthermore, a key advantage of our approach is its capability to identify cycles and latent confounders, a feat that DirectLiNGAM, as a baseline method, cannot accomplish.

## E EVALUATION UNDER SMALL-SAMPLE CONDITIONS

To systematically evaluate the performance of our method under small-sample conditions, we conducted additional experiments under Cases 1–4 and hybrid structures with exponential noise, using sample sizes of 200, 500, and 1000. The complete results are summarized in Table 5. These experiments demonstrate that our method remains effective even at a sample size of 200, though performance naturally improves with larger samples.

## F EVALUATION UNDER NONLINEAR CONDITIONS

To further validate the robustness of our method in nonlinear scenarios, we conducted additional experiments with exponential-distributed noise and three types of weak nonlinearities: $e^X$, $X^3$, and $\frac{1}{X}$. The experiments were performed across sample sizes of 500, 5000, 10000, maintaining the same coefficient ranges and experimental setup as in our original hybrid structure evaluation. The results, summarized in Table 6, indicate that our method achieves better performance than the compared baselines under these weak nonlinear settings.

Table 6: Accuracy of different methods on hybrid structures with weak nonlinearities and exponential noise

| Nolinear | Sample Size | DirectLiNGAM | Chen | ReLiNGAM | CCI | Ours |
|---|---|---|---|---|---|---|
| $e^X$ | 500 | 0.12 | 0.07 | 0.14 | 0.33 | **0.59** |
| | 5000 | 0.12 | 0.05 | 0.12 | 0.41 | **0.68** |
| | 10000 | 0.12 | 0.04 | 0.11 | 0.39 | **0.65** |
| $X^3$ | 500 | 0.00 | 0.00 | 0.41 | 0.31 | **0.62** |
| | 5000 | 0.00 | 0.00 | 0.41 | 0.41 | **0.73** |
| | 10000 | 0.00 | 0.00 | 0.42 | 0.37 | **0.75** |
| $\frac{1}{X}$ | 500 | 0.23 | 0.25 | 0.18 | 0.36 | **0.49** |
| | 5000 | 0.23 | 0.23 | 0.22 | 0.41 | **0.58** |
| | 10000 | 0.23 | 0.26 | 0.23 | 0.40 | **0.65** |

# G  LLMs USAGE

In the writing of this paper, we used Large Language Models (LLM) exclusively for post-editing and polishing the manuscript's language. This assistance was limited to improving grammatical accuracy, polishing sentence structures, and ensuring a consistent narrative flow. Crucially, the model played no role in the intellectual foundation of the work; the research questions, theoretical frameworks, experimental procedures, data analysis, and conclusions are entirely the product of the authors' own work.

