# OpenReview forum: "Identification of Causal Relationships in Linear Cyclic Models with Latent Variables"
_ICLR.cc/2026/Conference — Submitted to ICLR 2026_

### Official Review · Reviewer_HNMN · 2025-10-25

**Soundness:** 3
**Presentation:** 3
**Contribution:** 3
**Rating:** 6
**Confidence:** 3

**Summary:**

This paper proposes a causal discovery method utilizing higher-order statistical features that can simultaneously identify no edge, unidirectional causality, and feedback loops between two variables using only observational data, while accounting for potential confounding factors; it outperforms multiple baselines on both simulated and real-world data.

**Strengths:**

- The research problem is important and practically significant. By uniformly handling feedback loops and latent confounding, it is applicable to complex systems in biology, economics, and other domains.
- Through constructing J(k1,k2) and M(k1,k2), the paper provides identifiability-related theory based on rank criteria, distinguishing multiple scenarios including no edge, unidirectional causality, and cycles.
- The experiments are well-replicated, covering diverse scenarios including acyclic/cyclic/mixed cases and various non-Gaussian noise distributions.

**Weaknesses:**

- Currently serves as a pairwise variable identifier; no extension strategy from local to full graph, conflict resolution mechanism, or complexity analysis is provided.
- The theorems require k to grow with l, and experiments show that large sample sizes are needed for stable high accuracy; uncertainty quantification or robustness explanations are lacking for real-world small-sample cases.
- The paper's assumptions are quite strict; analysis of robustness and sensitivity to near-Gaussian noise, nonlinearity, and near-unstable systems is absent.

**Questions:**

See Weaknesses

---

> ### Author Response · Authors · 2025-11-29
> **Response to Reviewer HNMN - Q1 (Part 1/3)**
>
> We sincerely appreciate the reviewer's constructive comments and helpful feedback. Please see below for our point-by-point response.
>
> ---
>
> **(Q1)** Currently serves as a pairwise variable identifier; no extension strategy from local to full graph, conflict resolution mechanism, or complexity analysis is provided.
>
> **A1:** Thank you for your thoughtful feedback. We acknowledge that the current work focuses on pairwise causal discovery, which serves as a foundational step for identifying causal relationships between two observed variables. The causal structure between two variables is fundamental, as higher-dimensional causal graphs can be constructed by assembling pairwise relationships. In the presence of latent variables and cycles, discovering causal relationships between two variables becomes significantly more challenging, which is why we focus on this setting in our study.
>
> - Regarding the extension to full causal graphs, **our proposed method can be naturally extended to multivariable scenarios** by iteratively applying the pairwise causal discovery approach to all variable pairs. In Q1 (Part 2/3 and 3/3), we begin by providing a theoretical analysis to demonstrate that Theorems 1-3 hold in the multivariate context. Subsequently, we validate the theoretical analysis through experimental evaluation in scenarios involving more than two variables.
>
> - Regarding the concern about "conflict resolution mechanisms", we would like to clarify that **our method inherently avoids conflicts** due to its design. Specifically, our approach explicitly considers the presence of latent variables and cycles in the causal structure. These factors, which are often overlooked in traditional methods, ensure that conflicts do not arise during the identification process.
>
> - Regarding the complexity analysis, our proposed method primarily depends on calculating the rank of the cumulant matrix. In our implementation, we employ singular value decomposition (SVD) to compute the matrix rank. For a $k$-dimensional square matrix, the time complexity of SVD is $O(k^3)$. Since $k$ corresponds to the total number of observed and latent variables (denoted as $i$), each iteration has a complexity of $O(i^3)$. Given that the number of latent variables is $|L|$, the overall computational complexity of the algorithm is $\sum_{i=3}^{|L|+2} i^3 \approx O(|L|^4)$.

---

> ### Author Response · Authors · 2025-11-29
> **Response to Reviewer HNMN - Q1 (Part 2/3)**
>
> We begin by providing a theoretical analysis to demonstrate that Theorems 1-3 hold in the multivariate context.
>
> **Theoretical Guarantee**
>
> First, within the framework of the OICA model where $V = AS$, we clearly classify the noise that affects any two observed variables $X_i$ and $X_j$. As shown below, all independent sources $S$ are divided into the following five mutually exclusive components:
>
> (1) **Shared sources $G_{ij}$**: $G_{ij} = \\{g_k|g_k \in \mathbf{S} \setminus \\{e_i, e_j\\},A_{i,g_k}\neq 0 ,A_{j,g_k}\neq 0\\}$, which comprises all sources other than $e_i$ and $e_j$ that exert influence on both $X_i$ and $X_j$.
>
> (2) **Unique sources of $X_i$**: $U_{i} = \\{r_k|r_k \in \mathbf{S} \setminus \\{e_i, e_j\\},A_{i,r_k}\neq 0 ,A_{j,r_k}= 0\\}$.
>
> (3) **Unique sources of $X_j$**: $U_{j} = \\{n_k|n_k \in \mathbf{S} \setminus \\{e_i, e_j\\},A_{i,n_k} = 0 ,A_{j,n_k} \neq 0\\}$.
>
> (4) **Intrinsic noise of $X_i$**: $e_i$
>
> (5) **Intrinsic noise of $X_j$**: $e_j$
>
> For this proof, we consider the causal direction $X_i\rightarrow X_j$. Then, $X_i$ and $X_j$ are expressed as:
> \\begin{equation}
>     X_i = \\sum_{k=1}^{n=|G_{ij}|} A_{i,g_k} g_k + \\sum_{k=1}^{|U_{i}|} A_{i,r_k} r_k + e_i
> \\end{equation}
> \\begin{equation}
>     X_j = \\sum_{k=1}^{|G_{ij}|} A_{j,g_k} g_k + \\sum_{k=1}^{|U_{j}|} A_{j,r_k} r_k + A_{ji}e_i+ e_j
> \\end{equation}
>
> Under this formulation, the joint-cumulant is employed to capture the cumulative effect of the shared noise. Therefore, the $k$ order joint-cumulant between $X_i$ and $X_j$ is given as follows:
> \\begin{aligned}
> \\begin{bmatrix}
>      C_{k-1,1}(X_i,X_j)\\\\
>      C_{k-2,2}(X_i,X_j)\\\\
>      \\vdots\\\\
>      C_{1,k-1}(X_i,X_j)
> \\end{bmatrix} &=
> \\begin{bmatrix}
>      A_{i,g_1}^{(k-1)}A_{j,g_1} & A_{i,g_2}^{(k-1)}A_{j,g_2} & \\cdots & A_{i,g_n}^{(k-1)}A_{j,g_n} & A_{ji}\\\\
>      A_{i,g_1}^{(k-2)}A_{j,g_1}^2 & A_{i,g_2}^{(k-2)}A_{j,g_2}^2 & \\cdots & A_{i,g_n}^{(k-2)}A_{j,g_n}^2 & A_{ji}^2\\\\
>      \\vdots & \\vdots & \\ddots & \\vdots & \\vdots\\\\
>      A_{i,g_1}A_{j,g_1}^{(k-1)} & A_{i,g_2}A_{j,g_2}^{(k-1)} & \\cdots & A_{i,g_n}A_{j,g_n}^{(k-1)} & A_{ji}^{(k-1)}
> \\end{bmatrix}
> \\begin{bmatrix}
>      C_{k}(g_1)\\\\
>      C_{k}(g_2)\\\\
>      \\vdots\\\\
>      C_{k}(g_n)\\\\
>      C_{k}(e_i)
> \\end{bmatrix}
> \\end{aligned}
> It can be observed that the mathematical structure of the derived cross-cumulants
> $ \\{C_{k-1,1}(X_i,X_j), C_{k-2,2}(X_i,X_j),\dots, C_{1,k-1}(X_i,X_j)\\}$ between $X_i$ and $X_j$ is identical to that obtained under the classical causal model where two observed variables share $n$ latent confounders with a direct causal relationship $X_i\rightarrow {X_j}$.
>  This equivalence in the cross-cumulant patterns arises because both models account for sources of shared randomness in a similar manner. In the classical latent variable causal model, the shared stochasticity originates from both the shared latent confounders and the propagated noise from the cause variable to the effect variable, while in our OICA framework, the shared randomness is precisely categorized into the shared sources $G_{ij}$ and the propagated intrinsic noise $e_i$ from $X_i$ to $X_j$.
> Although the two models differ in their interpretation and classification of noise sources, they generate mathematically equivalent cross-cumulant structures. Due to this equivalence in the cross-cumulant patterns, the core argument in Theorem 2 for identifying the causal direction remains valid. The detailed proof, which primarily relies on analyzing the linear dependencies among cross-cumulants of different orders, is provided in Appendix A.2. The same line of reasoning applies to scenarios involving independent and cyclic causal relations.

---

> ### Author Response · Authors · 2025-11-29
> **Response to Reviewer HNMN - Q1 (Part 3/3)**
>
> **Experimental Results**
>
> To evaluate the performance of our method in scenarios involving more than two variables, we conduct experiments under the following experimental setup: a system of five observed variables with one latent confounder, as illustrated in the following equations, using exponentially distributed noise. Each dataset contained 10,000 samples, and causal order determination employed a rank-test threshold of 0.01. The experiment was repeated over 20 independently generated datasets, with the average across all trials reported in Table 1.
>
> \begin\{split\}
> \begin\{bmatrix\}
>       X\_1\\\\
>       X\_2\\\\
>       X\_3\\\\
>       X\_4\\\\
>       X\_5
> \end\{bmatrix\} &=
> \begin\{bmatrix\}
>       0 & b\_\{12\} & 0 & 0 & 0\\\\
>       b\_\{21\} & 0 & 0 & 0 & 0\\\\
>       0 & b\_\{32\} & 0 & 0 & 0\\\\
>       0 & b\_\{42\} & 0 & 0 & 0\\\\
>       0 & 0 & b\_\{53\} & b\_\{54\} & 0
> \end\{bmatrix\}
> \begin\{bmatrix\}
>       X\_1\\\\
>       X\_2\\\\
>       X\_3\\\\
>       X\_4\\\\
>       X\_5
> \end\{bmatrix\} +
> \begin\{bmatrix\}
>       0\\\\
>       0\\\\
>       1\\\\
>       0\\\\
>       1
> \end\{bmatrix\} L +
> \begin\{bmatrix\}
>       E\_\{X\_1\}\\\\
>       E\_\{X\_2\}\\\\
>       E\_\{X\_3\}\\\\
>       E\_\{X\_4\}\\\\
>       E\_\{X\_5\}
> \end\{bmatrix\}
> \end\{split\}
>
>
>
> **Table 1.** Precision, recall and f1-scores of our Method and baseline methods.
> | | DirectLiNGAM | ReLiNGAM | CCI | Ours |
> |--------|--------------|----------|-----|------|
> | Precision | 0.68 | 0.31 | 0.30 | **0.84**|
> | Recall | **0.86** | 0.39 | 0.38| 0.76 |
> | F1 | 0.76 | 0.35 | 0.34 | **0.80** |
>
> The experimental results demonstrated that our method outperforms ReLiNGAM and CCI. Notably, a significant advantage of our approach lies in its ability to identify cycles and latent confounders—an achievement that the baseline method, DirectLiNGAM, is unable to accomplish. Consequently, while DirectLiNGAM achieves the highest recall, it exhibits lower precision compared to our method.

---

> ### Author Response · Authors · 2025-11-29
> **Response to Reviewer HNMN - Q2 & Q3**
>
> **(Q2)** The theorems require k to grow with l, and experiments show that large sample sizes are needed for stable high accuracy; uncertainty quantification or robustness explanations are lacking for real-world small-sample cases.
>
>
> **A2:** We thank the reviewer for raising this point regarding sample size requirements. To systematically investigate this issue, we have conducted additional experiments under Cases 1–4 and hybrid structures with exponential noise, using smaller sample sizes of 200, 500, and 1000. The complete results are summarized in Table 2.
>
> **Table 2:** Accuracy of our and baseline methods in Cases 1–4 and hybrid structures.
> | Case | Sample Size | DirectLiNGAM | Chen | ReLiNGAM | CCI | Ours |
> |------|-------------|--------------|------|----------|-----|------|
> | **Case 1** | 200 | 0.04 | 0.36 | 0.14 | 0.01 | **0.44** |
> |  | 500 | 0.01 | 0.37 | 0.34 | 0.00 | **0.52** |
> | | 1000 | 0.00 | 0.53 | 0.36 | 0.00 | **0.64** |
> | **Case 2** | 200 | 0.45 | 0.10 | **0.56** | 0.00 | 0.49 |
> |  | 500 | 0.57 | 0.09 | **0.68** | 0.00 | 0.54 |
> |  | 1000 | 0.53 | 0.16 | **0.64** | 0.00 | 0.59 |
> | **Case 3** | 200 | 0.00 | 0.00 | 0.00 | **0.75** | 0.34 |
> | | 500 | 0.00 | 0.00 | 0.00 | **0.82** | 0.57 |
> |  | 1000 | 0.00 | 0.00 | 0.00 | **0.77** | 0.72 |
> | **Case 4** | 200 | 0.00 | 0.00 | 0.00 | **0.72** | 0.58 |
> |  | 500 | 0.00 | 0.00 | 0.00 | **0.79** | 0.72 |
> |  | 1000 | 0.00 | 0.00 | 0.00 | **0.90** | 0.84 |
> | **Hybrid structures** | 200 | 0.10 | 0.10 | 0.14 | 0.21 | **0.37** |
> |  | 500 | 0.11 | 0.07 | 0.18 | 0.36 | **0.45** |
> |  | 1000 | 0.14 | 0.08 | 0.17 | 0.33 | **0.50** |
>
> ---
>
> **(Q3)** The paper's assumptions are quite strict; analysis of robustness and sensitivity to near-Gaussian noise, nonlinearity, and near-unstable systems is absent.
>
> **A3:** To further validate the robustness of our method when violating our assumption. We conducted additional experiments with exponentially distributed noise and three types of nonlinear functions: $e^X$, $X^3$, and $\frac{1}{X}$. The experiments were performed across sample sizes of ${500, 5000, 10000}$, maintaining the same coefficient ranges and experimental setup as in our manuscript. The results are summarized in Table 4, which indicates that our method outperforms the compared baselines under these nonlinear settings.
>
> **Table 4.** Accuracy of our and baseline methods on hybrid structures in nonlinear settings.
> | Nonlinear | Sample Size | DirectLiNGAM | Chen | ReLiNGAM | CCI | Ours |
> |-----------|-------------|--------------|------|----------|-----|------|
> | **$e^X$** | 500 | 0.12 | 0.07 | 0.14 | 0.33 | **0.59** |
> |  | 5000 | 0.12 | 0.05 | 0.12 | 0.41 | **0.68** |
> |  | 10000 | 0.12 | 0.04 | 0.11 | 0.39 | **0.65** |
> | **$X^3$** | 500 | 0.00 | 0.00 | 0.41 | 0.31 | **0.62** |
> |  | 5000 | 0.00 | 0.00 | 0.41 | 0.41 | **0.73** |
> |  | 10000 | 0.00 | 0.00 | 0.42 | 0.37 | **0.75** |
> | **$\frac{1}{X}$** | 500 | 0.23 | 0.25 | 0.18 | 0.36 | **0.49** |
> |  | 5000 | 0.23 | 0.23 | 0.22 | 0.41 | **0.58** |
> |  | 10000 | 0.23 | 0.26 | 0.23 | 0.40 | **0.65** |

---

### Official Review · Reviewer_2dUj · 2025-10-28

**Soundness:** 3
**Presentation:** 3
**Contribution:** 3
**Rating:** 8
**Confidence:** 4

**Summary:**

This paper investigates how to infer the causal relationship between two variables while accounting for potential cycles and a known number of hidden confounders, under the assumption of a linear non-Gaussian model. The main idea is to exploit patterns in higher-order cumulants to infer causal directionality. The authors begin with an intuitive explanation for the case of a single hidden confounder: if the joint cumulant is identical to that of X but not to Y then X->Y, whereas if the joint cumulant matches both X and of Y then X<->Y. Then the authors extend this to the case with l hidden confounding.

**Strengths:**

New important theoretical contribution for causal discovery.

The authors validated their results on simulated data and applied it on real data.

**Weaknesses:**

The only limitation I can identify is that the paper focuses exclusively on the bivariate case. However, this seems reasonable given the complexity of the problem: starting with a simple setting is a natural first step. Hopefully, these results will pave the way toward a multivariate extension in future work.

**Questions:**

* It is not entirely clear to me how you distinguish between the case where X causes Y and Y causes X (with a hidden confounder) and the case where neither X nor Y causes the other, but there is a single hidden confounder affecting both. Could you please provide some intuition on how these two situations are differentiated? (In other words, how do you distinguish between Figure 1a and Figure 1e?)

* Can you please describe the contribution of the paper with respect to other known causal discovery algorithm like [1] [2] in the cyclic setting ?

* The arXiv version of [3] is cited in the paper. Can you please cite the UAI version instead?

* As far as know the CCD algorithm was introduced in [4] not in [5]. The algorithm introduced in [5] is LING-D.



References:

[1] Jin, Ni, Spence, Rubin, Xu. Directed Cyclic Graphs for Simultaneous Discovery of Time-Lagged and Instantaneous Causality from Longitudinal Data Using Instrumental Variables. JMLR, 2025


[2] Mooij, Claassen. Constraint-Based Causal Discovery using Partial Ancestral Graphs in the presence of Cycles. UAI, 2020.

[3] Joris Mooij and Tom Heskes. Cyclic causal discovery from continuous equilibrium data. UAI. 2013.

[4] Richardson. A Discovery Algorithm for Directed Cyclic Graphs. UAI. 1996

[5] Lacerda, Spirtes, Ramsey, Hoyer. Discovering Cyclic Causal Models by Independent Components Analysis. UAI. 2008

---

> ### Author Response · Authors · 2025-11-29
> **Response to Reviewer 2dUj - Q1 (Part 1/3)**
>
> We sincerely appreciate the reviewer's constructive comments and insightful feedback. Please see below for our response.
>
> ---
>
> **(Q1)** The only limitation I can identify is that the paper focuses exclusively on the bivariate case. However, this seems reasonable given the complexity of the problem: starting with a simple setting is a natural first step. Hopefully, these results will pave the way toward a multivariate extension in future work.
>
> **A1:** Thank you for your insightful comment. We agree that focusing on the bivariate case is a natural and reasonable starting point, especially given the complexity of the problem. The causal structure between two observed variables forms the fundamental building block for higher-dimensional structures, as all multivariate causal structures can be regarded as being assembled from pairwise relationships. Furthermore, in the presence of latent variables and cycles, identifying the causal relationship between two observed variables becomes significantly more challenging. For these reasons, we chose to focus on the bivariate case in this work and establish a solid foundation for causal discovery in such scenarios. Importantly, **our proposed method can be directly extended to multivariate settings by analyzing pairwise relationships between variables.**
>
> Below, we begin by providing a theoretical analysis to demonstrate that Theorems 1-3 hold in the multivariate context. Subsequently, we validate the theoretical analysis through experimental evaluation in scenarios involving more than two variables.

---

> ### Author Response · Authors · 2025-11-29
> **Response to Reviewer 2dUj - Q1 (Part 2/3)**
>
> **Theoretical Guarantee**
>
> First, within the framework of the OICA model where $V = AS$, we clearly classify the noise that affects any two observed variables $X_i$ and $X_j$. As shown below, all independent sources $S$ are divided into the following five mutually exclusive components:
>
> (1) **Shared sources $G_{ij}$**: $G_{ij} = \\{g_k|g_k \in \mathbf{S} \setminus \\{e_i, e_j\\},A_{i,g_k}\neq 0 ,A_{j,g_k}\neq 0\\}$, comprising all sources other than $e_i$ and $e_j$ that exert influence on both $X_i$ and $X_j$.
>
> (2) **Unique sources of $X_i$**: $U_{i} = \\{r_k|r_k \in \mathbf{S} \setminus \\{e_i, e_j\\},A_{i,r_k}\neq 0 ,A_{j,r_k}= 0\\}$.
>
> (3) **Unique sources of $X_j$**: $U_{j} = \\{n_k|n_k \in \mathbf{S} \setminus \\{e_i, e_j\\},A_{i,n_k} = 0 ,A_{j,n_k} \neq 0\\}$.
>
> (4) **Intrinsic noise of $X_i$**: $e_i$
>
> (5) **Intrinsic noise of $X_j$**: $e_j$
>
> For this proof, we consider the causal direction $X_i\rightarrow X_j$. Then, $X_i$ and $X_j$ are expressed as:
> \\begin{equation}
>     X_i = \\sum_{k=1}^{n=|G_{ij}|} A_{i,g_k} g_k + \\sum_{k=1}^{|U_{i}|} A_{i,r_k} r_k + e_i
> \\end{equation}
> \\begin{equation}
>     X_j = \\sum_{k=1}^{|G_{ij}|} A_{j,g_k} g_k + \\sum_{k=1}^{|U_{j}|} A_{j,r_k} r_k + A_{ji}e_i+ e_j
> \\end{equation}
>
> Under this formulation, the joint-cumulant is employed to capture the cumulative effect of the shared noise. Therefore, the $k$ order joint-cumulant between $X_i$ and $X_j$ is given as follows:
> \\begin{aligned}
> \\begin{bmatrix}
>      C_{k-1,1}(X_i,X_j)\\\\
>      C_{k-2,2}(X_i,X_j)\\\\
>      \\vdots\\\\
>      C_{1,k-1}(X_i,X_j)
> \\end{bmatrix} &=
> \\begin{bmatrix}
>      A_{i,g_1}^{(k-1)}A_{j,g_1} & A_{i,g_2}^{(k-1)}A_{j,g_2} & \\cdots & A_{i,g_n}^{(k-1)}A_{j,g_n} & A_{ji}\\\\
>      A_{i,g_1}^{(k-2)}A_{j,g_1}^2 & A_{i,g_2}^{(k-2)}A_{j,g_2}^2 & \\cdots & A_{i,g_n}^{(k-2)}A_{j,g_n}^2 & A_{ji}^2\\\\
>      \\vdots & \\vdots & \\ddots & \\vdots & \\vdots\\\\
>      A_{i,g_1}A_{j,g_1}^{(k-1)} & A_{i,g_2}A_{j,g_2}^{(k-1)} & \\cdots & A_{i,g_n}A_{j,g_n}^{(k-1)} & A_{ji}^{(k-1)}
> \\end{bmatrix}
> \\begin{bmatrix}
>      C_{k}(g_1)\\\\
>      C_{k}(g_2)\\\\
>      \\vdots\\\\
>      C_{k}(g_n)\\\\
>      C_{k}(e_i)
> \\end{bmatrix}
> \\end{aligned}
> It can be observed that the mathematical structure of the derived cross-cumulants
> $ \\{C_{k-1,1}(X_i,X_j), C_{k-2,2}(X_i,X_j),\dots, C_{1,k-1}(X_i,X_j)\\}$ between $X_i$ and $X_j$ is identical to that obtained under the classical causal model where two observed variables share $n$ latent confounders with a direct causal relationship $X_i\rightarrow {X_j}$.
>  This equivalence in the cross-cumulant patterns arises because both models account for sources of shared randomness in a similar manner. In the classical latent variable causal model, the shared stochasticity originates from both the shared latent confounders and the propagated noise from the cause variable to the effect variable, while in our OICA framework, the shared randomness is precisely categorized into the shared sources $G_{ij}$ and the propagated intrinsic noise $e_i$ from $X_i$ to $X_j$.
> Although the two models differ in their interpretation and classification of noise sources, they generate mathematically equivalent cross-cumulant structures. Due to this equivalence in the cross-cumulant patterns, the core argument in Theorem 2 for identifying the causal direction remains valid. The detailed proof, which primarily relies on analyzing the linear dependencies among cross-cumulants of different orders, is provided in Appendix A.2. The same line of reasoning applies to scenarios involving independent and cyclic causal relations.

---

> ### Author Response · Authors · 2025-11-29
> **Response to Reviewer 2dUj - Q1 (Part 3/3)**
>
> **Experimental Results**
>
> To evaluate the performance of our method in scenarios involving more than two variables, we conduct experiments under the following experimental setup: a system of five observed variables with one latent confounder, as illustrated in the following equations, using exponentially distributed noise. Each dataset contained 10,000 samples, and causal order determination employed a rank-test threshold of 0.01. The experiment was repeated over 20 independently generated datasets, with the average across all trials reported in Table 1.
>
> \begin\{split\}
> \begin\{bmatrix\}
>       X\_1\\\\
>       X\_2\\\\
>       X\_3\\\\
>       X\_4\\\\
>       X\_5
> \end\{bmatrix\} &=
> \begin\{bmatrix\}
>       0 & b\_\{12\} & 0 & 0 & 0\\\\
>       b\_\{21\} & 0 & 0 & 0 & 0\\\\
>       0 & b\_\{32\} & 0 & 0 & 0\\\\
>       0 & b\_\{42\} & 0 & 0 & 0\\\\
>       0 & 0 & b\_\{53\} & b\_\{54\} & 0
> \end\{bmatrix\}
> \begin\{bmatrix\}
>       X\_1\\\\
>       X\_2\\\\
>       X\_3\\\\
>       X\_4\\\\
>       X\_5
> \end\{bmatrix\} +
> \begin\{bmatrix\}
>       0\\\\
>       0\\\\
>       1\\\\
>       0\\\\
>       1
> \end\{bmatrix\} L +
> \begin\{bmatrix\}
>       E\_\{X\_1\}\\\\
>       E\_\{X\_2\}\\\\
>       E\_\{X\_3\}\\\\
>       E\_\{X\_4\}\\\\
>       E\_\{X\_5\}
> \end\{bmatrix\}
> \end\{split\}
>
>
>
> **Table 1.** Precision, recall and f1-scores of our Method and baseline methods.
> | | DirectLiNGAM | ReLiNGAM | CCI | Ours |
> |--------|--------------|----------|-----|------|
> | Precision | 0.68 | 0.31 | 0.30 | **0.84**|
> | Recall | **0.86** | 0.39 | 0.38| 0.76 |
> | F1 | 0.76 | 0.35 | 0.34 | **0.80** |
>
> The experimental results demonstrated that our method outperforms ReLiNGAM and CCI. Notably, a significant advantage of our approach lies in its ability to identify cycles and latent confounders—an achievement that the baseline method, DirectLiNGAM, is unable to accomplish. Consequently, while DirectLiNGAM achieves the highest recall, it exhibits lower precision compared to our method.

---

> ### Author Response · Authors · 2025-11-29
> **Response to Reviewer 2dUj - Q2 (Part 1/2)**
>
> **(Q2)** It is not entirely clear to me how you distinguish between the case where X causes Y and Y causes X (with a hidden confounder) and the case where neither X nor Y causes the other, but there is a single hidden confounder affecting both. Could you please provide some intuition on how these two situations are differentiated? (In other words, how do you distinguish between Figure 1a and Figure 1e?)
>
> **A2:** To distinguish these two structures, we characterize them by their corresponding matrices  $J$ and $M$ and explain how their ranks differ. A summary of the rank comparison is presented in  Table 3.
>
> **Table 3.** Ranks of matrices $J$, $M_X$ and $M_Y$ under causal strucutres depicted in Figure 1(a) and Figure 1(e).
> |  | $rank(J)$ | $rank(M_X)$ | $rank(M_Y)$ |
> |--------|-----------|-------------|-------------|
> | Figure 1(a) | 1 | 2 | 2 |
> | Figure 1(e) | 2 | 3 | 3 |
>
> Since Figure 1(a) and Figure 1(e) only have one latent variable, $k_1 = 3$ and $k_2 = 4$.
>
> (1) For the situation depicted in Figure 1(a),
> \begin\{split\}
> J^\{(3,4)\} &= \begin\{pmatrix\}
>          C\_\{2,1\}(X,Y) & C\_\{1,2\}(X,Y) \\\\
>         C\_\{3,1\}(X,Y) & C\_\{2,2\}(X,Y) \\\\
>          C\_\{2,2\}(X,Y) & C\_\{1,3\}(X,Y)
> \end\{pmatrix\} \\\\
> &= \begin\{pmatrix\}
>          \alpha\_1^2\alpha\_2 & 0 \\\\
>          0 & \alpha\_1^3\alpha\_2 \\\\
>         0 & \alpha\_1^2\alpha\_2^2
> \end\{pmatrix\}
> \begin\{pmatrix\}
>          C\_3(L) & 0 \\\\
>          0 & C\_4(L)
> \end\{pmatrix\}
> \begin\{pmatrix\}
>          1 & \alpha\_1^\{-1\}\alpha\_2  \\\\
>         1 & \alpha\_1^\{-1\}\alpha\_2
> \end\{pmatrix\},
> \end\{split\}
>
>
> \begin\{split\}
> M\_X^\{(3,4)\} &= \begin\{pmatrix\}
>          C\_\{3\}(X) & C\_\{2,1\}(X,Y) & C\_\{1,2\}(X,Y) \\\\
>         C\_\{4\}(X) & C\_\{3,1\}(X,Y) & C\_\{2,2\}(X,Y) \\\\
>          C\_\{3,1\}(X,Y) & C\_\{2,2\}(X,Y) & C\_\{1,3\}(X,Y)
> \end\{pmatrix\} \\\\
> &= \begin\{pmatrix\}
>          \alpha\_1^3 & 0 & \beta\_1^3 & 0 \\\\
>          0 & \alpha\_1^4 & 0 & \beta\_1^4 \\\\
>         0 & \alpha\_1^3\alpha\_2 & 0 & 0
> \end\{pmatrix\}
> \begin\{pmatrix\}
>          C\_3(L) & 0 & 0 & 0 \\\\
>         0 & C\_4(L) & 0 & 0 \\\\
>         0 & 0 & C\_3(E\_X) & 0 \\\\
>         0 & 0 & 0 & C\_4(E\_X)
> \end\{pmatrix\}
> \begin\{pmatrix\}
>          1 & \alpha\_1^\{-1\}\alpha\_2 & \alpha\_1^\{-2\}\alpha\_2^2  \\\\
>         1 & \alpha\_1^\{-1\}\alpha\_2 & \alpha\_1^\{-2\}\alpha\_2^2  \\\\
>         1 & \beta\_1^\{-1\}\beta\_2 & \beta\_1^\{-2\}\beta\_2^2 \\\\
>         1 & \beta\_1^\{-1\}\beta\_2 & \beta\_1^\{-2\}\beta\_2^2
> \end\{pmatrix\},
> \end\{split\}
>
> \begin\{split\}
> M\_Y^\{(3,4)\} &= \begin\{pmatrix\}
>          C\_\{3\}(Y) & C\_\{2,1\}(Y,X) & C\_\{1,2\}(Y,X) \\\\
>         C\_\{4\}(Y) & C\_\{3,1\}(Y,X) & C\_\{2,2\}(Y,X) \\\\
>          C\_\{3,1\}(Y,X) & C\_\{2,2\}(Y,X) & C\_\{1,3\}(Y,X)
> \end\{pmatrix\} \\\\
> &= \begin\{pmatrix\}
>          \alpha\_2^3 & 0 & \gamma\_2^3 & 0 \\\\
>          0 & \alpha\_2^4 & 0 & \gamma\_2^4 \\\\
>         0 & \alpha\_2^3\alpha\_1 & 0 & 0
> \end\{pmatrix\}
> \begin\{pmatrix\}
>          C\_3(L) & 0 & 0 & 0 \\\\
>           0 & C\_4(L) & 0 & 0 \\\\
>          0 & 0 & C\_3(E\_y) & 0 \\\\
>          0 & 0 & 0 & C\_4(E\_y)
> \end\{pmatrix\}
> \begin\{pmatrix\}
>          1 & \alpha\_2^\{-1\}\alpha\_1 & \alpha\_2^\{-2\}\alpha\_1^2  \\\\
>          1 & \alpha\_2^\{-1\}\alpha\_1 & \alpha\_2^\{-2\}\alpha\_1^2  \\\\
>         1 & \gamma\_2^\{-1\}\gamma\_1 & \gamma\_2^\{-2\}\gamma\_1^2 \\\\
>         1 & \gamma\_2^\{-1\}\gamma\_1 & \gamma\_2^\{-2\}\gamma\_1^2
> \end\{pmatrix\}.
> \end\{split\}
> Obviously, in the case where there is no direct edge between the two variables in Figure 1(a), $rank(J^{(3,4)})=1$, $rank(M_X^{(3,4)})=rank(M_Y^{(3,4)})=2$.

---

> ### Author Response · Authors · 2025-11-29
> **Response to Reviewer 2dUj - Q2 (Part 2/2)**
>
> (2) For the situation depicted in Figure (e),
> \begin\{split\}
> J^\{(3,4)\} &= \begin\{pmatrix\}
>          C\_\{2,1\}(X,Y) & C\_\{1,2\}(X,Y) \\\\
>         C\_\{3,1\}(X,Y) & C\_\{2,2\}(X,Y) \\\\
>          C\_\{2,2\}(X,Y) & C\_\{1,3\}(X,Y)
> \end\{pmatrix\} \\\\
> &= \begin\{pmatrix\}
>          \alpha\_1^2\alpha\_2 & 0 & \beta\_1^2\beta\_2 & \gamma\_1^2\gamma\_2 \\\\
>          0 & \alpha\_1^3\alpha\_2 & \beta\_1^3\beta\_2 & \gamma\_1^3\gamma\_2 \\\\
>         0 & \alpha\_1^2\alpha\_2^2 & \beta\_1^2\beta\_2^2 & \gamma\_1^2\gamma\_2^2
> \end\{pmatrix\}
> \begin\{pmatrix\}
>          C\_3(L) & 0 & 0 & 0 & 0 & 0 \\\\
>          0 & C\_4(L) & 0 & 0 & 0 & 0 \\\\
>         0 & 0 & C\_3(E\_X) & 0 & 0 & 0 \\\\
>         0 & 0 & 0 & C\_4(E\_X) & 0 & 0 \\\\
>          0 & 0 & 0 & 0 & C\_3(E\_Y) & 0 \\\\
>           0 & 0 & 0 & 0 & 0 & C\_4(E\_Y)
> \end\{pmatrix\}
> \begin\{pmatrix\}
>          1 & \alpha\_1^\{-1\}\alpha\_2 \\\\
>           1 & \alpha\_1^\{-1\}\alpha\_2 \\\\
>         1 & \beta\_1^\{-1\}\beta\_2 \\\\
>         1 & \beta\_1^\{-1\}\beta\_2 \\\\
>         1 & \gamma\_1^\{-1\}\gamma\_2 \\\\
>         1 & \gamma\_1^\{-1\}\gamma\_2
> \end\{pmatrix\},
> \end\{split\}
>
> \begin\{split\}
> M\_X^\{(3,4)\} &= \begin\{pmatrix\}
>          C\_\{2,1\}(X,Y) & C\_\{1,2\}(X,Y) \\\\
>         C\_\{3,1\}(X,Y) & C\_\{2,2\}(X,Y) \\\\
>          C\_\{2,2\}(X,Y) & C\_\{1,3\}(X,Y)
> \end\{pmatrix\} \\\\
> &= \begin\{pmatrix\}
>          \alpha\_1^2\alpha\_2 & 0 & \beta\_1^2\beta\_2 & \gamma\_1^2\gamma\_2 \\\\
>          0 & \alpha\_1^3\alpha\_2 & \beta\_1^3\beta\_2 & \gamma\_1^3\gamma\_2 \\\\
>         0 & \alpha\_1^2\alpha\_2^2 & \beta\_1^2\beta\_2^2 & \gamma\_1^2\gamma\_2^2
> \end\{pmatrix\}
> \begin\{pmatrix\}
>          C\_3(L) & 0 & 0 & 0 & 0 & 0 \\\\
>          0 & C\_4(L) & 0 & 0 & 0 & 0 \\\\
>         0 & 0 & C\_3(E\_X) & 0 & 0 & 0 \\\\
>         0 & 0 & 0 & C\_4(E\_X) & 0 & 0 \\\\
>          0 & 0 & 0 & 0 & C\_3(E\_Y) & 0 \\\\
>           0 & 0 & 0 & 0 & 0 & C\_4(E\_Y)
> \end\{pmatrix\}
> \begin\{pmatrix\}
>          1 & \alpha\_1^\{-1\}\alpha\_2 & \alpha\_1^\{-2\}\alpha\_2^2 \\\\
>           1 & \alpha\_1^\{-1\}\alpha\_2 & \alpha\_1^\{-2\}\alpha\_2^2 \\\\
>         1 & \beta\_1^\{-1\}\beta\_2 & \beta\_1^\{-2\}\beta\_2^2 \\\\
>         1 & \beta\_1^\{-1\}\beta\_2 & \beta\_1^\{-2\}\beta\_2^2 \\\\
>         1 & \gamma\_1^\{-1\}\gamma\_2 & \gamma\_1^\{-2\}\gamma\_2^2 \\\\
>         1 & \gamma\_1^\{-1\}\gamma\_2 & \gamma\_1^\{-2\}\gamma\_2^2
> \end\{pmatrix\},
> \end\{split\}
>
> \begin\{split\}
> M\_Y^\{(3,4)\} &= \begin\{pmatrix\}
>          C\_\{3\}(Y) & C\_\{2,1\}(Y,X) & C\_\{1,2\}(Y,X) \\\\
>         C\_\{4\}(Y) & C\_\{3,1\}(Y,X) & C\_\{2,2\}(Y,X) \\\\
>          C\_\{3,1\}(Y,X) & C\_\{2,2\}(Y,X) & C\_\{1,3\}(Y,X)
> \end\{pmatrix\} \\\\
> &= \begin\{pmatrix\}
>          \alpha\_2^2\alpha\_1 & 0 & \beta\_2^2\beta\_1 & \gamma\_2^2\gamma\_1 \\\\
>          0 & \alpha\_2^3\alpha\_1 & \beta\_2^3\beta\_1 & \gamma\_2^3\gamma\_1 \\\\
>         0 & \alpha\_2^2\alpha\_1^2 & \beta\_2^2\beta\_1^2 & \gamma\_2^2\gamma\_1^2
> \end\{pmatrix\}
> \begin\{pmatrix\}
>          C\_3(L) & 0 & 0 & 0 & 0 & 0 \\\\
>          0 & C\_4(L) & 0 & 0 & 0 & 0 \\\\
>         0 & 0 & C\_3(E\_X) & 0 & 0 & 0 \\\\
>         0 & 0 & 0 & C\_4(E\_X) & 0 & 0 \\\\
>          0 & 0 & 0 & 0 & C\_3(E\_Y) & 0 \\\\
>           0 & 0 & 0 & 0 & 0 & C\_4(E\_Y)
> \end\{pmatrix\}
> \begin\{pmatrix\}
>          1 & \alpha\_2^\{-1\}\alpha\_1 & \alpha\_2^\{-2\}\alpha\_1^2 \\\\
>         1 & \alpha\_2^\{-1\}\alpha\_1 & \alpha\_2^\{-2\}\alpha\_1^2 \\\\
>         1 & \beta\_2^\{-1\}\beta\_1 & \beta\_2^\{-2\}\beta\_1^2 \\\\
>         1 & \beta\_2^\{-1\}\beta\_1 & \beta\_2^\{-2\}\beta\_1^2 \\\\
>         1 & \gamma\_2^\{-1\}\gamma\_1 & \beta\_2^\{-2\}\beta\_1^2 \\\\
>         1 & \gamma\_2^\{-1\}\gamma\_1 & \beta\_2^\{-2\}\beta\_1^2
> \end\{pmatrix\}.
> \end\{split\}
>
> From the above analysis, in the case where there is no direct edge between the two variables in Figure 1(a), $rank(J^\{(3,4)\})=2$, $rank(M\_X^\{(3,4)\})=rank(M\_Y^\{(3,4)\})=3$.

---

> ### Author Response · Authors · 2025-11-29
> **Response to Reviewer 2dUj - Q3, Q4 & Q5**
>
> **(Q3)** Can you please describe the contribution of the paper with respect to other known causal discovery algorithm like [1] [2] in the cyclic setting?
>
> **A3:** Thank you for your insightful comments. Compared to other works in [1-2] within the cyclic setting, our key contributions are as follows:
>
> (1) We establish a theoretical framework for identifiability, enabling the recovery of causal structures between two observed variables.
>
> (2) We propose a causal discovery method that does not rely on prior knowledge of the presence of cycles or additional proxy variables.
>
> In contrast, the causal discovery algorithm introduced in [1] requires the use of instrumental variables to identify the causal graph, while the algorithm in [2] relies on additional assumptions (e.g., the causal faithfulness assumption) and cannot determine causal relationships between two dependent observed variables.
>
> ---
>
> **(Q4)** The arXiv version of [3] is cited in the paper. Can you please cite the UAI version instead?
>
> **A4:** Thank you for pointing out this issue. We have updated the manuscript to cite the UAI version of reference [3].
>
> ---
>
>
> **(Q5)** As far as know the CCD algorithm was introduced in [4] not in [5]. The algorithm introduced in [5] is LING-D.
>
> **A5:** Thank you for pointing out this important clarification. We have corrected this mistake in the updated manuscript and updated the citations accordingly. We appreciate your careful review and helpful feedback.
>
> ---
>
> **We want to thank the reviewer again for all the valuable feedback.**

---

### Official Review · Reviewer_ZTRC · 2025-10-31

**Soundness:** 2
**Presentation:** 3
**Contribution:** 3
**Rating:** 4
**Confidence:** 5

**Summary:**

The paper proposes a novel algorithm for causal discovery in the presence of cycles and latent confounders, based on cumulant–matrix rank analysis.

It provides theoretical guarantees for identifying direct edges, their directions, and the existence of cycles, with confounding.

Empirically, the method outperforms baselines in four two-variable settings (with and without confounders), and it includes a real-world case study.

**Strengths:**

1. The paper is well written and presents both theoretical guarantees and empirical demonstrations, including a real-world case study.

2. It tackles a challenging setting for causal discovery where both latent confounders and cycles may be present.

3. Across four two-variable scenarios (with/without a confounder), the proposed algorithm achieves competitive or superior accuracy.

**Weaknesses:**

1. Although the method iteratively increases $m,k_1,k_2$, there is no theoretical guarantee under misspecified orders. Ideally, none of the three rank scenarios should trigger when the orders are wrong; if they still do, that would be problematic. The paper does not clarify whether this can occur.

1.1 Empirically, the iterative procedure appears favorable in the reported experiments because the initialization matches the ground truth; consequently, when this holds, no substantive iteration occurs. Please first verify this by reporting the order values at termination for the conducted experiments. Please then further provide evidence that the method still works when the initial values are misspecified.

2. The proposed algorithm focuses on pairwise causal relations, leaving its applicability to general causal graphs unclear. Are there fundamental challenges that prevent extending it to a unified algorithm for arbitrary graphs?

3. Following the above, both the synthetic and real case studies are limited to two-variable settings, raising concerns about scalability. Additionally, the reported experiments also restrict the number of latent variables to one.

4. It would be helpful to report computational cost and running time.

Minor comments:

1.  Some symbols in Section 2 are used without first being introduced. For example, $\alpha,\beta$,  $X_a$ and $n$. Please define them upon first use or introduce them right after the first use.

2. When stating, “Although some methods can simultaneously address cycles and latent variables, they are constraint-based and have limitations in identifying certain causal edges” (line 68), please add the corresponding citations.

**Questions:**

1. From the experimental results, the CCI baseline appears sensitive to the noise distribution, which is reasonable given its reliance on conditional independence tests. What explains the pattern where the Gamma distribution sometimes works (Cases 1–2) but fails in Cases 3–4, while the other two distributions show the opposite behavior? If this is driven by test assumptions (as noted in the paper), one would expect consistency across all cases. Could you clarify this?

2. Empirically, is $n=5000$ the lower bound for effective sample size? With only two variables, 5000 observations seem relatively large compared to what other causal discovery methods typically require.

3. It stated: “This binary accuracy measure is assessed at significance levels ranging from $\alpha = 0.2$ to $\alpha = 0.01$." What does varying the significance level mean in this context? Additionally, how were the baseline parameters selected or tuned?

---

> ### Author Response · Authors · 2025-11-29
> **Response to Reviewer ZTRC - Q1 & Q1.1**
>
> We sincerely appreciate the reviewer's constructive comments and helpful feedback. Please see below for our response.
>
> ---
>
> **(Q1&Q1.1)** Although the method iteratively increases, there is no theoretical guarantee under misspecified orders... Please then further provide evidence that the method still works when the initial values are misspecified.
>
>
> **A1:** Our algorithm is designed to operate without requiring prior knowledge of the number of latent confounders and the presence of cycles. To achieve this, it employs an iterative search strategy to simultaneously determine the number of latent confounders and uncover the causal structure. In our work, $m$ represents the hypothesized number of latent confounders.
>
> Since the true number of latent confounders is unknown, the algorithm begins exploration from $m = 1$. Specifically, the algorithm progressively increases the hypothesized value of $m$ while adjusting the cumulant orders $k_1$ and $k_2$ accordingly to systematically explore higher-order statistical characteristics in the data. When starting from $m = 1$, we set $k_1 = m + 2 = 3$ and $k_2 = 2m + 2 = 4$ to construct the moment matrices $M_X$ and $M_Y$, as well as the joint statistic matrix $J$ (Line 3).
>
> If the specified value of $m$ is smaller than the true number of latent confounders (i.e., the initial value of $m$ is misspecified), the conditions statements are violated, where the matrix ranks become $rank(J^{(k_1,k_2)})=m+1$, $rank(M_X^{(k_1,k_2)})=rank(M_Y^{(k_1,k_2)})=m+2$. As a result, the algorithm bypasses the core causal discovery routines (Lines 6-16). It proceeds to Line 17, where the matrices $M_X^{(k_1,k_2)}$ and $M_Y^{(k_1,k_2)}$ are expanded to $M_X^{(k_1+1,k_2+1)}$ and $M_Y^{(k_1+1,k_2+2)}$, increasing their effective rank to $m+2$, which fails to satisfy the rank condition in Line 18. Then this automatically triggers an update where $m$ is incremented by 1, and $k_1$ and $k_2$ are adjusted for the next round of exploration (Lines 23 - 25). This process continues until $m $matches the actual number of confounders, at which point the statistics constructed with the corresponding $k_1$ and $k_2$ satisfy the identifiability conditions, and the algorithm outputs a causal structure.
>
> Notably, our algorithm is also applicable starting from $m=0$, meaning that even in the absence of latent variables, our method remains valid. In this case, all condition sets are empty.

---

> ### Author Response · Authors · 2025-11-29
> **Response to Reviewer ZTRC - Q2 (Part 1/3)**
>
> **(Q2)** The proposed algorithm focuses on pairwise causal relations, leaving its applicability to general causal graphs unclear. Are there fundamental challenges that prevent extending it to a unified algorithm for arbitrary graphs?
>
> **A2:** The causal structure between two observed variables is the fundamental structure, and all higher-dimensional structures can be regarded as being assembled from the structures of two variables. Meanwhile, in the presence of latent variables and cycles, discovering the causal structure between two observed variables becomes significantly more challenging. Therefore, the proposed algorithm focuses on identifying the causal relationship between two variables. Based on the two-variable causal discovery method we propose, it can be extended to handle multivariable scenarios.
>
> We begin by providing a theoretical analysis to demonstrate that Theorems 1-3 indeed hold for recovering the general causal graphs. Subsequently, we validate the theoretical analysis through experimental evaluation in scenarios involving more than two variables.

---

> > ### Author Response · Authors · 2025-11-29
> > **Response to Reviewer ZTRC - Q2 (Part 2/3)**
> >
> > **Theoretical Guarantee**
> >
> > First, within the framework of the OICA model where $V = AS$, we clearly classify the noise that affects any two observed variables $X_i$ and $X_j$. As shown below, all independent sources $S$ are divided into the following five mutually exclusive components:
> >
> > (1) **Shared sources $G_{ij}$**: $G_{ij} = \\{g_k|g_k \in \mathbf{S} \setminus \\{e_i, e_j\\},A_{i,g_k}\neq 0 ,A_{j,g_k}\neq 0\\}$, comprising all sources other than $e_i$ and $e_j$ that exert influence on both $X_i$ and $X_j$.
> >
> > (2) **Unique sources of $X_i$**: $U_{i} = \\{r_k|r_k \in \mathbf{S} \setminus \\{e_i, e_j\\},A_{i,r_k}\neq 0 ,A_{j,r_k}= 0\\}$.
> >
> > (3) **Unique sources of $X_j$**: $U_{j} = \\{n_k|n_k \in \mathbf{S} \setminus \\{e_i, e_j\\},A_{i,n_k} = 0 ,A_{j,n_k} \neq 0\\}$.
> >
> > (4) **Intrinsic noise of $X_i$**: $e_i$
> >
> > (5) **Intrinsic noise of $X_j$**: $e_j$
> >
> > For this proof, we consider the causal direction $X_i\rightarrow X_j$. Then, $X_i$ and $X_j$ are expressed as:
> > \\begin{equation}
> >     X_i = \\sum_{k=1}^{n=|G_{ij}|} A_{i,g_k} g_k + \\sum_{k=1}^{|U_{i}|} A_{i,r_k} r_k + e_i
> > \\end{equation}
> > \\begin{equation}
> >     X_j = \\sum_{k=1}^{|G_{ij}|} A_{j,g_k} g_k + \\sum_{k=1}^{|U_{j}|} A_{j,r_k} r_k + A_{ji}e_i+ e_j
> > \\end{equation}
> >
> > Under this formulation, the joint-cumulant is employed to capture the cumulative effect of the shared noise. Therefore, the $k$ order joint-cumulant between $X_i$ and $X_j$ is given as follows:
> > \\begin{aligned}
> > \\begin{bmatrix}
> >      C_{k-1,1}(X_i,X_j)\\\\
> >      C_{k-2,2}(X_i,X_j)\\\\
> >      \\vdots\\\\
> >      C_{1,k-1}(X_i,X_j)
> > \\end{bmatrix} &=
> > \\begin{bmatrix}
> >      A_{i,g_1}^{(k-1)}A_{j,g_1} & A_{i,g_2}^{(k-1)}A_{j,g_2} & \\cdots & A_{i,g_n}^{(k-1)}A_{j,g_n} & A_{ji}\\\\
> >      A_{i,g_1}^{(k-2)}A_{j,g_1}^2 & A_{i,g_2}^{(k-2)}A_{j,g_2}^2 & \\cdots & A_{i,g_n}^{(k-2)}A_{j,g_n}^2 & A_{ji}^2\\\\
> >      \\vdots & \\vdots & \\ddots & \\vdots & \\vdots\\\\
> >      A_{i,g_1}A_{j,g_1}^{(k-1)} & A_{i,g_2}A_{j,g_2}^{(k-1)} & \\cdots & A_{i,g_n}A_{j,g_n}^{(k-1)} & A_{ji}^{(k-1)}
> > \\end{bmatrix}
> > \\begin{bmatrix}
> >      C_{k}(g_1)\\\\
> >      C_{k}(g_2)\\\\
> >      \\vdots\\\\
> >      C_{k}(g_n)\\\\
> >      C_{k}(e_i)
> > \\end{bmatrix}
> > \\end{aligned}
> > It can be observed that the mathematical structure of the derived cross-cumulants
> > $ \\{C_{k-1,1}(X_i,X_j), C_{k-2,2}(X_i,X_j),\dots, C_{1,k-1}(X_i,X_j)\\}$ between $X_i$ and $X_j$ is identical to that obtained under the classical causal model where two observed variables share $n$ latent confounders with a direct causal relationship $X_i\rightarrow {X_j}$.
> > This equivalence in the cross-cumulant patterns arises because both models account for sources of shared randomness in a similar manner. In the classical latent variable causal model, the shared stochasticity originates from both the shared latent confounders and the propagated noise from the cause variable to the effect variable, while in our OICA framework, the shared randomness is precisely categorized into the shared sources $G_{ij}$ and the propagated intrinsic noise $e_i$ from $X_i$ to $X_j$.
> > Although the two models differ in their interpretation and classification of noise sources, they generate mathematically equivalent cross-cumulant structures. Due to this equivalence in the cross-cumulant patterns, the core argument in Theorem 2 for identifying the causal direction remains valid. The detailed proof, which primarily relies on analyzing the linear dependencies among cross-cumulants of different orders, is provided in Appendix A.2. The same line of reasoning applies to scenarios involving independent and cyclic causal relations.

---

> ### Author Response · Authors · 2025-11-29
> **Response to Reviewer ZTRC - Q2 (Part 3/3)**
>
> **Experimental Results**
>
> To evaluate the performance of our method in scenarios involving more than two variables, we conduct experiments under the following experimental setup: a system of five observed variables with one latent confounder, as illustrated in the following equations, using exponentially distributed noise. Each dataset contained 10,000 samples, and causal order determination employed a rank-test threshold of 0.01. The experiment was repeated over 20 independently generated datasets, with the average across all trials reported in Table 1.
>
> \begin\{split\}
> \begin\{bmatrix\}
>       X\_1\\\\
>       X\_2\\\\
>       X\_3\\\\
>       X\_4\\\\
>       X\_5
> \end\{bmatrix\} &=
> \begin\{bmatrix\}
>       0 & b\_\{12\} & 0 & 0 & 0\\\\
>       b\_\{21\} & 0 & 0 & 0 & 0\\\\
>       0 & b\_\{32\} & 0 & 0 & 0\\\\
>       0 & b\_\{42\} & 0 & 0 & 0\\\\
>       0 & 0 & b\_\{53\} & b\_\{54\} & 0
> \end\{bmatrix\}
> \begin\{bmatrix\}
>       X\_1\\\\
>       X\_2\\\\
>       X\_3\\\\
>       X\_4\\\\
>       X\_5
> \end\{bmatrix\} +
> \begin\{bmatrix\}
>       0\\\\
>       0\\\\
>       1\\\\
>       0\\\\
>       1
> \end\{bmatrix\} L +
> \begin\{bmatrix\}
>       E\_\{X\_1\}\\\\
>       E\_\{X\_2\}\\\\
>       E\_\{X\_3\}\\\\
>       E\_\{X\_4\}\\\\
>       E\_\{X\_5\}
> \end\{bmatrix\}
> \end\{split\}
>
>
>
> **Table 1.** Precision, recall and f1-scores of our Method and baseline methods.
> | | DirectLiNGAM | ReLiNGAM | CCI | Ours |
> |--------|--------------|----------|-----|------|
> | Precision | 0.68 | 0.31 | 0.30 | **0.84**|
> | Recall | **0.86** | 0.39 | 0.38| 0.76 |
> | F1 | 0.76 | 0.35 | 0.34 | **0.80** |
>
> The experimental results demonstrated that our method outperforms ReLiNGAM and CCI. Notably, a significant advantage of our approach lies in its ability to identify cycles and latent confounders—an achievement that the baseline method, DirectLiNGAM, is unable to accomplish. Consequently, while DirectLiNGAM achieves the highest recall, it exhibits lower precision compared to our method.

---

> ### Author Response · Authors · 2025-11-29
> **Response to Reviewer ZTRC - Q3 & Q4**
>
> **(Q3)** Following the above, both the synthetic and real case studies are limited to two-variable settings, raising concerns about scalability. Additionally, the reported experiments also restrict the number of latent variables to one.
>
>
> **A3:** (1) Regarding the scalability, we have conducted experiments in the high-dimensional case, and the experimental result is given in Table 1. From the results, our proposed method outperforms other baseline methods.
>
> (2) Regarding the single latent variables, in experiments on high-dimensional settings, each pairwise causal discovery test is performed within a specific context where the target variable pair is influenced not only by known latent confounders but also by other observed variables. This implies that the test for any given variable pair is effectively carried out under the interference of multiple latent confounders. This phenomenon is particularly pronounced when the selected variable pair consists of two leaf nodes in the global graph, as all their shared ancestors—including potentially multiple latent variables—can confound their observed relationship.
>
> ----
>
> **(Q4)** It would be helpful to report computational cost and running time.
>
> **A4:** (1) Regarding the computational cost, it primarily depends on calculating the rank of the cumulant matrix. In our algorithm, we employ singular value decomposition (SVD) to compute the matrix rank. For a $k$-dimensional square matrix, the time complexity of SVD is $O(k^3)$. Since $k$ corresponds to the total number of observed and latent variables (denoted as $i$), each iteration has a complexity of $O(i^3)$. Given that the number of latent variables is $|L|$, the overall computational complexity of the algorithm is $\sum_{i=3}^{|L|+2} i^3 \approx O(|L|^4)$.
>
> (2) Regarding the running time, in our experiments with the five variables and one latent confounder, the average running time of our method was 0.23 seconds per pairwise analysis.

---

> ### Author Response · Authors · 2025-11-29
> **Response to Reviewer ZTRC - Q5-Q8**
>
> **(Q5)** Minor comments
>
> **A5:** Thank you for raising these points. We have added the definition of symbols in Section 2 and added the corresponding reference that you mentioned in the updated version.
>
> ---
>
> **(Q6)** From the experimental results, the CCI baseline appears sensitive to the noise distribution, which is reasonable given its reliance on conditional independence tests. What explains the pattern where the Gamma distribution sometimes works (Cases 1–2) but fails in Cases 3–4, while the other two distributions show the opposite behavior? If this is driven by test assumptions (as noted in the paper), one would expect consistency across all cases. Could you clarify this?
>
> **A6:** Thank you very much for raising this issue. In our experiments, we used the default settings of the CCI source code, where the conditional independence test is performed using the Fisher-Z test. This method is highly accurate when applied to Gaussian-distributed data. However, the Gamma-distributed data used in this study exhibits strong non-Gaussian characteristics, which led to the failure of the Fisher-Z test in this scenario.
>
> ---
>
> **(Q7)** Empirically, N is 5000 the lower bound for effective sample size? With only two variables, 5000 observations seem relatively large compared to what other causal discovery methods typically require.
>
> **A7:**
> We thank the reviewer for raising this important point regarding sample size requirements. To systematically investigate this issue, we have conducted additional experiments under Cases 1–4 and hybrid structures with exponential noise, using smaller sample sizes of 200, 500, and 1000. The experimental results are summarized in Table 2. These experimental results demonstrate that our method remains effective even at a sample size of 200, though performance naturally improves with larger samples.
>
>
> **Table 2.** Accuracy of our and baseline methods in Cases 1–4 and hybrid structures.
> |  | Sample Size | DirectLiNGAM | Chen | ReLiNGAM | CCI | Ours |
> |------|-------------|--------------|------|----------|-----|------|
> | **Case 1** | 200 | 0.04 | 0.36 | 0.14 | 0.01 | **0.44** |
> |  | 500 | 0.01 | 0.37 | 0.34 | 0.00 | **0.52** |
> | | 1000 | 0.00 | 0.53 | 0.36 | 0.00 | **0.64** |
> | **Case 2** | 200 | 0.45 | 0.10 | **0.56** | 0.00 | 0.49 |
> |  | 500 | 0.57 | 0.09 | **0.68** | 0.00 | 0.54 |
> |  | 1000 | 0.53 | 0.16 | **0.64** | 0.00 | 0.59 |
> | **Case 3** | 200 | 0.00 | 0.00 | 0.00 | **0.75** | 0.34 |
> | | 500 | 0.00 | 0.00 | 0.00 | **0.82** | 0.57 |
> |  | 1000 | 0.00 | 0.00 | 0.00 | **0.77** | 0.72 |
> | **Case 4** | 200 | 0.00 | 0.00 | 0.00 | **0.72** | 0.58 |
> |  | 500 | 0.00 | 0.00 | 0.00 | **0.79** | 0.72 |
> |  | 1000 | 0.00 | 0.00 | 0.00 | **0.90** | 0.84 |
> | **hybrid structures** | 200 | 0.10 | 0.10 | 0.14 | 0.21 | **0.37** |
> |  | 500 | 0.11 | 0.07 | 0.18 | 0.36 | **0.45** |
> |  | 1000 | 0.14 | 0.08 | 0.17 | 0.33 | **0.50** |
>
> ---
>
> **(Q8)** It stated: “This binary accuracy measure is assessed at significance levels ranging from “ 0.2 to 0.01." What does varying the significance level mean in this context? Additionally, how were the baseline parameters selected or tuned?
>
> **A8:** The accuracy of our method fundamentally depends on the accuracy of the cumulative quantity estimation. The core of our approach involves performing Singular Value Decomposition (SVD) on these matrices. The rank of a matrix is then determined by counting the number of singular values whose absolute magnitude exceeds a predefined threshold. Since the statistical variability of empirical cumulants decreases with increasing sample size, we employ a sample-size-dependent threshold function of the form $\alpha =\frac{c}{\sqrt[3]{N}}$, where the constant $c$ is typically set to 2. However, it is important to note that selecting an excessively large threshold is potentially risky, as it may lead to increased false positives in causal discovery. To ensure robustness, particularly with small sample sizes, we impose a conservative upper bound of 0.2 on the threshold value. For example, in our small-sample experiments with sample size $N = 200$, we use a threshold of 0.2 instead of the larger value suggested by the formula. This design guarantees reliable performance across different sample sizes while maintaining an appropriate balance between sensitivity and specificity.
>
> ---
>
> **We want to thank the reviewer again for all the valuable feedback.**

---

### Official Review · Reviewer_pV6B · 2025-11-02

**Soundness:** 3
**Presentation:** 2
**Contribution:** 2
**Rating:** 4
**Confidence:** 4

**Summary:**

The paper studies identifying causal relations between two observed variables under linear non-Gaussian models that may contain cycles and latent confounders. The key idea is to build two higher-order cumulant matrices, $J^{k_1,k_2}$ (from joint cumulants) and $M^{k_1,k_2}$ (augmenting with marginal cumulants), and then use their ranks to decide: (i) whether there is an edge, (ii) the edge’s direction, and (iii) whether there is a cycle. Under the standard stability condition for cyclic SEMs and non-Gaussian independent noises, the paper proves rank-based necessary and sufficient criteria (Theorems 1–3) and summarizes identifiability (Theorem 4). An algorithm then iterates over $k_1,k_2$ and an estimated number of shared latents to classify the $X,Y$ relation. Experiments on synthetic cases (no edge/ single edge/cycle, with or without a latent) and a small psychological dataset illustrate the approach, with comparisons to DirectLiNGAM, a cumulant-based acyclic method, ReLiNGAM, and CCI.

**Strengths:**

- Unified handling of cycles and latents from observational data in the bivariate case. The same rank framework covers “no edge,” “directed edge,” and “cycle”.
- The paper explains how marginal vs joint higher-order cumulants share (or don’t share) noise terms across structures, which motivates the rank criteria.

**Weaknesses:**

- All theorems target a pair of observables at a time. There is no procedure (or guarantee) for constructing a globally consistent multivariate graph from pairwise decisions.

- High-order cumulants are statistically noisy; the method needs large sample sizes before ranks stabilize. The paper gives little guidance on regularization/thresholding for near-rank decisions.

- The manuscript needs a careful language edit.

**Questions:**

- How do you set numerical thresholds for deciding ranks of $J$ and $M$ from noisy cumulant estimates? Any bootstrap, or shrinkage scheme to stabilize near-boundary cases?

- The algorithm increases $m$ and adjusts $(k_1,k_2)$ (e.g., start with $(3,4)$, then $k_1 = m+2$, $k_2 = 2m+2$). Why this schedule, and how sensitive is performance to alternate choices (e.g., using multiple $(k_1,k_2)$ and aggregating decisions)? Please clarify lines 4-26 of Algorithm 1.

- For $p>2$ observables, how do you ensure a globally consistent graph when pairwise decisions disagree?

Minor comment: Please fix typos (“identifiablity,” “acylic,” “RCREPRODUCIBILITY,” etc.)

---

> ### Author Response · Authors · 2025-11-29
> **Response to Reviewer pV6B - Q1 (Part 1/3)**
>
> We sincerely appreciate the reviewer's constructive comments and helpful feedback. Please see below for our response.
>
> ---
> **(Q1)** All theorems target a pair of observables at a time. There is no procedure (or guarantee) for constructing a globally consistent multivariate graph from pairwise decisions.
>
> **A1:** Thank you for this insightful point. Our method primarily addresses the fundamental problem of the identification of the causal structure between two observed variables in the presence of latent variables and cycles. Though all theorems originally target a pair of observables at a time, they maintain their validity when applied to any variable pair within a global causal graph. This demonstrates its ability to reliably determine the causal ancestor and descendant relationships between any two variables in high-dimensional scenarios. Thereby, through pairwise analysis across all variables, we can not only reconstruct the complete causal order in acyclic structures, but also identify cycles directly at the pairwise level as well as the number and location of latent variables. Specifically, when two variables are found to form a cycle, our method yields a distinct output indicating their cyclic relationship. This inherent capability to detect cyclic structures during pairwise analysis represents a key advantage of our framework.
>
> We begin by providing a theoretical analysis to demonstrate that Theorems 1-3 indeed hold in the global context. Subsequently, we validate the theoretical analysis through experimental evaluation in scenarios involving more than two variables.

---

> ### Author Response · Authors · 2025-11-29
> **Response to Reviewer pV6B - Q1 (Part 2/3)**
>
> **Theoretical Guarantee**
>
> First, within the framework of the OICA model where $V = AS$, we clearly classify the noise that affects any two observed variables $X_i$ and $X_j$. As shown below, all independent sources $S$ are divided into the following five mutually exclusive components:
>
> (1) **Shared sources $G_{ij}$**: $G_{ij} = \\{g_k|g_k \in \mathbf{S} \setminus \\{e_i, e_j\\},A_{i,g_k}\neq 0 ,A_{j,g_k}\neq 0\\}$, comprising all sources other than $e_i$ and $e_j$ that exert influence on both $X_i$ and $X_j$.
>
> (2) **Unique sources of $X_i$**: $U_{i} = \\{r_k|r_k \in \mathbf{S} \setminus \\{e_i, e_j\\},A_{i,r_k}\neq 0 ,A_{j,r_k}= 0\\}$.
>
> (3) **Unique sources of $X_j$**: $U_{j} = \\{n_k|n_k \in \mathbf{S} \setminus \\{e_i, e_j\\},A_{i,n_k} = 0 ,A_{j,n_k} \neq 0\\}$.
>
> (4) **Intrinsic noise of $X_i$**: $e_i$
>
> (5) **Intrinsic noise of $X_j$**: $e_j$
>
> For this proof, we consider the causal direction $X_i\rightarrow X_j$. Then, $X_i$ and $X_j$ are expressed as:
> \\begin{equation}
>     X_i = \\sum_{k=1}^{n=|G_{ij}|} A_{i,g_k} g_k + \\sum_{k=1}^{|U_{i}|} A_{i,r_k} r_k + e_i
> \\end{equation}
> \\begin{equation}
>     X_j = \\sum_{k=1}^{|G_{ij}|} A_{j,g_k} g_k + \\sum_{k=1}^{|U_{j}|} A_{j,r_k} r_k + A_{ji}e_i+ e_j
> \\end{equation}
>
> Under this formulation, the joint-cumulant is employed to capture the cumulative effect of the shared noise. Therefore, the $k$ order joint-cumulant between $X_i$ and $X_j$ is given as follows:
> \\begin{aligned}
> \\begin{bmatrix}
>      C_{k-1,1}(X_i,X_j)\\\\
>      C_{k-2,2}(X_i,X_j)\\\\
>      \\vdots\\\\
>      C_{1,k-1}(X_i,X_j)
> \\end{bmatrix} &=
> \\begin{bmatrix}
>      A_{i,g_1}^{(k-1)}A_{j,g_1} & A_{i,g_2}^{(k-1)}A_{j,g_2} & \\cdots & A_{i,g_n}^{(k-1)}A_{j,g_n} & A_{ji}\\\\
>      A_{i,g_1}^{(k-2)}A_{j,g_1}^2 & A_{i,g_2}^{(k-2)}A_{j,g_2}^2 & \\cdots & A_{i,g_n}^{(k-2)}A_{j,g_n}^2 & A_{ji}^2\\\\
>      \\vdots & \\vdots & \\ddots & \\vdots & \\vdots\\\\
>      A_{i,g_1}A_{j,g_1}^{(k-1)} & A_{i,g_2}A_{j,g_2}^{(k-1)} & \\cdots & A_{i,g_n}A_{j,g_n}^{(k-1)} & A_{ji}^{(k-1)}
> \\end{bmatrix}
> \\begin{bmatrix}
>      C_{k}(g_1)\\\\
>      C_{k}(g_2)\\\\
>      \\vdots\\\\
>      C_{k}(g_n)\\\\
>      C_{k}(e_i)
> \\end{bmatrix}
> \\end{aligned}
> It can be observed that the mathematical structure of the derived cross-cumulants
> $ \\{C_{k-1,1}(X_i,X_j), C_{k-2,2}(X_i,X_j),\dots, C_{1,k-1}(X_i,X_j)\\}$ between $X_i$ and $X_j$ is identical to that obtained under the classical causal model where two observed variables share $n$ latent confounders with a direct causal relationship $X_i\rightarrow {X_j}$.
>
> This equivalence in the cross-cumulant patterns arises because both models account for sources of shared randomness in a similar manner. In the classical latent variable causal model, the shared stochasticity originates from both the shared latent confounders and the propagated noise from the cause variable to the effect variable, while in our OICA framework, the shared randomness is precisely categorized into the shared sources $G_{ij}$ and the propagated intrinsic noise $e_i$ from $X_i$ to $X_j$.
> Although the two models differ in their interpretation and classification of noise sources, they generate mathematically equivalent cross-cumulant structures. Due to this equivalence in the cross-cumulant patterns, the core argument in Theorem 2 for identifying the causal direction remains valid. The detailed proof, which primarily relies on analyzing the linear dependencies among cross-cumulants of different orders, is provided in Appendix A.2. The same line of reasoning applies to scenarios involving independent and cyclic causal relations.

---

> ### Author Response · Authors · 2025-11-29
> **Response to Reviewer pV6B - Q1 (Part 3/3)**
>
> **Experimental Results**
>
> To evaluate the performance of our method in scenarios involving more than two variables, we conduct experiments under the following experimental setup: a system of five observed variables with one latent confounder, as illustrated in the following equations, using exponentially distributed noise. Each dataset contained 10,000 samples, and causal order determination employed a rank-test threshold of 0.01. The experiment was repeated over 20 independently generated datasets, with the average across all trials reported in Table 1.
>
> \begin\{split\}
> \begin\{bmatrix\}
>       X\_1\\\\
>       X\_2\\\\
>       X\_3\\\\
>       X\_4\\\\
>       X\_5
> \end\{bmatrix\} &=
> \begin\{bmatrix\}
>       0 & b\_\{12\} & 0 & 0 & 0\\\\
>       b\_\{21\} & 0 & 0 & 0 & 0\\\\
>       0 & b\_\{32\} & 0 & 0 & 0\\\\
>       0 & b\_\{42\} & 0 & 0 & 0\\\\
>       0 & 0 & b\_\{53\} & b\_\{54\} & 0
> \end\{bmatrix\}
> \begin\{bmatrix\}
>       X\_1\\\\
>       X\_2\\\\
>       X\_3\\\\
>       X\_4\\\\
>       X\_5
> \end\{bmatrix\} +
> \begin\{bmatrix\}
>       0\\\\
>       0\\\\
>       1\\\\
>       0\\\\
>       1
> \end\{bmatrix\} L +
> \begin\{bmatrix\}
>       E\_\{X\_1\}\\\\
>       E\_\{X\_2\}\\\\
>       E\_\{X\_3\}\\\\
>       E\_\{X\_4\}\\\\
>       E\_\{X\_5\}
> \end\{bmatrix\}
> \end\{split\}
>
>
>
> **Table 1.** Precision, recall and f1-scores of our Method and baseline methods.
> | | DirectLiNGAM | ReLiNGAM | CCI | Ours |
> |--------|--------------|----------|-----|------|
> | Precision | 0.68 | 0.31 | 0.30 | **0.84**|
> | Recall | **0.86** | 0.39 | 0.38| 0.76 |
> | F1 | 0.76 | 0.35 | 0.34 | **0.80** |
>
> The experimental results demonstrated that our method outperforms ReLiNGAM and CCI. Notably, a significant advantage of our approach lies in its ability to identify cycles and latent confounders—an achievement that the baseline method, DirectLiNGAM, is unable to accomplish. Consequently, while DirectLiNGAM achieves the highest recall, it exhibits lower precision compared to our method.

---

> ### Author Response · Authors · 2025-11-29
> **esponse to Reviewer pV6B - Q2, Q3, Q4 & Q5**
>
> **(Q2)**: High-order cumulants are statistically noisy; the method needs large sample sizes before ranks stabilize. The paper gives little guidance on regularization/thresholding for near-rank decisions.
>
> **A2:** The accuracy of our method fundamentally depends on the accuracy of the cumulative quantity estimation. The core of our approach involves performing Singular Value Decomposition (SVD) on these matrices. The rank of a matrix is then determined by counting the number of singular values whose absolute magnitude exceeds a predefined threshold. Since the statistical variability of empirical cumulants decreases with increasing sample size, we employ a sample-size-dependent threshold function of the form $\alpha =\frac{c}{\sqrt[3]{N}}$, where the constant $c$ is typically set to 2. However, it is important to note that selecting an excessively large threshold is potentially risky, as it may lead to increased false positives in causal discovery. To ensure robustness, particularly with small sample sizes, we impose a conservative upper bound of 0.2 on the threshold value. For example, in our small-sample experiments with sample size $N = 200$, we use a threshold of 0.2 instead of the larger value suggested by the formula. This design guarantees reliable performance across different sample sizes while maintaining an appropriate balance between sensitivity and specificity.
>
> ---
>
> **(Q3)** The manuscript needs a careful language edit.
>
> **A3:** We have carefully revised the language throughout the manuscript to ensure clearer, more accurate expression.
>
> ---
>
> **(Q4)** How do you set numerical thresholds for deciding ranks of $J$ and $M$ from noisy cumulant estimates? Any bootstrap, or shrinkage scheme to stabilize near-boundary cases?
>
> **A4:** The core of our approach involves performing Singular Value Decomposition (SVD) on the matrices $J$ and $M$. To determine their ranks, we count the number of singular values whose absolute magnitude exceeds a predefined threshold.
> This threshold is chosen empirically based on the sample size of the data. Specifically, we employ a sample-size-dependent threshold function of the form $\alpha =\frac{c}{\sqrt[3]{N}}$, where the constant $c$ is typically set to 2 and $N$ denotes the sample size. While we do not explicitly employ bootstrap or shrinkage schemes in the current implementation, these techniques could indeed be valuable for stabilizing rank decisions in near-boundary cases, especially when the singular values are close to the threshold. Incorporating such strategies is an interesting direction for future work to further enhance the robustness of our method.
>
> ---
> **(Q5)**  The algorithm increases $m$ and adjusts $k_1,k_2$ Why this schedule, and how sensitive is performance to alternate choices (e.g. start with m=1, k1=3, k2=4, using multiple $k_1$, $k_2$ and aggregating decisions)? Please clarify lines 4-26 of Algorithm 1.
>
> **A5:**
> Our algorithm is designed to operate without requiring prior knowledge of the number of latent confounders and the presence of cycles. To achieve this, it employs an iterative search strategy to simultaneously determine the number of latent confounders and uncover the causal structure. In our work, $m$ represents the hypothesized number of latent confounders. Since the true number of latent confounders is unknown, the algorithm begins exploration from $m = 1$. Specifically, the algorithm progressively increases the hypothesized value of $m$ while adjusting the cumulant orders $k_1$ and $k_2$ accordingly to systematically explore higher-order statistical characteristics in the data. When starting from $m = 1$, we set $k_1 = m + 2 = 3$ and $k_2 = 2m + 2 = 4$ to construct the moment matrices $M_X$ and $M_Y$, as well as the joint statistic matrix $J$ (Line 3).
> If the specified value of $m$ is smaller than the true number of latent confounders, the conditions statements are violated, where the matrix ranks become $rank(J^{(k_1,k_2)})=m+1$, $rank(M_X^{(k_1,k_2)})=rank(M_Y^{(k_1,k_2)})=m+2$. As a result, the algorithm bypasses the core causal discovery routines (Lines 6-16). It proceeds to Line 17, where the matrices $M_X^{(k_1,k_2)}$ and $M_Y^{(k_1,k_2)}$ are expanded to $M_X^{(k_1+1,k_2+1)}$ and $M_Y^{(k_1+1,k_2+2)}$, increasing their effective rank to $m+2$, which fails to satisfy the rank condition in Line 18. Then this automatically triggers an update where $m$ is incremented by 1, and $k_1$ and $k_2$ are adjusted for the next round of exploration (Lines 23 - 25). This process continues until $m $matches the actual number of confounders, at which point the statistics constructed with the corresponding $k_1$ and $k_2$ satisfy the identifiability conditions, and the algorithm outputs a causal structure. Notably, our algorithm is also applicable starting from $m=0$, meaning that even in the absence of latent variables, our method remains valid. In this case, all condition sets are empty.

---

> ### Author Response · Authors · 2025-11-29
> **Response to Reviewer pV6B - Q6**
>
> **(Q6)** For $p>2$ observables, how do you ensure a globally consistent graph when pairwise decisions disagree?\\
>
> **A6:** When dealing with more than two observed variables, our method ensures consistency in pairwise decision-making, avoiding any contradictions. This robustness stems from our theorem, which accounts for the presence of both cycles and latent variables. For any two variables, $X$ and $Y$, all their shared ancestor nodes are treated as latent variables during pairwise analysis. Using our method, the causal order between $X$ and $Y$ can still be uniquely identified. Even in cases where a cycle exists between two variables, our approach can still yield correct results. By integrating all pairwise results, we can construct the global causal order and identify structures that include cycles. The theoretical analysis and the experimental result for the case of $p>2$ observables can be referred to $Q1A1$.
>
> ---
>
> We want to thank the reviewer again for all the valuable comments.

---

### Author Response · Authors · 2025-12-03
**Summary**

Dear Area Chair,

Thank you for taking over the meta-review!

Our paper addresses the limitations of current causal discovery methods in simultaneously handling cyclic causality and latent confounders. We propose a unified higher-order cumulant-based framework that establishes systematic identifiability theories under linear non-Gaussian assumptions. By analyzing the rank properties of two constructed cumulant matrices $ J^{(k_1,k_2)} $ and $ M^{(k_1,k_2)} $, our method can **uniformly distinguish five causal patterns, including direct causation, cycles, and latent confounding**. Notably, our theorems can **determine causal directions for any variable pair and be extended to infer global causal order**. The approach requires only observational data, maintains computational efficiency, and provides a robust tool for causal discovery in complex systems with feedback loops and unobserved confounders.

Below is a consolidated summary of the clarifications and additions incorporated across the revised manuscript.

**1.Scalability to general graph**

In response to the reviewers' question regarding how to extend bivariate causal discovery to multivariate scenarios, our approach is based on a core property: Theorems 1–3 remain valid for any variable pair within the global causal graph. This enables us to directly determine the causal order between any two variables through systematic pairwise analysis without altering the core framework, thereby obtaining the global causal order. Moreover, the method inherently embeds the capability to detect cyclic structures during pairwise analysis—when two variables form a cyclic relationship. It outputs a clear indication of such cyclic interaction. Additionally, we have added experimental results on multivariate data, empirically validating the scalability and consistency of our approach in multivariate causal discovery.

**2.New experiments with small sample sizes and violation of assumptions**

To address the reviewers' concerns regarding the impact of sample size and violation of the assumptions on the accuracy of our proposed method, we have conducted additional experiments with sample sizes of 200, 500, and 1000, as well as a nonlinear case. The results demonstrate that our proposed method maintains effective causal identification performance even with a sample size as small as 200 and nonlinear scenarios.

All new results are included in the appendix due to space constraints.

We have implemented all reviewer-requested changes across the manuscript, including the theorems for extending our method to the multivariate case, notations clarification, typos and the addition of multiple new experimental components. We believe the revised manuscript addresses all concerns comprehensively.

Thank you very much for your time and consideration!

---

### Meta-Review · Area_Chair_QJ5L · 2025-12-28

**Summary:**

This submission proposes a higher-order cumulant, rank-based approach for identifying pairwise causal relations in linear non-Gaussian models that may include both feedback cycles and latent confounders. The core contribution is a set of identifiability claims (via rank relationships between constructed cumulant matrices) that aim to distinguish several bivariate causal patterns (directed edge, cycle, latent confounding, and absence of a direct edge), together with an iterative procedure that searches over cumulant orders and a hypothesized number of latent confounders. The rebuttal clarifies the intended extension to multivariate settings by asserting that the pairwise theorems apply within a larger graph, and it adds limited multivariate and small-sample experiments, as well as practical guidance on SVD-based rank thresholding and runtime.

**Reviewer Concerns:**

Across the reviews, there is agreement that the bivariate theoretical development is interesting, but two reviewers raise concerns that are central to acceptance at this venue. The main concern is that the paper’s claims about extending beyond the bivariate setting are not yet convincingly realized as a complete, globally consistent multivariate discovery procedure with clear guarantees; the rebuttal argues that pairwise decisions can be integrated into a global causal order and that cycles can be detected pairwise, but it does not fully specify a conflict-resolution/integration algorithm nor provide a rigorous end-to-end guarantee for general graphs with multiple observed variables, cycles, and latent confounding. A second concern is robustness and reliability in practice: rank decisions based on higher-order cumulants can be statistically fragile, and while the rebuttal introduces an SVD threshold heuristic and additional small-sample results, the overall methodology still depends on tuning/thresholding choices and finite-sample behavior that may not be sufficiently characterized, especially in near-degenerate or misspecified settings. Relatedly, one reviewer explicitly questioned whether incorrect cumulant orders could spuriously trigger the rank scenarios; the rebuttal describes an iterative search behavior, but the remaining risk is that the method’s correctness may be sensitive to order selection and to the stability of the rank tests under noise and model deviations. Finally, although additional experiments were provided (including a 5-variable synthetic setting and smaller sample sizes), the empirical evaluation remains limited in scope for the breadth of the claims (e.g., restricted graph families, limited diversity in latent structure, and modest evidence for scalable multivariate performance), which leaves uncertainty about general applicability and practical competitiveness beyond the carefully controlled settings.
Given these remaining issues—especially the incomplete multivariate story and the practical sensitivity of rank-based cumulant tests—the rebuttal improves clarity but does not fully eliminate the primary reasons two reviewers were below threshold. On balance, I recommend rejection at this stage, while viewing the work as promising if the multivariate integration/guarantees and robustness characterization are strengthened in a future version.

**Reviewer Scores:**

Based on the rebuttal content—particularly the added multivariate and small-sample experiments and the clarified SVD-threshold procedure—I estimate that Reviewer pV6B would likely move to a 5 (marginally above threshold) while keeping confidence around 4, Reviewer ZTRC could move to a 5 but may also reasonably remain at 4 given their emphasis on missing theoretical guarantees under misspecified orders (confidence likely remaining 5), and Reviewer 2dUj would likely remain at 8 with similar confidence. Even under this optimistic inferred update, the overall reviewer stance remains split with unresolved core concerns from the higher-confidence borderline reviewer, which supports a reject recommendation.

---

### Decision · Program_Chairs · 2026-01-26

Reject